# *SMARCB1* missense mutants disrupt SWI/SNF complex stability and remodeling activity

Garrett W. Cooper[1,2], Benjamin P. Lee [1,2], Won Jun Kim[3,4], Yongdong Su [1,2], Victor Z. Chen[1,2], Eliseo Salas[5], Xiaoping Yang [4], Robert E. Lintner [4], Federica Piccioni [4], Andrew O. Giacomelli [6], Thomas P. Howard[3], Pritha Bagchi [7], Karen N. Conneely [8], David E. Root [4], Bo Liang [5], James C. Gumbart [9], William C. Hahn [3,4], David U. Gorkin [10], Jaclyn A. Biegel[11], Susan N. Chi [3,12] & Andrew L. Hong [1,2,13] ✉

Chromatin remodeling complexes, such as the SWItch/Sucrose Non-Fermentable (SWI/SNF) complex, play key roles in regulating gene expression by modulating nucleosome positioning. The core subunit SMARCB1 is essential for these functions, as it anchors the complex to the nucleosome acidic patch, enabling effective chromatin remodeling. While biallelic inactivation of *SMARCB1* is a hallmark of several aggressive pediatric malignancies, the functional implication of missense mutations is not fully understood. Current diagnostic approaches focus on detecting the presence or absence of SMARCB1 by immunohistochemistry often without consideration of mutation status. Here, we present a comprehensive deep mutational scanning of *SMARCB1*, encompassing 8418 alterations, to assess their functional impact. We show that RPT2 missense mutations disrupt SMARCB1 antiproliferation function by destabilizing the SWI/SNF complex and impairing chromatin remodeling and transcriptional regulation comparable to nonsense mutations. These functional defects occur despite maintaining detectable protein expression thereby challenging current diagnostic reliance on IHC. These findings provide deeper understanding of the role of SMARCB1 in chromatin remodeling and cancer biology, highlighting limitations of mutation classification approaches.

The SWItch/Sucrose Non-Fermentable (SWI/SNF) chromatin remodeling complex, also known as the BRG1/BRM-associated factors (BAF) complex, regulates DNA accessibility and gene expression by positioning nucleosomes at promoters and enhancers[1]. One of the core complex subunits, SMARCB1 (also known as INI1 or BAF47), is required for regulation of enhancers[2,3]. Recent structural studies reveal that SMARCB1, in part, accomplishes this by anchoring the complex to the nucleosome acidic patch, thereby facilitating the remodeling activity of the SWI/SNF complex[4].

Biallelic inactivation of *SMARCB1* is found in over 90% of several aggressive pediatric malignancies including malignant rhabdoid tumor of the kidney (MRTK), atypical teratoid rhabdoid tumor (ATRT),

[1]Department of Pediatrics, Emory University School of Medicine, Atlanta, GA, USA. [2]Aflac Cancer and Blood Disorders Center—Children's Healthcare of Atlanta, Atlanta, GA, USA. [3]Dana-Farber Cancer Institute, Boston, MA, USA. [4]Broad Institute of MIT and Harvard, Cambridge, MA, USA. [5]Department of Biochemistry, Emory University School of Medicine, Atlanta, GA, USA. [6]Humber Polytechnic, Toronto, ON, Canada. [7]Emory Integrated Proteomics Core, Emory University, Atlanta, GA, USA. [8]Department of Human Genetics, Emory University School of Medicine, Atlanta, GA, USA. [9]School of Physics and School of Chemistry and Biochemistry, Georgia Institute of Technology, Atlanta, GA, USA. [10]Department of Biology, Emory University, Atlanta, GA, USA. [11]Department of Pathology, Children's Hospital Los Angeles and Keck School of Medicine, University of Southern California, Los Angeles, CA, USA. [12]Boston Children's Hospital, Boston, MA, USA. [13]Winship Cancer Institute, Atlanta, GA, USA. ✉e-mail: andrew.hong2@emory.edu

and renal medullary carcinomas (RMC)[5,6]. Rare SMARCB1-deficient cancers have also been observed in adults such as epithelioid malignant peripheral nerve sheath tumor, myoepithelial carcinoma, and poorly differentiated chordoma[7]. Diagnosis of SMARCB1-deficient cancers largely relies on the absence of detection of SMARCB1 through immunohistochemistry (IHC)[8,9]. The role of SMARCB1 in pediatric cancers has been well characterized as a tumor suppressor; however in adult cancers, its role appears more complex, with studies suggesting that SMARCB1 may also act as an oncogene in certain contexts, including liver cancer and melanoma[10,11]. However, beyond a few well studied examples like SMARCB1 R377H that disrupt SWI/SNF remodeling function[4], the functional impact of most *SMARCB1* missense mutations remains unknown. This knowledge gap creates a significant challenge for clinicians interpreting novel *SMARCB1* variants encountered in cancer genetic testing.

Three primary strategies have been developed to classify variants of uncertain significance (VUS). The first leverages large-scale sequencing data to identify patterns of mutation recurrence across patient cohorts[12]. While powerful for frequently mutated genes, this approach is underpowered for low-frequency genes like *SMARCB1*, which exhibits a somatic mutation rate of 0.9% across 239 cancer types compared to 39.6% for *TP53*[13]. The second strategy employs computational models that integrate evolutionary conservation, protein structure, physiochemical properties, and functional domain annotations to predict pathogenicity. Despite these advances, computational models prioritize general pathogenicity prediction over specific phenotypic modeling[14], such as impacts on protein stability, enzymatic activity, or chromatin remodeling. The third strategy employs experimental assays to directly measure the functional consequences of variants one by one on defined cellular phenotypes, including proliferation, metabolism, or pathway-specific activity. While highly informative, such approaches are technically challenging and resource-intensive, limiting their application to rare mutations[15]. Deep mutational scanning (DMS) addresses these limitations by systematically evaluating the function of variants in a gene within a defined phenotypic context. This approach has proven successful for highly mutated oncogenes and tumor suppressors such as *TP53*, *BRCA1*, *KRAS*, and *EGFR*[16–19], providing insights that inform both mechanistic understanding and clinical variant interpretation.

Here, we perform DMS of *SMARCB1* by systematically evaluating the functional consequences across its entire coding sequence using a proliferation-based phenotypic framework. While recognizing that mutations in chromatin remodeling genes may contribute to tumorigenesis through multiple mechanisms, we employ proliferation as a quantifiable and clinically relevant proxy for malignant potential. Through this analysis, we identify missense mutations that, despite retaining protein expression, disrupt SWI/SNF complex composition and chromatin remodeling activity, thereby impairing SMARCB1's antiproliferative function. These findings have important clinical implications, as they demonstrate that certain loss-of-function mutations may evade detection through standard IHC based diagnostic approaches.

## Results

### Mutational landscape and functional diversity of *SMARCB1* variants

The integration and sharing of genomic and clinical data have generated a powerful resource to detect mutational trends across a broad spectrum of cancer-associated genes. We analyzed data from two leading consortia, AACR Project GENIE v16.1 and COSMIC v100, to explore recurrent missense mutations in *SMARCB1* across its key functional domains (Fig. 1a). Both datasets identified low-frequency missense mutations (affecting fewer than 10 patients) distributed across the entire coding sequence of *SMARCB1*. The AACR GENIE dataset identified high-frequency mutations (affecting more than 10

patients) in 10 specific residues, particularly within the coiled-coiled domain (CCD) at residues R374 and R377. The most frequent of these mutations, R377H, has been shown to disrupt the remodeling function of the SWI/SNF complex; however, its role in disrupting the antiproliferative function of SMARCB1 has not been characterized[4].

To experimentally interrogate the antiproliferative function of *SMARCB1* variants, we used a previously described inducible SMARCB1 re-expression system in the SMARCB1-deficient G401 cell line[20]. We defined functional reference points by comparing wild type SMARCB1 (65.5% ± 13.1% growth suppression) against a patient-derived ATRT nonsense mutation[21], W281* (28.7% ± 4.2% growth suppression), establishing upper and lower bounds of antiproliferative activity (Supplementary Fig. 1a−c).

To investigate the recurrent R377H variant, we re-expressed the corresponding silent mutant, R377R, the missense mutant, R377H, and the nonsense mutant R377* variants in G401 cells and compared proliferation upon induction (Supplementary Fig. 1d). Compared to the R377R control (74.5% ± 6.8% growth suppression), both R377H and R377* showed reduced antiproliferative function with 62.0% ± 10.3% and 62.0% ± 4.0% suppression, respectively (Supplementary Fig. 1e, f). While both variants showed a reduction in antiproliferative activity, both retained substantially more activity than the ATRT-derived W281* nonsense mutation (28.7% ± 4.2%). These results demonstrate that SMARCB1 antiproliferative function can be disrupted to different degrees, with R377H showing only modest functional impairment. These findings may be consistent with clinical observations that individuals with Coffin-Siris syndrome harboring germline R377H mutations (ClinVar Accession: VCV000030203.16) do not develop cancers at higher rates than the population despite having other phenotypic effects[22].

To assess computational predictions of *SMARCB1* variant pathogenicity, we analyzed predictions from CADD, AlphaMissense, and REVEL[23–25], which classify variants based on general pathogenicity rather than specific phenotypic outcomes (Supplementary Fig. 1g−i, respectively). These tools classified 76.5%, 96.5%, and 37.8% of missense variants as deleterious, respectively (Fig. 1b, Supplementary Fig. 1j, k, *Methods*). However, predictions for R377H did not align with our experimental observations. R377H ranked in the 90th, 97th and 82nd percentiles for CADD, AlphaMissense and REVEL, respectively, yet exhibited only intermediate functional impairment in our proliferation assays.

The complexity of individual variant effects and the limitations of computational predictions highlight the need for systematic functional characterization of all possible *SMARCB1* missense mutations[14]. To address this, we characterized the impact of nearly all *SMARCB1* missense mutations on antiproliferative function through comprehensive DMS.

### Deep mutational scanning of *SMARCB1* coding sequence

To validate the feasibility of a large-scale open reading frame (ORF) screen in SMARCB1-deficient cell lines such as a DMS screen, we introduced the ORFeome V8.1 Library composed of 16,100 ORF's, including *SMARCB1*, in the G401 cell line at a low multiplicity of infection (MOI) (*Methods*). Our results showed that cells with wild type *SMARCB1* were negatively selected with a z-score of −3.12 (Supplementary Fig. 1l). We then generated a DMS library of *SMARCB1* variant 1 (ENST00000644036.2), the predominant wild type (WT) isoform, under a constitutive EF1-α promoter, comprising 8418 alterations (corresponding to 24,709 nucleotide variants) including silent, missense, frameshift, and nonsense mutations (*Methods*). We observed SMARCB1 protein expression levels approximately 2.2-fold higher than endogenous levels in normal kidney tissue cell line 2494N, but comparable to levels seen in Wilms tumor cell line 2494T (1.6-fold difference) (Supplementary Fig. 1m)[26]. We employed a proliferation-based DMS approach to model *SMARCB1* variant function in 3 patient-derived

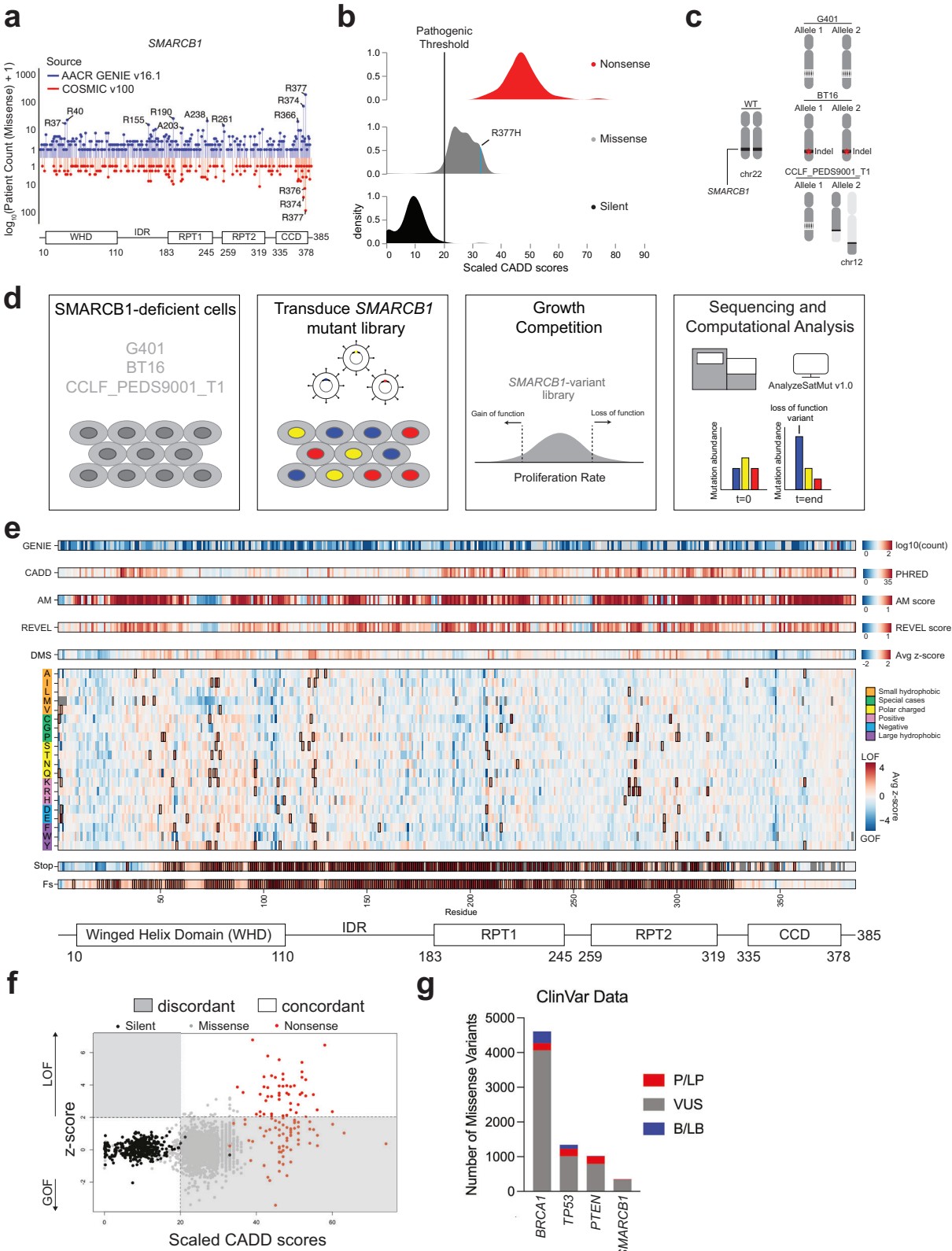

SMARCB1-deficient cell lines: G401−MRTK (biallelic deletions encompassing *SMARCB1*), BT16−ATRT (biallelic indel in exon 1), and CCLF_PEDS9001_T1−RMC[20] (monoallelic deletion with a disruptive balanced translocation in *SMARCB1*) (Fig. 1c, Supplementary Fig. 1n).

We introduced the *SMARCB1* DMS library into the three SMARCB1-deficient cell lines at a low MOI (0.2−0.5) (Fig. 1d; *Methods*). Fitness of these mutants based on proliferation was carried out for 8−14 days to

reflect 4−8 doublings, and we used AnalyzeSaturationMutagenesis v1.0 to assess variant abundance following fitness competition[27]. To account for regional technical artifacts where variants showed systematic shifts in log2 fold-change values (Supplementary Fig. 1o), we applied a rolling z-score normalization that compares each variant to local silent mutation behavior rather than global averages, under the assumption that silent mutations have no effect on SMARCB1 antiproliferative function

**Fig. 1 | Deep Mutational Scanning of *SMARCB1* Reveals Novel Regions of Mutational Intolerance in *SMARCB1*. a** Observed log10 mutational frequency of all cancer associated missense mutations in *SMARCB1* coding sequence based on AACR GENIE v16.1 and COSMIC v100. Functional domains of SMARCB1 denoted below. **b** Density plots of all CADD v1.7 mutations categorized by mutation type. Pathogenic threshold applied at a PHRED value of 20. **c** *SMARCB1* alterations present in each cell line used in the study. G401 contains biallelic deep deletion of *SMARCB1*. BT16 has a biallelic 1nt frameshift indel in exon 1. CCLF_PEDS9001_T1 has a monoallelic deep deletion and a balanced translocation in chromosome 12. **d** Experimental setup for deep mutational scan of *SMARCB1* including analysis steps. **e** Comparison of *SMARCB1* deep mutational scanning (DMS) data with patient mutation frequencies and computational predictions. The top tracks display patient mutation frequency from the GENIE database (log10 transformed

counts) and computational deleteriousness predictions from CADD (PHRED scores), AlphaMissense (AM pathogenicity), and REVEL. The DMS average track reflects mean z-scores of all possible 19 amino acid substitutions at each position across all three cell lines. In the central heatmap, each row represents a specific amino acid substitution, color-coded by residue type (see legend). The bottom two rows reflect the z-scores from stop and frameshift mutations. Missing values are denoted with a gray box and mutations which reached significance (average z-score > 2) are denoted with a black outline. Domain structure of SMARCB1 is displayed at the bottom. **f** Scatterplot of all mutations SNVs predicted by CADD and their corresponding average z-score in DMS screen. Mutations colored by mutation type. **g** ClinVar variants (December 2, 2024 release) observed in *BRCA1, TP53, PTEN, and SMARCB1*.

(*Methods*). Variants with a z-score >2 (or $p < 0.0455$) were considered LOF. Variant z-scores were averaged across the three cell lines to identify alleles with consistent functional effects (Fig. 1e, Supplementary Fig. 2a). The variability observed among missense mutations may reflect both biological sensitivity to specific substitutions and technical noise inherent to the screening approach. An interactive heatmap of DMS results is available at https://thehonglab.github.io/SMARCB1_DMS/. All code is archived on Zenodo[28].

We found that integrating our DMS dataset with the corresponding SNVs from the CADD dataset revealed largely concordant predicted functional outcomes for silent and nonsense mutants (Fig. 1f). Specifically, 44.1% (52/118) of nonsense mutants showed concordant functional predictions. Similarly, both datasets concordantly identified 99.2% (362/365) of silent mutants as non-pathogenic. However, for missense mutations, there was discordance, with only 4.18% (95/2274) of these mutations having concordant functional predictions between the datasets.

To contextualize our data within the framework of currently recognized pathogenic mutations, we examined the number of established pathogenic missense mutations in three well-characterized tumor suppressors, *BRCA1*, *TP53*, and *PTEN*, as compared to *SMARCB1* (Fig. 1g, Supplementary Fig. 2b, c). We found that only 12 missense mutations in *SMARCB1* were classified as pathogenic or likely pathogenic, with two of these classified under hereditary cancer-predisposing syndrome (P14H and R53L); however, clinically these variants have been associated with schwannomatosis, a non-malignant tumor syndrome[29-31]. The remaining mutations were associated with intellectual disability ($n = 6$), NK-cell enteropathy ($n = 1$), or not provided ($n = 3$). In comparison, *BRCA1*, *TP53* and *PTEN* had a higher number of missense variants classified as pathogenic/likely pathogenic. Specifically, *BRCA1* had 3.6% (209/5807) pathogenic variants, 203 of which were linked to cancer pathogenesis; *TP53* had 15.9% (214/1348) with 212 variants associated with cancer; and *PTEN* had 21.75% (223/1025), with 148 associated with cancer.

## Functional scores of 8418 *SMARCB1* alterations

To understand how each mutation type (silent, missense, frameshift, and nonsense) impacted SMARCB1 function in our screen, we plotted the distribution of functional scores across mutation types (Fig. 2a). As anticipated, nonsense and frameshift mutants generally exhibited increased fitness compared to the baseline WT function, consistent with the expectation that truncated proteins often lack functional activity. We observed that both N- and C-terminal nonsense mutants showed either full or partial functional activity (Fig. 2b). C-terminal nonsense and frameshift mutants starting at residue 350 retained partial functionality, suggesting that these truncations preserve critical domains required for antiproliferative activity. This finding is consistent with the previously characterized R377* mutant, which had a z-score of 0.296 and correspondingly achieved a mean $62\% \pm 4.0\%$ decrease in proliferation upon re-expression over 8 days (Supplementary Fig. 1d–f).

Although missense mutations showed substantial overlap with silent mutations in contrast to CADD predictions (Fig. 2a), a subset may be functionally disruptive. Proline and lysine substitutions exhibited the most pronounced deleterious effects, displaying both the highest mean z-scores and severe functional impairment (z-score > 2) at the greatest number of residues (11 each) (Supplementary Fig. 2d). While these patterns suggest specific biochemical constraints, we note that technical variability in the screen necessitates experimental validations of individual variants to confirm functional effects. Applying a LOF z-score threshold of >2.0 identified 101 candidate deleterious missense variants (Fig. 2c). Notably the recurrent R377H missense mutant fell below this threshold (z-score = 0.985), consistent with our earlier findings (Supplementary Fig. 1d–f). To systematically identify mutation-intolerant positions, we ranked each residue by the mean z-score across all 19 possible missense substitutions. Using a cutoff of two standard deviations above the mean (residue averaged z-score >0.679), we identified 13 highly intolerant residues: six within the WHD domain (R52, A55, I63, K77, L90, L91), three within the IDR (E122, Q123, and A125), and four within the RPT2 domain (D277, W281, E300, and I315) (Fig. 2d, Supplementary Fig. 2e). Structural mapping revealed that β-sheet residues exhibited higher amino acid-specific selectivity and buried residues (relative solvent accessibility (RSA) < 0.1) showed higher overall mutational intolerance (e.g., higher average z-score), indicating that mutation effects depend on both structural context and substitution identity (Supplementary Fig. 2f).

## Functional and structural insights in mutation-intolerant regions of SMARCB1

The first of these mutation intolerant regions, the WHD, adopts distinct conformations in the two SMARCB1-containing subfamilies of SWI/SNF complexes: cBAF (canonical BRG1-associated factor) and PBAF (polybromo-associated BRG1-associated factor). In cBAF, the WHD is positioned distal to the nucleosome[32], whereas in PBAF it is proximal[33] (Supplementary Fig. 3a). Mutation-intolerant residues were mapped onto the cryo-EM structure of the PBAF complex, revealing that several intolerant residues, including R52, are located at positions that place them in spatial proximity to DNA in this structural model (Supplementary Fig. 3b).

Given the incomplete resolution of certain residues in available cryo-EM SMARCB1 structures, particularly in the IDR, we used the AlphaFold v2.0 computationally-predicted SMARCB1 structure to comprehensively map mean functional scores for all possible missense mutations[34]. SMARCB1 consists of two globular functional domains—one at each terminus—connected by an IDR. A cluster of mutation-intolerant residues was identified within the IDR at positions 122,123, and 125, which did not display obvious sequence-based functional characteristics.

Additional mutation-intolerant residues were observed within the RPT2 domain. Structural analysis revealed that these intolerant residues (D277, W281, E300, I315) are positioned in close three-dimensional proximity (Fig. 2e), with residues W281 and I315

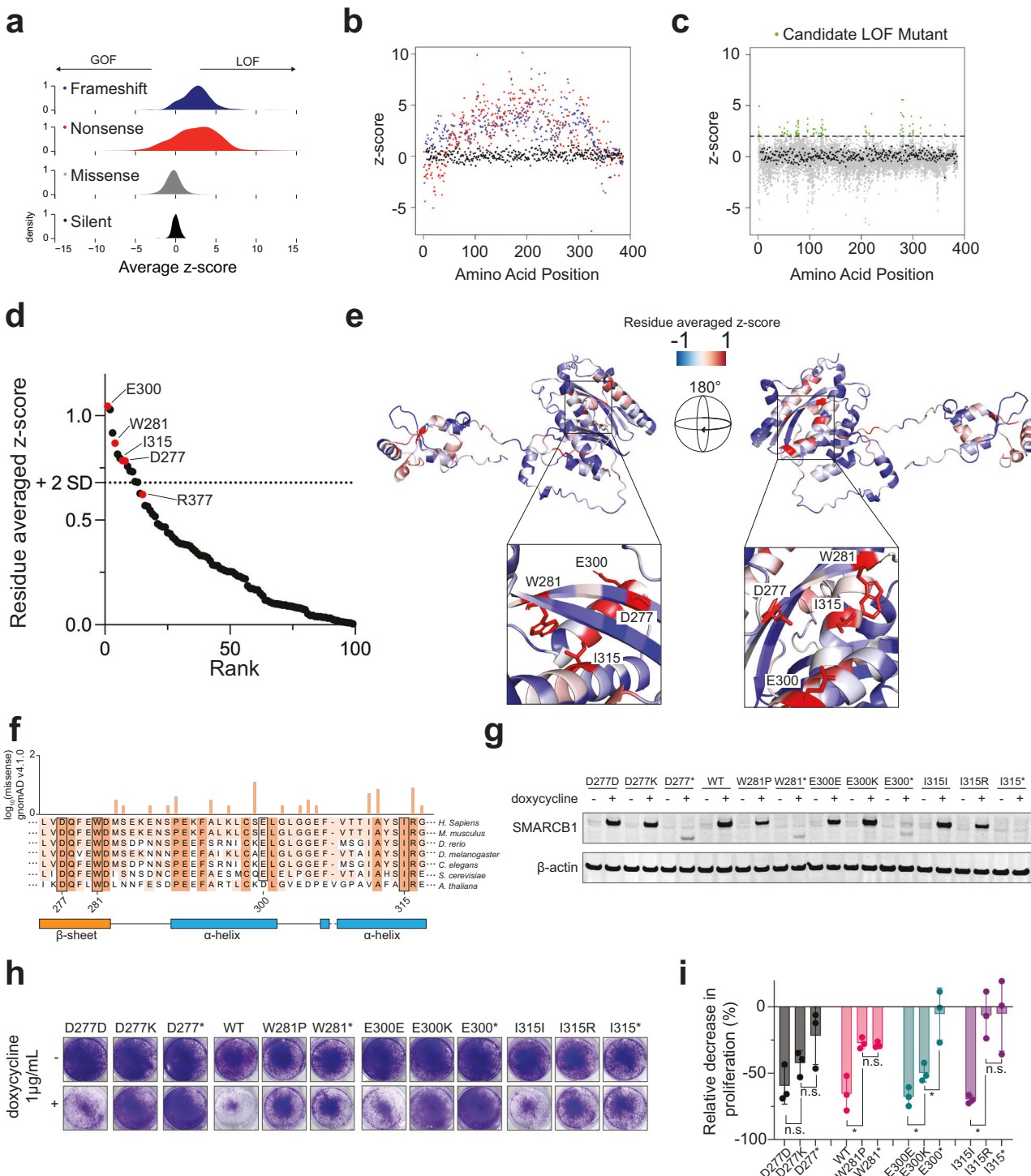

positioned within 4.2 Å of each other (Supplementary Fig. 3c). This region exhibits high evolutionary constraint based on strong conservation across eukaryotic species and intolerance to missense mutations in healthy individuals, as reported in gnomAD v4.1.0[35] (Fig. 2f, Supplementary Fig. 4). Based on these observations, we then assessed the mechanism of these candidate LOF missense mutants in the RPT2 domain.

To evaluate the functional impact of the top-hit missense variants within the RTP2 domain (D277K, W281P, E300K, and I315R), we transduced the G401 cell line with inducible vectors encoding the corresponding silent/wild type (hereafter wild type), missense, or nonsense variants. Here, "wild type" refers specifically to the re-

expressed SMARCB1 ORF rather than the parental cell line. While the DMS screen used cell lines from three SMARCB1-deficient cancer contexts to ensure generalizability, subsequent functional assays were focused on the G401 model to study the effects of SMARCB1 re-expression.

Using total protein lysates, we detected protein expression in both the wild type and missense mutants as compared to the uninduced samples (Fig. 2g). Crystal violet staining over a 10-day period showed reduced growth suppression in cells re-expressing missense mutant constructs compared to wild type ($n = 4$) (Fig. 2h, Supplementary Fig. 5a). In a separate 8-day proliferation assay, all missense mutants exhibited impaired antiproliferative function: W281P

**Fig. 2 | In vitro Validation of Top DMS Hits Reveals Novel Missense Mutants Drive LOF Phenotype. a** Density plots showing the functional z-score distribution averaged across all replicates of screen across the 4 different types of mutations: frameshift, nonsense, missense, and silent. A functional z-score >0 denotes mutation enrichment and <0 denotes mutation depletion compared to silent mutations **b** Averaged functional z-score across all replicates for silent (in black), frameshift (in blue), and nonsense mutations (in red) across the length of the *SMARCB1* coding sequence. **c** Averaged functional z-score across all replicates for silent (in black) and missense mutations (in gray) across the length of the *SMARCB1* coding sequence. Candidate LOF missense mutations with an average functional z-score above 2 are highlighted in green. **d** Residue rank plot showing the residue average functional z-score of all possible 19 amino acid substitutions for all residues with a positive value (*n* = 99). Top hits in the RPT2 domain and recurrent cancer associated R377 residue are labeled in red. **e** Structural mapping of residue average functional z-scores focusing on top intolerant residues in RPT2 domain on computationally predicted structure of SMARCB1 from AlphaFold v2.0. Image has been rotated 180° to further visualize interactions. Zoomed interaction of intolerant RPT2 intolerant residues is depicted. **f** Evolutionary conservation of residues 277 to 317 of SMARCB1 across 7 eukaryotic species. Number of observed missense mutations in gnomAD v4.1.0 depicted above. **g** Immunoblot showing inducible expression from total protein lysates in the G401 cell line of the wild type, missense mutant or nonsense mutant for residues D277, W281, E300, and I315 after 48 h of induction with 1ug/mL doxycycline. Data shown are representative of 2 biological replicates. **h** Crystal violet staining of G401 cells after 10 days of induction (*n* = 4 independent biological replicates from separate passages). Crystal violet staining for wild type and W281* are also presented in Supplementary Fig. 1c, n. **i** Cell counts as assessed by trypan blue exclusion after 8 days of induction in the G401 cell line for each construct. Data are presented as mean values ± SD with individual data points overlaid (*n* = 3 independent biological replicates, each from a separate passage). Statistical significance was assessed using two-tailed unpaired *t* tests (df = 4). Exact *p*-values: D277D vs D277K: 0.156; D277K vs D277*: 0.204; WT vs W281P: 0.009; W281P vs W281*: 0.661; E300E vs E300K: 0.035; E300K vs E300*: 0.021; I315I vs I315R: 0.004; I315R vs I315*: 0.954; *$p < 0.05$ from a Student's two-tailed unpaired *t* test. No adjustments were made for multiple comparisons. Cell proliferation data for wild type and W281* are also presented in Supplementary Fig. 1b and Supplementary Fig. 5e.

achieved 27.4% ± 4.2% (mean ± SD) growth suppression, E300K achieved 49.8% ± 6.9%, and I315R achieved 6.4% ± 17.6%, while D277K showed 42.4% ± 9.4% growth suppression that did not reach statistical significance (Fig. 2i). Further, W281R, a variant seen in a patient with ovarian carcinosarcoma[13], which had a z-score of 2.06, also showed impaired antiproliferative function (15.34% ± 16.38%, Supplementary Fig. 5b–e). We then focused on understanding the mechanism by which missense mutants W281P and I315R, the highest-scoring variants at these two spatially adjacent residues (z-scores 5.57 and 3.83 respectively), led to impaired antiproliferative function.

## Missense mutations in *SMARCB1* disrupt SWI/SNF complex integrity

To evaluate the effects of the W281P and I315R mutations on SWI/SNF complex stability, we performed SMARCA4 (BRG1) co-immunoprecipitation (co-IP) followed by mass spectrometry using nuclear protein extracts after 48 h of induction (Fig. 3a; *Methods*; Supplementary Fig. 6a). SMARCA4 is a core, stable subunit of the SWI/SNF complex involved in chromatin remodeling[36]. Targeting SMARCA4 allows for reliable detection of complex-associated proteins, even when the SMARCB1 antibody may not efficiently capture the complex due to mutations or subunit instability. Quantification of the input revealed decreased SMARCB1 protein abundance in the missense mutant extracts compared to wild type (Supplementary Fig. 6b).

Both mutants showed reduced association (L2FC < −1.5, $p < 0.05$) of multiple SWI/SNF subunits, with 11 subunits showing decreased association in the W281P mutant and 12 in the I315R mutant (Fig. 3b, Supplementary Fig. 6c). Ten subunits showed reduced association in both missense mutants, including four (ARID1A, ARID1B, PBRM1, and SMARCC1) that are recurrently mutated in various cancers (Fig. 3c)[1]. Consistent with prior findings, we observed markedly reduced DPF2 association[37] (Supplementary Fig. 6d). Similar patterns of reduced SWI/SNF subunit association were observed when comparing respective nonsense mutants to wild type (Supplementary Fig. 6e, f).

The reduced associations primarily involved subunits from SMARCB1-containing subfamilies. Both mutants showed decreased association with subunits found in the cBAF complexes (DPF2, ARID1A/B) and PBAF complexes (PBRM1), while other PBAF associated subunits (BRD7, ARID2, PHF10) maintained association levels closer to wild type (Fig. 3d). Subunits specific to the SMARCB1-absent GBAF complex (BRD9, GLTSCR1) showed no significant change in association. These observations demonstrate that the W281P and I315R mutations alter the composition of SMARCB1-associated SWI/SNF complexes.

To orthogonally validate the mass spectrometry findings, we performed glycerol gradient sedimentation assays to assess SWI/SNF complex assembly in cells re-expressing wild type SMARCB1 or the W281P and I315R mutants (Fig. 3e, f, Supplementary Fig. 6g). Upon re-expression of wild type SMARCB1, SMARCA4 signal peaked in fractions 14–15. This pattern was altered by both mutants: W281P shifted the peak to fraction 13, while I315R distributed signal across fractions 13–14. Both mutants also increased relative SMARCA4 abundance in lower molecular weight fractions (6–12) compared to wild type (3% ± 1%), with W281P showing 19.3% ± 5.2% and I315R showing 12.3% ± 0.3%, consistent with the formation of smaller complexes. Additionally, ARID1A signal was reduced across fractions 13–16 in both mutants compared to wild type. Together, these changes demonstrate that W281P and I315R disrupt SWI/SNF complex integrity and provide independent validation of our mass spectrometry findings.

## Missense mutants disrupt attractive van der Waals forces in hydrophobic core of RPT2

To determine the molecular mechanism underlying the observed SWI/SNF complex destabilization, we performed twelve 200 ns molecular dynamics (MD) simulations on the AlphaFold-derived SMARCB1 structure containing the W281P and I315R mutations alongside wild type and a control population variant (S299L) (Fig. 2f, *Methods*). The RPT2 domain contains two alpha helices opposing two beta sheets, creating a hydrophobic barrel-like core[38]. Both mutations significantly increased unfavorable van der Waals (VDW) interaction energy between these structural elements (Fig. 4a). Relative to controls (WT and S299L), W281P showed a 4.9 ± 1.2 kcal/mol increase and I315R showed a 4.4 ± 0.57 kcal/mol increase in inter-regional energy (both $p < 0.01$, $n = 12$).

Contact frequency analysis revealed widespread loss of close contacts (<4 Å) between alpha-helical and beta-sheet regions in both mutants compared to wild type, with W281P and I315R showing distinct but overlapping patterns of contact disruption (Fig. 4b). We then quantified the energetic consequence of these structural changes through detailed VDW energy calculations. Five residue pairs showed significant energy increases: four pairs (281–290, 281–293, 281–315, 281–319) were disrupted by both mutations, while 279–293 was specifically affected by I315R (Fig. 4c). Notably, we observed one compensatory favorable interaction between residues 279 and 315 in I315R simulations. No significant changes were detected in electrostatic interactions.

Despite arising from different structural positions, both mutations converged on disruption of a shared hydrophobic network centered on residue 281 (Fig. 4d). Interactions between residue 281 and residues 290, 293, 315 and 319 emerged as critical stabilizing contacts, with their loss driving RPT2 destabilization. The cumulative VDW

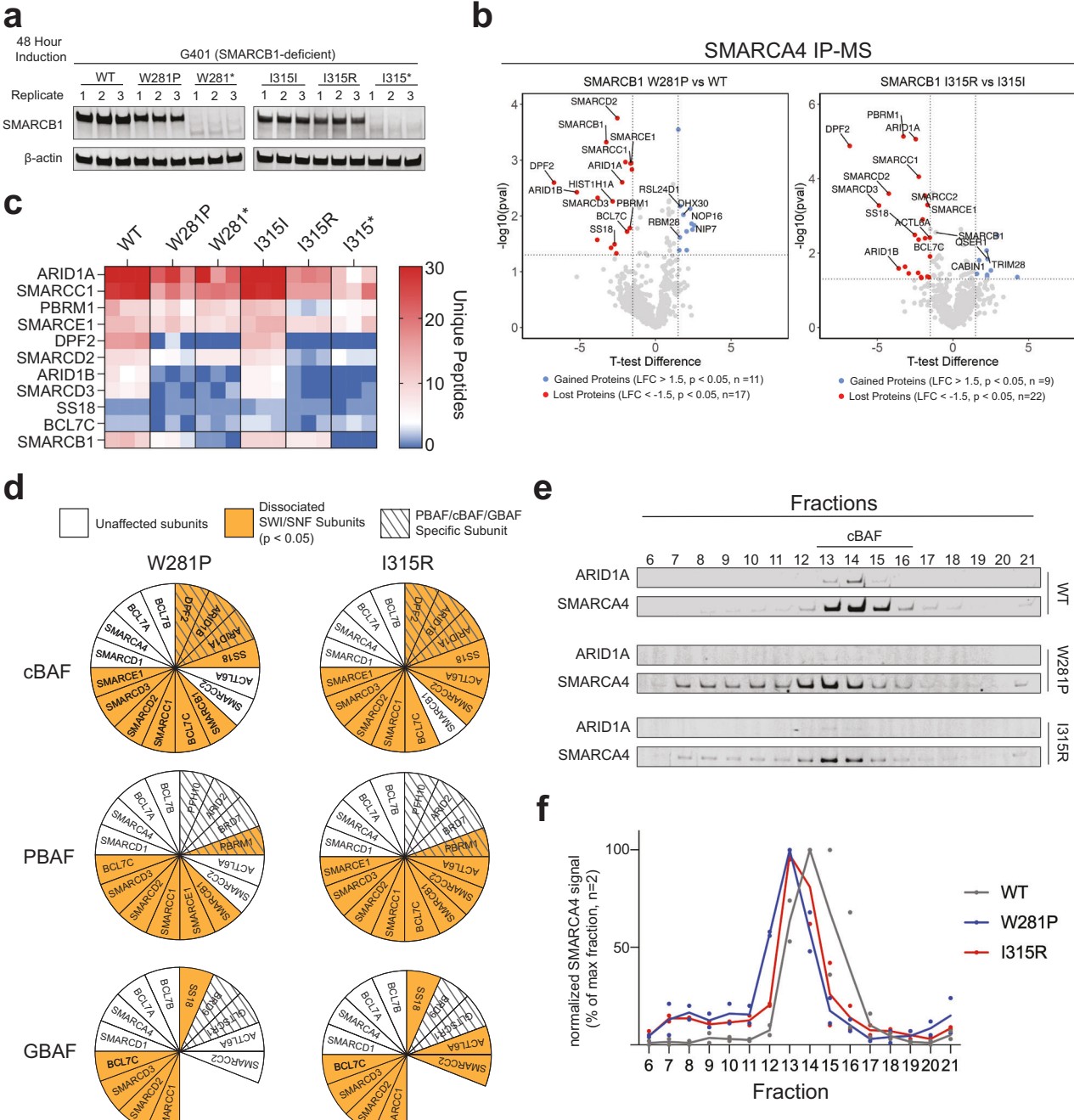

**Fig. 3 | Top DMS Hits Lead to SWI/SNF Complex Destabilization. a** Immunoblot from nuclear extracts from three biological replicates that were loaded as input into the SMARCA4-IP and submitted for mass spectrometry analysis. Data shown are representative of 2 independent experiments. **b** Volcano plot of proteins significantly depleted (red) and enriched (blue) in the mutant condition compared to the wild type. Proteins with |log2FC| > 1.5 and uncorrected $p < 0.05$ were considered significant (two-tailed two-sample $t$ tests, no adjustment for multiple comparisons). SWI/SNF subunits are labeled. **c** Unique peptide counts for overlapping SWI/SNF subunits depleted in both mutant conditions, shown across three biological replicates for all six conditions. **d** Schematic showing subfamily-specific SWI/SNF subunit depletion in missense mutants. Subfamily-specific (cBAF, PBAF, GBAF) subunits are indicated by hashed backgrounds, with those significantly depleted as observed through mass spectrometry highlighted in orange. Significance determined by |log2FC| > 1.5 and uncorrected $p < 0.05$ (two-tailed two-sample $t$ tests, no adjustment for multiple comparisons). **e** Nuclear extracts from G401 cells inducibly expressing WT or mutant (W281P, I315R) SMARCB1 for 48 h were separated by glycerol gradient sedimentation (10–30%). Fractions were collected and analyzed by immunoblotting for ARID1A and SMARCA4. Data shown are representative of 2 independent experiments (Supplementary Fig. 4e). **f** SMARCA4 signal intensity was measured across glycerol gradient fractions 6–21 from Fig. 4e and Supplementary Fig 4e, normalized to the maximum value for each replicate, and plotted for two biological replicates of G401 cells expressing WT or mutant SMARCB1 (W281P, I315R).

energy penalties were 4.50 kcal/mol for W281P and 3.54 kcal/mol for I315R, both exceeding the ~2 kcal/mol threshold commonly cited for functionally significant destabilization in structural studies[39–41], indicating severe structural compromise.

## SMARCB1 missense mutations selectively disrupt distal enhancer regulation

To further explore the downstream consequences of the observed SWI/SNF complex instability in these mutants, we performed ATAC-

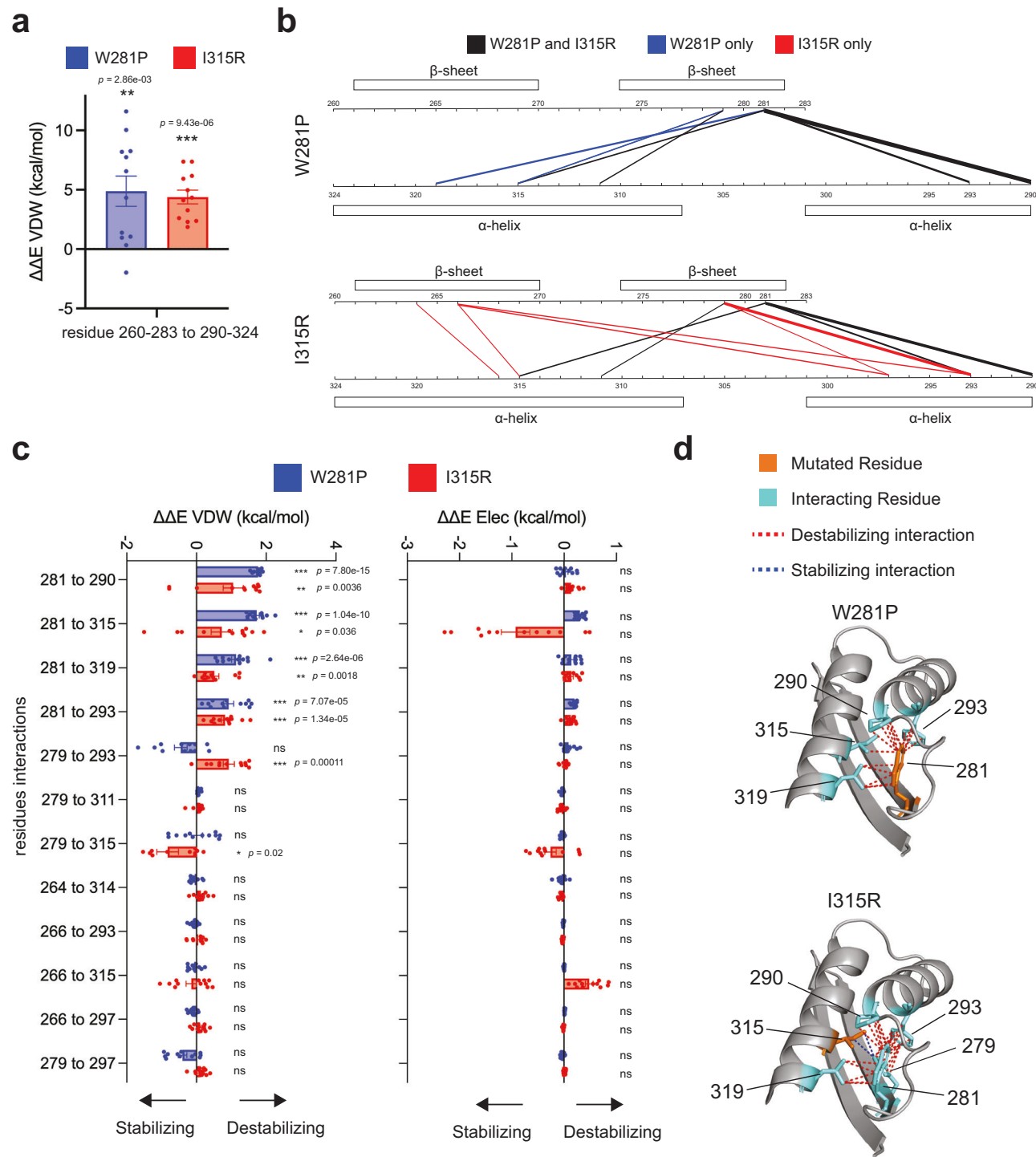

seq to evaluate global changes in chromatin remodeling. Prior studies have demonstrated that re-expressing SMARCB1 in SMARCB1-deficient cell lines leads to global increases in SWI/SNF occupancy and chromatin accessibility particularly at distal regulatory regions[2,3,42]. To identify robust accessibility changes associated with SMARCB1 disruption in our specific model, we established a consensus set of 92,028 accessibility peaks observed upon wild type SMARCB1 re-expression (*Methods*). Analysis of differential accessibility within this consensus set revealed 31,863 gained peaks (showing increased accessibility upon SMARCB1 re-expression), 231 lost peaks (showing decreased

accessibility upon SMARCB1 re-expression), and 59,934 persistent peaks (showing no significant differential accessibility upon SMARCB1 re-expression) (FDR < 0.05). Due to the small number of lost peaks, subsequent analyses focused on gained and persistent peaks. These results demonstrate that approximately one-third of accessible chromatin sites are SMARCB1-dependent, while the remaining maintain accessibility independent of SMARCB1 function. We then analyzed ATAC-seq signal intensity for wild type, missense, and nonsense mutants across these genomic regions. Chromatin accessibility was markedly reduced at gained peaks in the presence of W281P and I315R

**Fig. 4 | Molecular dynamics simulations reveal hydrophobic core disruption by *SMARCB1* missense variants. a** Van der Waals energy changes between RPT2 structural regions (residues 260–283 to 290–324) for W281P and I315R mutations relative to wild type. Both mutations significantly decreased favorable interactions (W281P: 4.9 ± 1.2 kcal/mol, I315R: 4.4 ± 0.57 kcal/mol). Data are presented as mean values ± SEM with individual data points overlaid ($n = 12$ independent molecular dynamics simulation trajectories initiated with different random seeds). Statistical significance assessed using two-tailed one-sample $t$ tests (df = 11) against a null hypothesis mean of zero. **$p < 0.01$, ***$p < 0.001$. No adjustments were made for multiple comparisons. **b** Contact frequency maps showing inter-residue interactions (<4 Å) between alpha-helical (α-helix) and beta-sheet (β-sheet) regions in the RPT2 domain. Top panel: W281P simulations; bottom panel: I315R simulations. Line thickness represents magnitude of contact loss, with black indicating contacts present in both W281P and I315R mutant simulations, blue showing W281P-only and red showing I315R-only contacts. Only interactions with >20% loss of contacts are presented. **c** Pairwise van der Waals (left) and electrostatic (right)

energy differences between specific residue pairs for W281P (blue) and I315R (red) relative to wild type/S299L control. Positive values indicate destabilizing energy increases. Significant results shown meet dual criteria of $p < 0.05$ and $|\Delta\Delta E| > 0.5$ kcal/mol for VDW interactions and $|\Delta\Delta E| > 1$ kcal/mol for electrostatic interaction. Data are presented as mean values ± SEM with individual data points overlaid ($n = 12$ independent molecular dynamics simulation trajectories initiated with different random seeds). Statistical significance assessed using two-tailed one-sample $t$ tests (df = 11) against a null hypothesis of zero. *$p < 0.05$, **$p < 0.01$, ***$p < 0.001$; ns, not significant. No adjustments were made for multiple comparisons. **d** Structural representation of the wild type RPT2 domain highlighting interactions less than 4.2 Å. Top panel shows residues with significant disruption in W281P simulations; bottom panel show residues with significant disruption in I315R simulations. Orange residues indicate the positions of mutated residues (281 for W281P, 315 for I315R) in the wild type structure, cyan residues show key interacting residues that lose or gain contacts upon mutation. Red dotted lines highlight destabilizing interactions; blue dotted lines show stabilizing interactions.

mutations, highlighting the impact of these missense mutations on the remodeling function of the SWI/SNF complex at SMARCB1-dependent regions (Fig. 5a).

Analysis of genomic annotation distributions revealed distinct spatial patterns between persistent and gained accessibility sites (Fig. 5b). Persistent sites were enriched at promoter-TSS regions (16.8%) relative to gained sites (2.0%), consistent with SMARCB1-independent chromatin accessibility being concentrated at core promoters. In contrast, SMARCB1-dependent gained sites were more frequently located in intronic (50.4%) and intergenic (44.4%) regions compared to persistent sites (41.7% and 33.3%, respectively). Although persistent sites were also largely intronic and intergenic, their relative enrichment at promoters indicate that a subset of essential transcriptional programs remain accessible independent of SMARCB1. Thus, SMARCB1 appears dispensable for core promoter accessibility but essential for establishing and maintaining distal regulatory regions.

Motif enrichment analysis revealed the sequence features underlying these genomic distributions and their differential sensitivity to SMARCB1 disruption (Fig. 5c). Persistent peaks were primarily enriched for CTCF binding motifs (20.6%), with AP-1 motifs present at lower frequency (10.1%). In contrast, gained peaks showed strong enrichment for AP-1 (29.5%) and TEAD2 (19.9%), with AP-1 sites being the most predominantly affected by SMARCB1 disruption, consistent with previous reports[2,29,43]. Analysis of motif density across all accessible peaks in each condition further supported these trends (Supplementary Fig. 7a). Although both persistent and gained peaks are largely intergenic or intronic, they are enriched for distinct distal motifs, with CTCF dominating persistent sites and AP-1 dominating gained sites. Together these data show that AP-1-enriched distal regulatory regions are preferentially lost upon SMARCB1 disruption, whereas CTCF-enriched structural elements are predominantly retained.

We then performed CUT&RUN of SMARCA4, SMARCB1, and SMARCE1 in wild type and mutant conditions to link accessibility and motif patterns with SWI/SNF occupancy. We defined a comprehensive set of 16,501 SWI/SNF binding sites by merging reproducible peaks for SMARCA4, SMARCB1 and SMARCE1 in wild type conditions (*Methods*). This union-based approach captures regions bound by one or more core subunits, accounting for the dynamic and transient nature of SWI/SNF binding to provide the most inclusive representation of chromatin remodeling activity. Intersection of these binding sites with our ATAC-seq peak groups yielded 12,124 sites overlapping with persistent accessible regions, 4357 sites overlapping with gained accessible regions, and 20 sites overlapping with lost accessible regions.

CUT&RUN analysis across these regions revealed global reductions in chromatin occupancy across both missense mutants (Fig. 5d), directly explaining the chromatin accessibility defects

observed in ATAC-seq. Across the SMARCB1 dependent peaks in W281P and I315R, all three SWI/SNF subunits showed decreased chromatin association similar to that of the respective nonsense mutants. For SMARCB1 independent peaks, SMARCE1 showed more severe chromatin occupancy loss in both missense mutants than either nonsense control, indicating that missense mutations can be more disruptive to subunit targeting than complete protein absence.

However, missense mutation specific defects were observed. W281P caused global reduction in chromatin occupancy across both persistent and gained accessible regions for all three subunits, suggesting broad disruption of SWI/SNF complex targeting. In contrast, I315R showed more selective effects. Specifically we saw relatively preserved occupancy at persistent regions but marked reduction at gained regions for SMARCA4 and SMARCB1. This differential pattern indicates that W281P more severely compromises overall complex targeting, while I315R preferentially disrupts binding at SMARCB1-dependent regulatory sites, with both mutations exerting particularly severe effects on SMARCE1 localization. Analysis of genomic annotation distributions revealed similar patterns as between persistent and gained accessible regions (Fig. 5e). Persistent SWI/SNF binding sites were enriched for promoter-TSS regions (17.9%) relative to gained sites (2.8%), while gained sites were predominantly intronic (62.9% vs. 50.9% for persistent sites) and intergenic (30.6% vs. 22.0% for persistent sites).

To evaluate whether these defects in chromatin accessibility impact transcriptional regulation, we performed bulk RNAseq for each condition. Both W281P and I315R mutants exhibited *SMARCB1* transcript levels comparable to wild type (Supplementary Fig. 7b), indicating that the reduced SMARCB1 protein levels observed in nuclear extracts (Supplementary Fig. 6b) are driven by post-transcriptional effects. Principal component analysis (PCA) revealed clear transcriptional separation, with both missense mutants clustering closely to their corresponding nonsense controls (Fig. 5f). Re-expression of wild type SMARCB1 resulted in 619 significantly differentially expressed genes (DEGs) compared to W281*, whereas the W281P missense mutant yielded only 77; similarly, I315I produced 499 DEGs relative to I315*, while I315R showed only 94 (Fig. 5g, Supplementary Fig. 7c). Genes normally upregulated upon wild type SMARCB1 re-expression were consistently attenuated in both mutants (Supplementary Fig. 7d). To determine whether these transcriptional effects were associated with chromatin changes, we next compared differential accessibility and expression. The WT/W281* and I315I/I315* comparisons showed 96.5% (3991/4138) and 96.1% (3026/3148) concordance, respectively, confirming that chromatin accessibility changes translate to functional transcriptional consequences (Fig. 5h).

In summary, these data demonstrate that the W281P and I315R missense mutations severely impair SMARCB1 antiproliferative

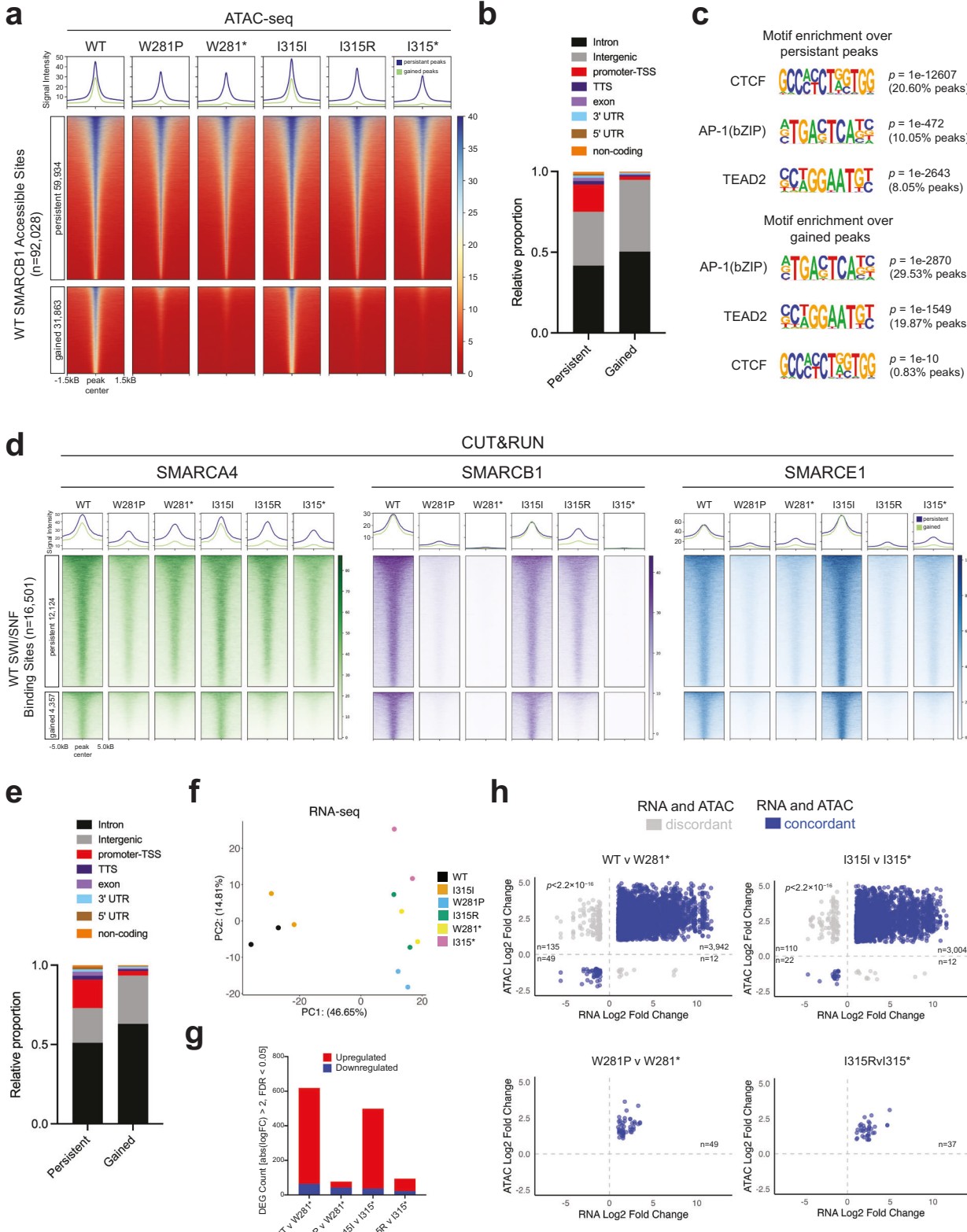

function by disrupting SWI/SNF complex integrity, chromatin remodeling activity, and transcriptional regulation. Molecular dynamics simulations revealed that both mutations destabilize the RPT2 domain through disruption of hydrophobic interactions, providing a structural mechanism for the observed functional defects and demonstrating how specific missense variants can be functionally equivalent to loss-of-function mutations (Fig. 6).

## Discussion

This study employs deep mutational scanning (DMS) to systematically characterize the functional impact of *SMARCB1* variants in the context of antiproliferative activity. From 8418 *SMARCB1* alterations profiled across three patient-derived cell lines, we identified 101 candidate deleterious missense variants and revealed 13 highly mutation-intolerant residues distributed across three domains: the WHD, IDR,

**Fig. 5 | Missense mutations disrupt chromatin remodeling and transcriptional regulation. a** ATACseq signal across all mutant conditions at both persistent and significantly gained accessible peaks when comparing wild type to nonsense mutants. Significantly lost accessible peaks are not shown. **b** Annotated genomic regions for both persistent and gained peaks. **c** Motif enrichment analysis for AP-1(bZIP), TEAD2, and CTCF binding motifs for persistent and gained regions. Statistical testing was performed using HOMERv4.11.1 with a binomial test (one-sided) against a dinucleotide-shuffled background, with Benjamini-Hochberg correction for multiple comparisons. Uncorrected *p*-values are displayed in the figure; all Benjamini-Hochberg corrected *p*-values were below machine precision. **d** CUT&RUN signal for SMARCA4, SMARCB1, and SMARCE1 binding at regions that are either persistently accessible or gained in ATAC-seq analysis (Fig. 5a) and bound by SWI/SNF (defined as the union of SMARCA4, SMARCB1, and SMARCE1 peaks). Normalized signal intensity is displayed ±5 kb from peak centers in WT and mutant lines following re-expression, with average profiles above. **e** Annotated genomic

regions for persistent or gained SWI/SNF binding sites. **f** Principal component analysis of bulk RNA sequencing for all constructs. **g** Number of differentially expressed genes across each comparison based on bulk RNA sequencing. **h** Scatterplot showing the log2 fold change (L2FC) for each significantly differentially accessible peak and its associated annotated gene for each comparison. The data includes only peaks and genes with an adjusted *p*-value (*p*adj) <0.05 and an absolute L2FC > 1. Concordance between chromatin accessibility and gene expression changes was assessed independently for each comparison using two-sided Fisher's exact tests. Two comparisons (W281P v W281* and I315R v I315*) showed complete concordance between chromatin accessibility and gene expression directionality (100% of peaks in concordant quadrants), precluding formal statistical testing. For the remaining two comparisons, Fisher's exact tests confirmed significant concordance (Comparison WT v W281*: OR = 80.38, 95% CI: 59.81–110.41; Comparison I315I v I315*: OR = 136.38, 95% CI: 88.56–220.25; $p < 2.2 \times 10^{-16}$ for both). No adjustments were made for multiple comparisons.

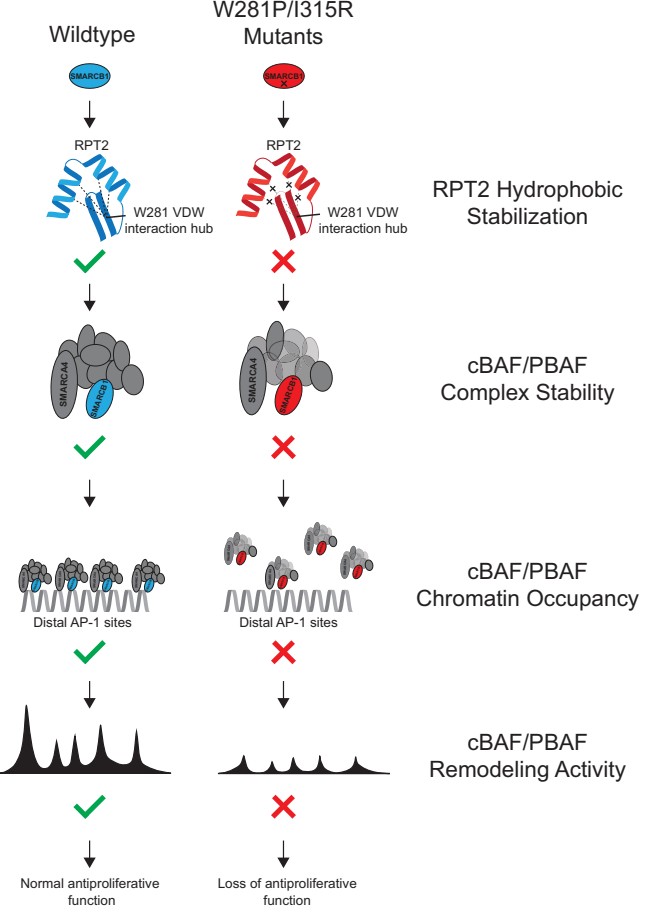

**Fig. 6 | Model.** Proposed model for how missense mutations in the RPT2 domain disrupt SMARCB1 function.

and RPT2 domain. Notably, several missense mutations in the RPT2 domain, particularly W281P and I315R, exhibited significant antiproliferative defects despite retained protein expression. Comprehensive functional validation demonstrated that these mutations disrupt SWI/SNF complex stability, localization, remodeling activity, and alter transcriptional regulation. Molecular dynamics simulations revealed that these mutations disrupt stabilizing intramolecular van der Waals interactions, providing a structural rationale for the observed complex destabilization. These findings demonstrate that missense mutations can severely compromise SMARCB1 antiproliferative function through mechanisms beyond simple protein loss, highlighting potential limitations of IHC-based diagnostic approaches that rely solely on protein expression status.

To elucidate the mechanistic basis for these functional defects, we employed complementary biochemical and chromatin profiling approaches. Chromatin binding assays revealed substantial loss of SWI/SNF subunit occupancy at regions with altered accessibility in both mutants, with SMARCE1 showing more occupancy loss in missense mutants than nonsense controls. These data, combined with biochemical analysis, reveal a hierarchy of SWI/SNF complex vulnerability that extends beyond the sequential assembly model previously described[44]. While cBAF complexes showed complete subfamily-specific disruption (loss of DPF2, ARID1A/B), PBAF complexes were only partially affected, with PBRM1 dissociating while other subfamily-specific subunits (BRD7, ARID2, PHF10) remained unaffected. Subunits specific to GBAF complexes were entirely unaffected, consistent with these complexes naturally lacking SMARCB1. Importantly, we observed destabilization of shared initial BAF core subunits (SMARCC1, SMARCD2, SMARCD3), suggesting these W281P and I315R mutants create post-assembly structural instability that propagates differentially across complex subtypes.

This differential complex disruption provides a framework for understanding the observed chromatin accessibility patterns. The preferential vulnerability of cBAF complexes, which typically occupy enhancers and distal regulatory elements[45], likely underlies the accessibility loss at AP-1 enriched distal regulatory sites. The retention of key PBAF-specific subunits, combined with PBAF's preferential binding to promoters[46], might initially suggest that residual PBAF function maintains promoter accessibility. However, the near-complete loss of SMARCE1 occupancy and minimal promoter response to SMARCB1 re-expression indicate that promoter regions are inherently less dependent on SWI/SNF activity than enhancer elements. The parallel between biochemical complex disruption and loss of chromatin occupancy provides strong evidence that the two presented *SMARCB1* missense mutations drive functional defects through destabilization rather than reduced activity of assembled complexes.

At the chromatin level, this creates a hierarchy of SMARCB1 dependence, where AP-1-dependent distal regulatory regions show strong sensitivity to SMARCB1 disruption, while CTCF-associated structural elements and promoters maintain accessibility independent of SMARCB1 function. This selective disruption pattern suggests that the antiproliferative function of SMARCB1 may depend primarily on cBAF-mediated enhancer accessibility and distal regulatory programs rather than PBAF-mediated promoter function and GBAF-mediated structural chromatin organization. This mechanistic framework demonstrates that SMARCB1's antiproliferative function operates through selective disruption of enhancer-driven gene regulation rather than global chromatin remodeling defects.

Our findings across other domains of SMARCB1 reveal additional structure-function relationships that warrant further investigation. The WHD's mutational intolerance may reflect its distinct conformations in cBAF versus PBAF complexes observed in cryo-EM structures,

potentially explaining complex-specific functional requirements. Within this domain, our identification of R52 as highly mutation-intolerant aligns with structural predictions of its insertion into the DNA major groove in PBAF structures, suggesting a direct role in DNA binding that could explain its functional importance. The cluster of mutation-intolerant residues within the IDR (E122, Q123, A125) suggests this region may facilitate conformational flexibility required for dynamic protein-protein interactions. These structure-based hypotheses provide promising directions for future experimental validation.

Our DMS data revealed that N-terminal nonsense mutations retain partial to full antiproliferative function. While the precise mechanism underlying this observation remains to be established, we note the presence of 5 methionine residues in the N-terminal region which could serve as alternative start sites. However, alternative start site usage alone is unlikely to fully account for this observation, and additional mechanisms are likely.

While our primary focus was identifying loss-of-function variants, our scoring framework is symmetric and enables interpretation of variants at both functional extremes. More negative z-scores indicate stronger depletion upon SMARCB1 re-expression, consistent with greater antiproliferative activity. Variants with z-scores less than -2 may exhibit enhanced activity through improved protein stability, SWI/SNF complex incorporation, or chromatin interactions. These represent functionally enhanced variants rather than oncogenic gain-of-function mutations. However, strong depletion signals driving highly negative z-scores can be more sensitive to experimental noise in pooled screens, whereas enrichment signals are more robustly detectable[47]. This underscores the potential of using saturation mutagenesis to uncover functionally enhancing variants.

While our proliferation-based assay provides insights into SMARCB1's antiproliferative function, we acknowledge this readout may not fully capture pathogenicity in developmental contexts such as Coffin-Siris syndrome, where *SMARCB1* mutations primarily affect developmental processes rather than proliferation. This limitation underscores the importance of selecting functional readouts appropriate to the specific disease mechanism and biological context of interest. Our DMS approach also has inherent technical limitations that warrant consideration. The pooled screening strategy, while cost effective and designed to minimize batch effects by having all variants compete within the same cellular population, introduces potential noise from lentiviral template switching during reverse transcription. Template switching rates can be substantial over long genomic distances, potentially creating cells with multiple mutations rather than the intended single variants. However, our reproducible results, particularly at the beginning time points, across independent replicates suggest this technical noise did not prevent detection of meaningful functional differences between variants. A separate technical constraint arises from current short-read sequencing technologies, which limit our ability to directly identify and exclude molecules containing multiple variants that span long distances. Advances in long-read sequencing technologies could improve the efficiency of future DMS screens by enabling detection and removal of recombinant molecules, thereby reducing noise and improving signal clarity. While our analysis pipeline normalized for coverage differences and Nextera fragmentation bias, these technical factors may still contribute to variability in variant detection. These considerations highlight the importance of replication and careful controls in large-scale functional screens with downstream validation.

Taken together, these findings have important clinical implications for SMARCB1-deficient cancers. Our comprehensive functional map of *SMARCB1* variants reveals that IHC maybe insufficient for diagnosing SMARCB1-deficient cancers, as functionally deleterious missense mutations can occur without complete protein loss. Complementary DNA-based approaches such as targeted panels or exome sequencing would capture functionally deleterious variants regardless of protein stability, improving diagnostic accuracy, guiding patient enrollment in SWI/SNF-targeted therapies, and informing germline variant counseling. Furthermore, these findings challenge the assumption that mutation frequency alone predicts pathogenicity and highlight the need for functional assays in assessing variant effects.

## Methods

### ORFeome overexpression screen

**Library and viral production.** The ORFeome library consisted of 16,172 human open reading frames cloned into lentiviral expression vectors as previously described[48]. Lentiviral particles were produced at the Broad Institute Genetic Perturbation Platform (GPP) following standard protocols (https://portals.broadinstitute.org/gpp/public/resources/protocols).

**Transduction and MOI optimization.** Viral supernatant was titrated in G401 cells using 50–200 μL of virus per 3 million cells to achieve a multiplicity of infection (MOI) of 0.2–0.6. To ensure low MOI and minimize cells receiving multiple viral integrations, a separate aliquot of transduced cells was plated and puromycin-resistant colonies were counted to confirm MOI remained within the target range. For the screen, G401 cells were transduced at the optimized viral concentration to maintain a representation of >500 cells per ORF throughout the experiment.

**Screen timeline and sample collection.** Following overnight transduction, cells underwent puromycin selection for 48 h. Cells were then harvested at two timepoints: (1) the early timepoint ($t = 0$) at 72 h post-transduction (24 h transduction + 48 h selection), representing the initial distribution of integrated viral constructs; and (2) the end timepoint ($t = $ end) at 7 days post-transduction (-10 days total culture time), after cells were passaged every 3–4 days upon reaching 80% confluence. This endpoint allowed sufficient time for proliferative effects to manifest.

**DNA extraction and sequencing.** Genomic DNA was extracted from cell pellets using the QIAamp DNA Blood Maxi Kit (QIAGEN) according to GPP standards. Integrated ORF barcodes were amplified by PCR, and sequencing libraries were prepared and sequenced on Illumina platforms at the GPP. FASTQ files were demultiplexed and processed using PoolQ v2 software. Barcode counts were normalized and converted to log2 fold changes ($t = $ end/$t = 0$) for downstream analysis.

### Cell lines and genetic characterization

G401 cells were obtained from ATCC (ATCC-CRL1441). BT16 cells were from Charles David James, Northwestern University. CCLF_PEDS9001_T1 was previously published in our group[49]. Cell lines used in this study were then genetically characterized for known mutations and genomic alterations and tested for mycoplasma. Mutation data for BT16 and G401 cell lines were retrieved from the Cancer Dependency Map (DepMap) Public portal (https://depmap.org/portal/, accessed October 9, 2025). Genomic alterations for patient-derived cell line CCLF_PEDS9001_T1 were obtained from our previous study[49]. A complete list of mutations is provided in Source Data.

### Deep mutational scanning

**DMS library generation.** SMARCB1 variant library was produced following the principle of the Mutagenesis by Integrated TilEs (MITE) method[50]. The SMARCB1 open reading frame is partitioned into 90-base tiles. DNA Oligos representing all mutations in the space of a tile were synthesized. Each 90 base tile is flanked by 30 base sequences that are complementary to the reference ORF. To avoid interference during tile amplification, oligos for adjacent tiles were synthesized on separate chips. Oligo tile pools were amplified by emulsion PCR using

primers to the 30 base flanking regions and purified on a 2% agarose gel. Entry vector cloning is performed on a per tile basis by linearizing the entry vector at the appropriate tile position using Phusion polymerase (New England Biolabs) and primers to the regions flanking the tiles. The linearized vectors were purified on 1% agarose gel and DpnI treated. The DpnI-treated linear plasmid backbones were then mixed with the relevant PCR amplified tile and assembled by in vitro recombination with the NEBuilder HiFi DNA Assembly Master Mix (New England BioLabs). The assembly reactions were purified, electroporated into TG1 *E. coli* cells (Lucigen), and recovered for one hour at 37 °C in Recovery Media (Lucigen). Aliquots from the transformations were used to inoculate overnight cultures of LB containing 25 μg/mL of Zeocin (ThermoFisher). Cells were harvested by centrifugation and plasmid DNA was isolated using the QIAGEN Midiprep Plus kit (QIAGEN). Plasmid pools, each corresponding to a tile, were verified by sequencing Nextera XT (Illumina). A pool of all pUC57-tile libraries was made with equal weight per variant. This pool was subjected to restriction digest by NheI and BamHI. After purification by 1% agarose gel, the excised linear fragment library is ready for cloning ligation with an expression vector that was pre-processed with NheI and BamHI. The ligated construct was used to transform Stbl4 bacterial cells and plasmid DNA (pDNA) was extracted using QIAGEN Maxi Prep Kits. The resulting pooled pDNA ORF variant library was sequenced via Illumina Nextera XT platform to determine the distribution of variants within the library.

**Lentiviral transduction.** 293 T viral packaging cells were transfected using TransIT-LT1 transfection reagent (Mirus Bio) with three plasmids: the pooled SMARCB1 pDNA library, a packaging plasmid containing gag, pol and rev genes (psPAX2, Addgene), and an envelope plasmid containing VSV-G (pMD2.G, Addgene). Media was changed 6–8 h after transfection and virus was harvested 24 h thereafter. In transduction, virus was placed on cells dropwise with polybrene and spun down at 2000 rpm for 30 min at 30 °C. Antibiotic selection was performed the following day.

**Transduction and MOI validation.** Viral supernatant was titrated in each cell line (G401, BT16, and CCLF-PEDS9001_T1) using 50–200 μL of virus per 3 million cells to achieve an MOI of 0.2–0.6. Low MOI is critical to ensure that the majority of transduced cells receive only a single viral integration. To validate MOI, a separate aliquot of transduced cells was plated at limiting dilution and puromycin-resistant colonies were quantified to confirm MOI remained within the target range. For the screen, cells were transduced at the optimized viral concentration and centrifuged at 2000 rpm for 30 min at 30 °C to enhance viral transduction. A representation of >700 cells per variant was maintained throughout the experiment (Source Data).

**Screen timeline and sample collection.** Following overnight transduction, cells were selected with puromycin for 48 h (G401) or 72 h (BT16 and CCLF-PEDS9001_T1). Cells were harvested at two timepoints: (1) the early timepoint ($t = 0$) immediately after selection (72 h for G401; 96 h for BT16 and CCLF-PEDS9001_T1), representing the initial distribution of integrated variants as indicated in Supplementary Fig. 2a (start of screen); and (2) the end timepoint ($t$ = end) at 8–14 days post-transduction, depending on cell line growth rates. Cells were passaged every 3–4 days upon reaching 80% confluence to maintain log-phase growth. The extended culture period allowed for increased dynamic range in detecting proliferative effects of loss-of-function variants. Two independent biological replicates were performed for each cell line.

**ORF extraction from gDNA by PCR.** PCR reactions were set up in 96-well plates using two primers: Forward: 5′-ATTCTCCTTG-GAATTTGCCCTT-3′; Reverse: 5′-CATAGCGTAAAAGGAGCAACA-3′. and

Q5 DNA polymerase (New England Biolabs). All PCR reactions for each gDNA sample were pooled, concentrated with a PCR cleanup kit (QIAGEN), and separated by gel electrophoresis. Bands of the expected size were excised and DNA was purified using a QIAquick kit (QIAGEN) followed by an AMPure XP kit (Beckman Coulter).

**Nextera sequencing.** Nextera reactions were performed according to the Illumina Nextera XT protocol. For each sample, we set up 6 Nextera reactions, each with 1 ng of purified ORF DNA. Reactions for each screen sample were indexed with a unique i7/i5 index pair. After the limited-cycle PCR step, the Nextera reaction products were purified with AMPure XP kit. All samples were then pooled and sequenced using an Illumina Nextseq flowcell.

**Processing next-generation sequencing data.** The NGS reads were processed with a second-generation variant calling software called AnalyzeSaturationMutagenesis (ASMv1.0), as part of GATK, downloadable at https://github.com/broadinstitute/gatk/releases[25].

### Functional Z-score calculation
**Baseline calculation using silent mutants.** For each silent mutation, the rolling average and standard deviation of L2FC values were computed within a window of 5 codons (±2 codons flanking the silent mutation). This rolling window approach was implemented to reduce noise from local codon context and sequence-specific effects while ensuring that baseline estimates were still derived from neighboring codons for residues lacking silent mutations. The rolling average represents the expected abundance change for wild type codons, and the rolling standard deviation reflects the variability of L2FC values for silent codons across the entire DMS library.

The rolling statistics were computed separately for each replicate (G401, BT16, CCLF_PEDS9001T) to capture replicate-specific effects on wild type abundance. For each codon, the rolling average and standard deviation were calculated as follows:
1. For each codon, we selected all silent codons within the window of ±2 positions.
2. The mean and standard deviation of L2FC values for these silent codons were calculated for each replicate.

**Z-score calculation for variants.** The z-score for each variant was calculated by comparing its L2FC value to the rolling average and standard deviation derived from the silent mutants. For each mutation of interest, the rolling silent average for the corresponding codon was subtracted from the variant L2FC value, and the result was divided by the rolling silent standard deviation. This provides a z-score that reflects how much the variant deviates from the expected abundance changes of silent mutations. Mathematically, the z-score for a given variant was calculated as:

$$Z_{score} = \frac{L2FC_{variant} - Rolling\ Average_{silents}}{Rolling\ SD_{silents}} \quad (1)$$

This calculation was performed separately for each replicate, allowing the assessment of functional impact across different experimental conditions.

**Z-scores across replicates.** For each variant, z-scores were computed for all experimental replicates. The final functional z-score for each variant was calculated by averaging the z-scores across all replicates. The standard deviation of z-scores across replicates was also computed to assess the consistency of variant effects across experiments.

### Pathogenicity prediction score thresholds
We applied established thresholds from three computational pathogenicity prediction tools to classify SMARCB1 missense variants:

REVEL: Variants with scores ≥0.75 were classified as pathogenic. This stringent threshold captures 52.1% of known disease mutations while maintaining low false positive rates (3.3% of neutral variants and 4.1% of evolutionary structural variants misclassified as pathogenic), as reported by the REVEL developers[24].

CADD: We utilized scaled PHRED scores with a threshold of ≥20 to classify variants as potentially pathogenic, identifying variants in the top 1% of all possible reference genome single nucleotide variants in terms of predicted deleteriousness[25]. While the CADD developers note that optimal thresholds depend on analysis-specific factors and recommend ranking variants rather than applying arbitrary cutoffs, we selected the PHRED score ≥20 threshold as it represents a stringent cutoff and binary classification remains standard practice in medical genetics research.

AlphaMissense: Variants with scores ≥0.564 were classified as likely pathogenic using the standard threshold established by the AlphaMissense developers[23]. This calibrated cutoff achieves 90% expected precision for both pathogenic and benign classifications when validated against ClinVar data. The AlphaMissense score can be interpreted as the approximate probability of a variant being clinically pathogenic.

### Nuclear protein extraction

Cells were trypsinized and counted using trypan blue exclusion. 25 million cells were spun down at 1000 rpm for 5 min at 4 °C. Cells were then washed with cold PBS and spun down at 1000 rpm for 5 min at 4 °C. Cell pellets were then resuspended in 5 mL of EB0 hypotonic buffer (50 mM Tris-HCl, 1 mM EDTA, 1 mM $MgCl_2$, 0.1% NP-40 and supplemented with 1 mM PMSF and protease inhibitor), and incubated on ice for 5 min. Lysates were then spun down at 4750 rpm for 5 min at 4 °C. Supernatants were aspirated and remaining pellet was resuspended in 700 μL of EB300 (50 mM Tris-HCl, 300 mM NaCl, 1% NP-40, 1 mM EDTA, 1 mM $MgCl_2$, and supplemented with 1 mM PMSF and protease inhibitor). Lysates were incubated on ice and occasionally vortexed for 10 min. Lysates were pelleted at 21,000 rpm for 10 min and supernatant was collected. Protein concentrations were calculated using a bicinchoninic acid (BCA) assay and frozen at −80 °C.

### Lentiviral transduction for inducible cell lines

Lentiviral particles were generated using 293T cells co-transfected with the gene deliver vector and packaging vectors Δ8.9 and VsVg. More specifically 293T cells were plated in 6 cM plates at a density of 1 million cells per plate in penicillin/streptavidin free DMEM. The following day the 293T cells were transfected. Media was changed with high serum DMEM (30% FBS) and harvested the following 2 days. The combined viral supernatant was then filtered with 0.45 μm filters. 500 μL of virus was then applied dropwise to target cells in the presence of polybrene and spun down at 2000 rpm for 30 min at 30 °C. Antibiotic selection was performed the following day.

### Cell proliferation assays

G401 cells were plated in a 6 well plate at a density of 25,000 cells per well and induced with 1 μg/mL doxycycline. After 96 h, cells were trypsinized and counted using trypan blue exclusion. Cells were replated again at a density of 25,000 cells per well and induced again at 1 μg/mL doxycycline. After another 96 h, cells were trypsinized and counted using trypan blue exclusion. All counts were completed in 2 technical replicates and in biological triplicates for each condition. Cells were then stained using crystal violet at day 10.

### Immunoblots

For experiments using total protein lysate, cells were lysed using 1x RIPA (Cell Signaling Technologies, 9806) with protease inhibitors (coMplete, Roche, 42484600) and phosphatase inhibitors (PhosSTOP,

Roche, 04906837001). 4–12% SDS-PAGE gels (SMOBIO Technology, Inc) were transferred onto PVDF (Millipore, IPFL00010) or nitrocellulose membranes (ThermoFisher Scientific, IB23001). Blots were then visualized using Odyssey Classic (LICORbio, Lincoln, NE). Immunoblot was performed using: SMARCB1 (Santa Cruz−sc-166165), β-actin (CST −8457S), DPF2 (Cell signaling−71642S), SMARCC2 (Cell signaling− 12760S), SMARCA4 (Santa Cruz- sc-17796). Secondary antibodies: Alexa Fluor® IRDye 680RD goat anti-rabbit (ThermoFisher) or IRDye 800CW goat anti-mouse (LI-COR, Lincoln, NE); with imaging on an Odyssey® scanner (LI-COR). Immunoblot source data are provided in Supplementary Information.

### SWI/SNF Co-immunoprecipitation for MS/IP

To perform BAF complex co-immunoprecipitations, BRG1 (G7) antibody was cross-linked with dynabeads (Pierce Protein G Magnetic Beads, Thermo Scientific). Briefly, 5 μg of antibody was incubated on a rotator with 50 μL of dynabeads for 1 h at room temperature (RT). Beads were then washed twice with 0.2 M NaBorate (pH 9.02) and crosslinked using 6 mg/mL Dess−Martin periodinane (DMP) in 0.2 M Naborate (pH 9.02) and incubated on a rotator for 45 min at RT. Beads were then incubated on a rotator for 1 h in 0.2 M ethanolamine (pH 8.2). Beads were then washed twice with 20 mM Tris-HCl, twice with 100 mM glycine (pH 2.5), and twice with 100 mM Tris-HCl. Beads were then resuspended in IP buffer.

For G401 nuclear extract immunoprecipitations, 800 μg of nuclear protein was incubated with crosslinked BRG1-dynabead mixture overnight at 4 °C. Beads were then washed 3 times with IP buffer, and 3 times with PBS (1:100 PMSF) and eluted in 50 μL of sample buffer (1x NuPAGE LDS Buffer and 100 mM DTT). To ensure equivalent loading, 10% of the IP was eluted and analyzed by gel electrophoresis. Silver staining was performed using Pierce™ Silver Stain Kit (Lot YC367461). Complex-bound beads were then frozen at −20 °C.

### On-bead/in-solution digestion

For protein digestion, 3 biological replicates per genotype were processed at the same time using a published protocol was followed for each condition with one negative control (n = 10 for each mutant)[51]. Digestion buffer (50 mM NH4HCO3) was added to the complex-bound beads, and the mixture was then treated with 1 mM dithiothreitol (DTT) at RT for 30 min, followed by 5 mM iodoacetimide (IAA) at RT for 30 min in the dark. Proteins were digested with 2 μg of lysyl endopeptidase (Wako) at RT for overnight and were further digested overnight with 2 μg trypsin (Promega) at RT. The resulting peptides were desalted with HLB column (Waters) and were dried under vacuum.

### LC-MS/MS

The data acquisition by LC-MS/MS was adapted from a published procedure[52]. Derived peptides were resuspended in the loading buffer (0.1% trifluoroacetic acid, TFA) and were separated on a Water's Charged Surface Hybrid (CSH) column (150 μm internal diameter (ID) × 15 cm; particle size: 1.7 μm). The samples were run on an EVOSEP liquid chromatography system using the 30 samples per day preset gradient (44 min) and were monitored on a Orbitrap Fusion Lumos Mass Spectrometer (ThermoFisher Scientific, San Jose, CA). The mass was operated in data dependent mode in top speed mode with a cycle time of 3 s. Survey scans were collected in the Orbitrap with a 60,000 resolution, 400–1600 m/z range, 400,000 automatic gain control (AGC), 118 ms max injection time and rf lens at 30%. Higher energy collision dissociation (HCD) tandem mass spectra were collected in the Orbitrap with a 30,000 resolution, collision energy of 30%, an isolation width of 1.6 m/z, AGC target of 50,000, and a max injection time of 54 ms. Dynamic exclusion was set to 30 s with a 10 ppm mass tolerance window.

## MaxQuant

Label-free quantification analysis was adapted from a published procedure[52]. Spectra were searched using the search engine Andromeda, integrated into MaxQuant, against 2022 human UniProtKB/Swiss-Prot database (20,387 target sequences). Methionine oxidation (+15.9949 Da), asparagine and glutamine deamidation (+0.9840 Da), and protein N-terminal acetylation (+42.0106 Da) were variable modifications (up to 5 allowed per peptide); cysteine was assigned as a fixed carbamidomethyl modification (+57.0215 Da). Only fully tryptic peptides were considered with up to 2 missed cleavages in the database search. A precursor mass tolerance of ±20 ppm was applied prior to mass accuracy calibration and ±4.5 ppm after internal MaxQuant calibration. Other search settings included a maximum peptide mass of 6000 Da, a minimum peptide length of 6 residues, 0.05 Da tolerance for orbitrap and 0.6 Da tolerance for ion trap MS/MS scans. The false discovery rate (FDR) for peptide spectral matches, proteins, and site decoy fraction were all set to 1 percent. Quantification settings were as follows: re-quantify with a second peak finding attempt after protein identification has completed; match MS1 peaks between runs; a 0.7 min retention time match window was used after an alignment function was found with a 20-min RT search space. Quantitation of proteins was performed using summed peptide intensities given by MaxQuant. The quantitation method only considered razor plus unique peptides for protein level quantitation.

## Mass spectrometry data analysis

Statistical analysis was performed with Perseus v2.0.11.0 using LFQ intensity data from single shot mass spectrometry experiments. LFQ data was log2 transformed and proteins with missing values in greater than three samples were removed. Missing values were then imputed based on the assumption of a normal distribution with standard deviation of 0.3 and down shift of 1.8. This down shift was applied to ensure imputed values were biased towards lower intensities. Statistical analysis was then performed using two-tailed two-sample $t$ tests comparing each mutant. Proteins with $t$ test difference (log2FoldChange) > |1.5| and $p < 0.05$ (no correction for multiple comparisons) were considered significant.

## Glycerol gradient sedimentation assays

Nuclear extract (600 µg, quantified by BCA) suspended in EB300 was overlaid onto an 11 mL 10-30% glycerol (in HEMG buffer) gradient prepared in a 14 ×89 mm polyallomer centrifuge tube (Beckman Coulter, No. 331327). Tubes were ultracentrifuged in a SW40 rotor at 4 °C for 16 h at 40,000 r.p.m. Fractions (550 µl) were collected and used in immunoblot analysis.

## RNA sequencing

RNA was extracted from G401 cells following an 8-day induction at 1 µg/mL doxycycline for each condition. For mechanistic studies, RNA was collected in biological replicates Libraries were prepared using Illumina TruSeq. Samples were run with at least 20 million paired-end reads using Novoseq 6000 (Illumina). Fastq or BAM files were mapped and aligned using Illumina Dragen v3.7.5 to GrCh38 on Amazon Web Services. Samples were then quantified using salmon through Illumina Dragen v3.7.5. Gene count files were converted to a counts matrix using tximport. The counts matrix was used as input into DESeq2 to evaluate differential gene expression. Normalized read counts matrices were used as input into PCA to visualize clustering between samples. Versions used: R v4.4.2; R Studio 2022.07.2 Build 576; tximport v1.26.1, DESeq2 v1.46.0, ggplot v3.5.1.

## ATAC sequencing

ATAC-seq libraries were made using 50,000 freshly harvested G401 cells with viability >95%. Tagmentation was performed by incubating cells with Tn5 transposase loaded with Illumina Nextera

adaptors for 60 min at 37 °C. Paired end sequencing for mutant accessibility changes were performed using a Nextseq 2000 (Illumina). Sequencing was completed at the Emory Integrated Genomics Core and samples were sequenced at a depth of at least 50 M PE reads. Read trimming, alignment, and duplicate removal was performed using Illumina Dragen v4.2.4. BAM files were then sorted by query name rather than by read coordinate, a step necessary for the subsequent analysis using samtools v1.19.2. The BAM file metadata was then corrected to ensure proper pairing of reads using samtools v1.19.2. Quality control measures were applied to filter out low-quality reads and remove improperly paired or unmapped reads, ensuring only high-quality data were retained for further analysis using samtools v1.19.2. Following this, reads that mapped to non-chromosomal regions and those overlapping known blacklisted regions were excluded using bedtools v2.31.1. These blacklisted regions were sourced from the ENCODE project. After filtering, the BAM file was resorted to restore correct read alignment and indexed to facilitate efficient retrieval and downstream processing using samtools v1.19.2. To enable peak calling, the BAM file was converted to BED format using bedtools v2.31.1. Due to the use of Tn5 transposase during library preparation, read start positions were adjusted according to the known shift introduced by the enzyme, with strand-specific shifts applied. Peak calling was performed using the MACS3 algorithm v3.0.1 with a q-value of 0.01. The --nomodel option was used to avoid fragment size estimation, and instead reads were shifted -100 bp and extended 200 bp to approximate the size of accessible chromatin regions surrounding Tn5 insertion sites. Summits were called to define the point of maximal accessibility. We then established a consensus set of accessible regions observed upon wild type SMARCB1 re-expression to serve as a reference for identifying robust changes associated with SMARCB1 disruption.

We established a consensus set of accessibility peaks observed upon wild type SMARCB1 re-expression to serve as a reference for identifying robust accessibility changes associated with SMARCB1 disruption. ATAC-seq consensus peak sets were generated from MACS3 narrowPeak files by first identifying replicate-reproducible peaks separately for both WT and I315I conditions. Overlapping regions between biological replicates (R1 and R2) were identified using bedtools intersect with the -u flag, then peak coordinates were extracted, sorted by genomimc position, and merged using bedtools merge. A high-confidence wild type SMARCB1 accessible peak set was then generated by intersecting regions present in both I315I and WT replicate consensus peaks and merging to eliminate overlapping intervals. This approach ensured that the final consensus peak set was supported across all four individual replicates (I315I-R1, I315I-R2, WT-R1, WT-R2) and represented high-confidence accessible regions associated with functional SMARCB1 re-expression, yielding 92,028 consensus accessibility peaks. We then performed differential accessibility analysis using DiffBind v3.8.4 with DESeq2 for statistical testing. Two separate conditions were conducted: I315I vs I315* and WT vs W281*. Gained peaks were defined as regions showing increased accessibility in function SMARCB1 conditions (WT or I315I) with FDR < 0.05 and Log2FoldChange > 1. Lost peaks were defined as regions showing decreased accessibility in functional conditions with FDR < 0.05 and Log2FoldChange < -1. We then combined the significant peaks from both comparisons to create unified set gained and lost regions that differed between functional and non-functional SMARCB1. The wild type consensus accessible peaks were classified into functional categories by intersection with differential accessibility regions identified through DiffBind analysis. Peaks were categorized as: (1) significant gain peaks—consensus regions overlapping with unified gain regions from DiffBind comparisons; (2) significant loss peaks—consensus regions overlapping with unified loss regions; and (3) persistent peaks—consensus regions that did not overlap with any differential accessibility region. Classification was performed using bedtools intersect with appropriate inclusion (-u) and exclusion (-v) flags to ensure

mutually exclusive peak sets. Finally, the BAM files were converted to BigWig format using deepTools v3.5.4 to allow for efficient visualization. ATACseq intensity tornado plots were visualized using computeMatrix and plotHeatmap from deepTools v3.5.4. Replicates were confirmed to be consistent, and subsequently, the bigwigCompare command in deepTools v3.5.4 was used to average signal across biological duplicates for ATAC-seq signal intensity plots.

Corresponding RNAseq and ATACseq L2FC were assessed by annotating the closest gene associated with each differentially accessible peak using HOMER v4.11.1. The L2FC value associated with each peak was plotted with the corresponding L2FC for the annotated gene. A Fischer's exact Test was used to calculate significance using base R. Code for this analysis can be found at https://doi.org/10.5281/zenodo.18716542.

## CUT&RUN-seq

CUT&RUN analysis of SMARCA4, SMARCB1, SMARCE1 was performed as previously described[53] using the CUTANA EpiCypher ChIC CUT&RUN Kit Version 5 (EpiCypher, Cat# 14-1048) according to manufacturer's protocol. Briefly, 500,000 G401 cells per reaction were harvested following 8 days of doxycycline induction (1 µg/mL). Nuclei were extracted using the EpiCypher CUTANA Pre-Nuclei Extraction Buffer (EpiCypher, Cat#21-1026a) according to established protocols. Nuclei were attached to concanavalin-A-coated magnetic beads in Wash Buffer containing 0.01% digitonin by incubation for 10 min at room temperature. After removing the supernatant, the bead-nuclei mixture was resuspended in Antibody Buffer and incubated overnight at 4 °C on a nutator with 1 µL of primary antibodies: rabbit IgG control (EpiCypher, Cat# 13-0042k), anti-SMARCA4 (CST, D1Q7F), anti-SMARCB1 (CST, D8M1X), and anti-SMARCE1 (CST, E6H5J). Following two washes with Cell Permeabilization Buffer, 2.5 µL of CUTANA pAG-MNase (EpiCypher, Cat# 15-1016k) was added to each reaction and incubated at room temperature for 10 min with intermittent flicking. After two additional washes with Cell Permeabilization Buffer, 1 µL of cold $CaCl_2$ was added and reactions were incubated for 2 h at 4 °C on a nutator. Reactions were stopped by adding 33 µL of Stop Buffer containing 0.5 ng of *Escherichia Coli* spike-in DNA (EpiCypher, Cat# 18-1401k) and vortexing briefly. The mixture was incubated at 37 °C for 10 min to release fragmented DNA, then placed on a magnet for 2 min. The supernatant was collected and purified using 1.4X volume SPRI beads with two washes in fresh 85% ethanol. DNA was quantified using the 1X dsDNA HS Assay Kit. The sequencing libraries were prepared using CUTANA CUT&RUN Library Prep Kit v1.5, according to manufacturer's protocols. DNA fragments were end repaired and adaptors ligated using provided reagent, followed by 1X SPRI bead cleanup with two 85% ethanol washes. Following indexing PCR, libraries underwent a final 1x SPRI bead cleanup. Sequencing was performed at Novogene using the NovaSeqX Plus platform with 5% PhiX spike-in.

## Sequencing analysis of CUT&RUN data

Raw paired-end reads (150 bp) were processed using Illumina DRAGEN v4.2.4, including adapter trimming, duplicate removal, and alignment to the GRCh38.p13 reference genome. Aligned reads were cleaned and processed using a custom bash pipeline. BAM files were initially sorted by query name using samtools and filtered to retain high-quality paired reads (MAPQ ≥ 10), while excluding unmapped, unpaired, duplicate, and secondary alignment reads (SAMtools flags -F 3852 -f 2). Reads mapping to ENCODE blacklisted regions (ENCFF356LFX.bed) and non-canonical chromosomes (chrM, chrUn, random contigs, chrEBV) were removed using bedtools intersect and awk filtering. Filtered BAM files were converted to paired-end BED format (BEDPE) using BEDTools bamtobed, and fragment pairs were further filtered to retain only same-chromosome pairs with fragment sizes <1000 bp. Fragment coordinates were extracted as BED intervals representing the span from the 5′ end of the first read to the 3′ end of the second read.

Coverage tracks were generated using BEDTools genomecov with the GRCh38.p13 chromosome size file to produce bedGraph format files showing fragment density across the genome. All processing steps were parallelized across 18 cores using GNU parallel, with individual sample logs retained for quality control.

Normalization factors were calculated using the relative proportion of spike-in DNA in each sample using the following equation:

$$\frac{1}{\left(\frac{number\ of\ uniquely\ aligned\ E.\ coli\ reads}{number\ of\ uniquely\ aligned\ human\ reads}\right) \times 100} \qquad (2)$$

BedGraph files were then normalized by multiplying coverage values by the sample-specific normalization factor using awk. Peak calling was then performed using SEACRv1.3[54] in stringent mode with normalized bedgraph files and corresponding IgG controls for background subtraction. SEACR was run separately for each biological replicate using the non-AUC threshold and stringent peak calling parameters to identify high-confidence binding sites. To ensure peak quality, peaks were filtered based on maximum signal intensity, retaining only peaks with intensity >5 after examining the distribution of peak intensities across all samples. Reproducible peaks were identified by finding overlapping regions between biological replicates using BEDTools intersect, requiring peaks to be present in both replicates of the same genotype-target combination.

To comprehensively identify regions bound by the SWI/SNF complex, we generated consensus binding sites by taking the union of reproducible peaks for the three core subunits (SMARCA4, SMARCB1, SMARCE1) separately for the wild type and I315I conditions. This union-based approach captures the full spectrum of SWI/SNF occupancy, accounting for the dynamic and transient nature of chromatin remodeling complex binding. For each condition, reproducible peaks from all three subunits were concatenated, overlapping or proximal regions (within 100 bp) were merged using BEDTools merge to create condition-specific unified binding regions. To define the most confident SWI/SNF binding sites, we then intersected the wild type and I315I unified regions, retaining only sites that showed evidence of binding in both genetic backgrounds. This approach ensured that consensus binding sites represented high-confidence SWI/SNF occupancy supported across both genetic backgrounds.

The resulting consensus SWI/SNF binding regions were then classified according to their chromatin accessibility changes between conditions. SWI/SNF binding sites were categorized into three mutually exclusive groups using BEDTools intersect: (1) sites with significant accessibility gains in the wild type condition (overlapping with significantly gained ATAC-seq peaks), (2) sites with significant accessibility losses (overlapping with significantly lost ATAC-seq peaks), and (3) sites with unchanged accessibility (not overlapping with either significantly altered ATAC-seq peak set). This classification enabled analysis of SWI/SNF occupancy in the context of condition specific chromatin accessibility dynamics.

## Molecular dynamic modeling

MD simulations were performed using the AlphaFold2 structure for SMARCB1 (UniProt ID: Q12824) as the starting conformation[34]. The structure file was obtained from the AlphaFold Protein Structure Database and used directly as input for all MD simulations. Mutations were introduced using Pymol v2.5.4. All MD simulations were conducted using AmberTools v25 and Amber24[55], and performed on NVIDIA GPUs using pmemd.cuda. The AlphaFold2 structure for SMARCB1 was prepared using pdb4amber and parameterized with the Amber ff19SB force field. The protein was solvated in a cubic TIP3P water box with a 15.0 Å buffer, and sodium ions were added to neutralize the system using tleap.

Energy minimization was performed in two sequential stages using pmemd.cuda. First, restrained minimization was conducted for

10,000 steps (500 steps steepest descent followed by conjugate gradient) with harmonic restraints (10.0 kcal/mol·Å²) applied to all protein heavy atoms (residues 1–385). Second, unrestrained minimization was performed for 10,000 steps using the same protocol without positional restraints. Both minimization steps used constant volume periodic boundary conditions with a 10.0 Å nonbonded cutoff. System equilibration proceeded through two phases with convergence monitored through potential energy and system density stabilization. NVT heating equilibration was performed for 10 ns (5 million steps, dt = 2 fs) starting from 0 K and heating to 310 K using a Langevin thermostat with collision frequency of 1.0 ps$^{-1}$. SHAKE constraints were applied to hydrogen bonds. Subsequently, NPT equilibration was conducted for 5 ns (2.5 million steps, dt = 2 fs) at 310 K and 1 atm pressure using isotropic pressure scaling with a relaxation time of 2.0 ps and increased collision frequency of 5.0 ps$^{-1}$ for faster equilibration. Production molecular dynamics was performed in the NPT ensemble for 200 ns (100 million steps, dt = 2 fs) at 310 K and 1 atm using pmemd.cuda. Temperature coupling employed a Langevin thermostat with collision frequency of 1.0 ps$^{-1}$, and pressure was maintained using isotropic scaling with a 2.0 ps relaxation time. SHAKE constraints were applied to hydrogen bonds, and coordinates were saved every 100 ps (50,000 steps) in NetCDF format with periodic boundary wrapping. Trajectory analysis was performed using cpptraj to assess structural stability, conformational changes, and dynamic properties throughout the simulation.

### Molecular dynamics trajectory analysis

Trajectory analysis was performed using VMD (v1.9.3)[56] and pairwise interaction energies were calculated with the NAMD Energy plugin (v2.14)[57]. For each inter-residue or inter-domain comparison, van der Waals or electrostatic interaction energies were extracted from all trajectory frames and averaged to obtain a mean interaction energy per replicate. To quantify the effect of each mutation, ΔE was calculated as the difference between the mean mutant and wild type interaction energies:

$$\Delta E_{mutant} = E_{mutant} - E_{WT} \qquad (3)$$

To control for simulation-specific artifacts, ΔE values from experimental mutants (W281P and I315R) were further normalized by subtracting the ΔE of a control population variant (S299L) for each replicate:

$$\Delta\Delta E_{W281P} = \Delta E_{W281P} - \Delta E_{S299L} \text{ or } \Delta\Delta E_{I315R} = \Delta E_{I315R} - \Delta E_{S299L} \qquad (4)$$

These ΔΔE values were averaged across 12 independent replicates, and statistical significance was assessed using a two-tailed one-sample *t*-test against a null hypothesis mean of zero. Interactions were considered biologically significant only if they met both statistical and effect-size thresholds: $p < 0.05$ and $|\Delta\Delta E| > 0.5$ kcal/mol for VDW interactions or $|\Delta\Delta E| > 1.0$ kcal/mol for electrostatic interactions. Only interactions fulfilling both thresholds are reported as significant.

Residue-residue contact frequencies were calculated using an in-house VMD TCL script. Contacts were defined as any pair of heavy atoms within 4.0 Å across all trajectory frames. For each trajectory, the frequency of contact between residues in cleft 1 (residues 260–283) and cleft 2 (residues 290–324) was computed:

$$f(res_1, res_2) = \frac{n_{contact}}{n_{frames}} \qquad (5)$$

where $n_{contact}$ is the number of frames in which the residue pair is in contact. For each replicate, contact difference maps were first generated for WT versus experimental mutant (W281P or I315R) or for WT

versus the control variant S299L:

$$\Delta f(res_1, res_2) = f_{WT}(res_1, res_2) - f_{Mut}(res_1, res_2) \qquad (6)$$

To normalize for simulation-specific artifacts, the WT-S299L difference was subtracted from the WT-experimental mutant differences for each replicate and residue pair, yielding normalized difference maps:

$$\Delta\Delta f_{W281P}(res_1, res_2) = \Delta f_{W281P} - \Delta f_{S299L}, \Delta\Delta f_{I315R}(res_1, res_2) = \Delta f_{I315R} - \Delta f_{S299L} \qquad (7)$$

Positive values indicate contacts preferentially lost in the experimental mutant relative to the control, while negative values indicate contacts gained. All analyses were performed on 12 independent replicates to ensure statistical robustness. All code for MD simulations can be found at https://doi.org/10.5281/zenodo.18716542.

### Quantification and statistical analysis

GraphPad PRISM 10, R (v4.4.2), and Python (scipy.stats) were used to perform statistical analysis. For pairwise comparisons between two groups, a two-tailed student's unpaired *t*-test was used. For statistical analysis of ATAC-seq and RNA-seq concordance, observed counts for each quadrant were compared to expected counts under a null hypothesis of uniform distribution using Fisher's exact test to assess deviations from random association. Statistical analysis of RNA-seq differential expression analysis and ATAC-seq differential accessibility analysis was performed using DESeq2. For RNA-seq analysis, all genes that had an FDR < 0.05 and log2FoldChange > |2| were considered significant. For ATAC-seq analysis, all peaks that had an FDR < 0.05 and log2FoldChange > |1| were considered significant. At least two independent experiments with a least two technical replicates were performed to support statistically analyzed findings.

### Reporting summary

Further information on research design is available in the Nature Portfolio Reporting Summary linked to this article.

## Data availability

The plasmids generated in this study have been deposited in the Addgene database [https://www.addgene.org/Andrew_Hong/]. The *SMARCB1* DMS library is available through the Broad Institute's Genomic Perturbations Platform [https://www.broadinstitute.org/genetic-perturbation-platform]. The WGS, RNA-seq, ATAC-seq, and CUT&RUN data generated in this study have been deposited in the dbGaP database under accession code phs003896.v1.p1 [https://www.ncbi.nlm.nih.gov/projects/gap/cgi-bin/study.cgi?study_id=phs003896.v1.p1]. These data are available under restricted access to protect patient privacy in accordance with the informed consent under which samples were collected. Access can be obtained by submitting a request through dbGaP [https://grants.nih.gov/policy-and-compliance/policy-topics/sharing-policies/accessing-data/dbgap]. The mass spectrometry data generated in this study has been deposited in the PRIDE database under the accession code PXD062226. The molecular dynamics simulation coordinates (initial and final configurations) generated in this study have been deposited in Zenodo and are available at https://doi.org/10.5281/zenodo.18929429[58]. The SMARCB1 protein structure used in this study is available in the UniProt database under accession code Q12824. Source data generated in this study are provided in the Supplementary Information/Source Data file. Source data are provided with this paper.

## Code availability

All custom code used in this study is available on GitHub (https://github.com/thehonglab/SMARCB1_DMS) and archived on Zenodo (https://doi.org/10.5281/zenodo.18716542).

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

## Acknowledgements

Research reported in this publication was supported by the following: NIGMS T32GM008490 (G.W.C.), NCI F31CA278008 (G.W.C.), NCI R01CA289761 (A.L.H.), DOD HT9425-23-0609 (Y.S.), DOD W81XWH1910281 (A.L.H.), ACS MRSG-18-202-01 (A.L.H.), R01-GM148586 (J.C.G), R01-HL168894 (K.N.C), R01-HD102534 (D.U.G). Wong Family Award in Translational Oncology (A.L.H.). Team Lick Cancer (S.N.C. and A.L.H.). Research reported in this publication was supported in part by the Emory University Emory Integrated Proteomics Core Facility (RRID:SCR_023530) and in part by the Emory Integrated Genomics Core (EIGC) (RRID:SCR_023529)—both shared resources of Winship Cancer Institute of Emory University and NIH/NCI under award number P30CA138292. Additional support was provided by the Georgia Clinical & Translational Science Alliance of the National Institutes of Health under Award Number UL1TR002378. The content is solely the responsibility of the authors and does not necessarily reflect the official views of the National Institutes of Health. The authors would like to acknowledge the American Association for Cancer Research and its financial and material support in the development of the AACR Project GENIE registry, as well as members of the consortium for their commitment to data sharing. Interpretations are the responsibility of the study authors. We thank the patients and their families for their participation. We thank Carlos Moreno, Roger Deal, and David Katz along with the Hong, Gorkin, and Spangle labs for their thoughtful comments and suggestions.

## Author contributions

G.W.C., S.N.C., A.L.H. designed the study. G.W.C. and A.L.H. wrote the manuscript. X.Y., R.E.L., F.P., A.O.G., T.P.H., A.L.H. developed the SMARCB1 DMS pools. W.J.K., A.L.H. performed the pooled screens. G.W.C., E.S., J.C.G. performed molecular dynamic studies. G.W.C., Y.S., V.Z.C. performed cellular experiments and growth assays. G.W.C. and P.B. performed LC-MS/MS experiments and mass-spec processing. G.W.C, B.P.L., and X.Y. performed computational analyses and interpreted results. K.N.C., D.E.R., B.L., W.C.H., D.U.G., J.A.B., S.N.C, and A.L.H. supervised the studies. All authors discussed the results and implications and edited the manuscript.

## Competing interests

G.W.C. completed an internship with GRAIL, Inc in Summer 2024. R.E.L. is currently employed by RAN Biotechnologies. F.P. is a current employee of Merck Research Laboratories. D.E.R. receives research funding from members of the Functional Genomics Consortium (Abbvie, BMS, Jannsen, and Merck), and is a director of Addgene, Inc. W.C.H. is a consultant for Thermo Fischer, Solasta Ventures, KSQ Therapeutics, Frontier Medicines, Jubilant Therapeutics, RAPPTA Therapeutics, Serinus Biosciences, Kestral Therapeutics, Crane Biotherapeutics, Function Oncology, Perceptive, Biotherapeutics, Function Oncology and Calyx. The remaining authors declare no competing interests.
