## [Transparent Peer Review file · Nature Communications]

SMARCB1 missense mutants disrupt SWI/SNF complex stability and remodeling activity

Corresponding Author: Dr Andrew Hong

Version 0:

Reviewer comments:

Reviewer #1

(Remarks to the Author)

SMARCB1, a core subunit of the SWI/SNF complex, is biallelically inactivated in approximately 95% of Malignant Rhabdoid Tumor (MRT) cases. However, the functional impact of SMARCB1 point mutations remains poorly understood, as they are rare outside of a few cancer types and syndromes like Coffin-Siris syndrome.

In this manuscript, Cooper et al. employ deep mutational scanning (DMS) in rhabdoid cells to systematically investigate the functional consequences of SMARCB1 point mutations. Their mutagenesis study identifies critical domains and residues essential for SMARCB1 activity while reinforcing its role in SWI/SNF complex stabilization. Importantly, this work challenges the conventional view that complete loss of SMARCB1 is necessary for disease pathogenesis, suggesting that certain missense mutations may also contribute to oncogenic dysfunction.

These findings hold significant clinical and research implications, offering new insights into SMARCB1 biology and potential diagnostic or therapeutic considerations. While the manuscript is well-constructed and compelling, the following points should be addressed to strengthen its impact and clarity:

Major Comments:

1. Limitations of the proliferation assay for R377H in Coffin-Siris syndrome (CSS)

o The conclusion that "predicted pathogenicity does not correlate with patient frequency" may be overstated. Since CSS arises from developmental defects rather than hyperproliferation, the lack of correlation could reflect negative selection during embryogenesis—only viable mutations lead to live births with CSS.

o Additionally, CSS patients with SMARCB1 mutations often exhibit severe clinical outcomes, suggesting these variants are indeed pathogenic. Thus, the discrepancy between CADD predictions and patient frequency might stem from an inappropriate phenotypic readout (proliferation) rather than false-positive predictions.

2. Quantification of SMARCB1 protein levels for W281P and I315R

o The Western blot in Fig. 3a suggests reduced protein levels for these mutants compared to WT. To strengthen the claim that they functionally mimic nonsense mutations, the authors should quantify SMARCB1 protein levels (e.g., via densitometry normalized to a loading control).

3. Evidence for SWI/SNF complex destabilization by W281P and I315R

o The assertion that these mutations disrupt SWI/SNF assembly/stability relies on proteomics data. To validate this, the authors should perform density sedimentation assays (e.g., sucrose gradient centrifugation) to assess complex integrity. Confirm via immunoblotting SWI/SNF subunits in mutant vs. WT cells.

4. Impact of W281P/I315R on SWI/SNF targeting

o Do these mutations alter chromatin occupancy of the SWI/SNF complex?

o Cut&Run or ChIP-seq for ARID1A/SMARCA4 in W281P/I315R vs. WT could clarify whether genomic targeting is affected.

5. Global chromatin accessibility changes in W281P/I315R mutants

o Fig. 4a focuses on differential peaks, but a genome-wide heatmap comparing WT vs. W281P and WT vs. I315R would better illustrate the scale and distribution of accessibility changes.

Minor Comments:

• Line 177: The linked heatmap fails to load ("data too large"). Please provide an accessible version.

• Figure consolidation: Some panels are redundant (e.g., Fig. 3c and 3d convey similar data, with 3d being clearer). Condensing such repetitions would improve readability.

(Remarks on code availability)

Reviewer #2

(Remarks to the Author)

This manuscript systematically assesses the effects of point mutations to the SMARCB1 tumor suppressor by deep mutational scanning. The study finds that groups of residues that are clustered in three-dimensional space have destabilising effects on SMARCB1 activity that are comparable to truncating mutations. This suggests that a proportion of missense SMARCB1 mutations are likely to be oncogenic. This provides deeper insight as to how SMARCB1 functions in chromatin and cancer biology.

The following questions should be addressed:

The effects of SMARCB1 mutations were assessed using an expression system in a SMARCB1 deficient cell lines. If this system fully recapitulates the oncogenic effects of SMARCB1 mutations, then it would be anticipated to reflect the lower depth patient mutation data. Looking at the data for truncating mutations, this does not appear to be the case. Missense mutations are broadly distributed across the gene in the patient data (Figure 1b) but depleted from the N-terminal 50 aa in the re-expression assay (Figure 1f). As a result, the authors should discuss the potential limitation that the three regions of enhances mutation meeting the two standard deviation cutoff, are specific to the re-expression system and may not function similarly in humans. In some cases text such as “we show that missense mutations in the RPT2 domain of SMARCB1 disrupt tumor suppressor function” should be modified to remove tumor suppressor function as this has not been tested directly.

Changes in SMARCB1 associated proteins are plotted as meeting a t-test difference (Figure 3b). It would be also useful to show this data as fold-changes in the abundance of subunits within complexes. If the change in association is less than 50%, then a substantial proportion of complexes including the SMARCB1 mutants remain intact.

If the strong effects on chromatin accessibility are compared to the proportion of complexes remaining substantially intact, is it possible that mutations primarily affect the activity of complexes rather than acting by destabilisation a combination of the two, or destabilisation only as suggested in the text?

Supporting a lack of involvement in cancer the manuscript states “individuals with Coffin-Siris syndrome harboring germline mutations in the C 133 terminal CCD domain of SMARCB1, and specifically R377H (ClinVar Accession: VCV000030203.13), do not develop cancers at higher rates than the population” However, the fact this alteration is detected in patients with Coffin-Siris syndrome indicates that it is likely to be functional.

(Remarks on code availability)

Reviewer #3

(Remarks to the Author)

The tumor suppressor gene, SMARCB1, is biallelically inactivated through mutagenesis in a few pediatric malignancies, yet the pathologic significance of missense mutations remain unclear particularly given the lack of hotspot variants. In this manuscript, the authors carry out a deep mutational scan (DMS) of SMARCB1 in order to identify specific missense mutations that can disrupt SMARCB1 tumor suppressor function. The authors screened for fitness effects conferred upon expression of a DMS orf library of SMARCB1 variants in 3 separate SMARCB1-deficient cancer models. Despite identifying a few different “clusters” of candidate loss of function missense mutations (ie. WHD domain, IDR and RPT2), the authors focused their more extensive experimental follow up to only 2 variants that mapped to the RPT2 domain.

Overall, the data generally supported that the 2 residues they chose to experimentally model resulted in a disruption of SWI/SNF activity in this context. However, the interpretation was limited by the fact that one engineered residue (W to a proline) would likely be expected to cause a significant disruption to protein structure/function and the other variant would likely not occur in nature due to codon usage (Ile>Arg, would require multiple mutations). The engineered non-sense mutations at these residues would be expected to be detrimental. Hence, it still remains a question regarding the pathologic relevance of the RPT2 domain variants in cancer. The paper could have benefited by a more rigorous assessment of additional RPT2 variants, with a consideration for naturally occurring variants in this domain.

Comments:

(1) For the initial assessment of missense mutations across cancer (Fig. 1), can the authors provide any information on zygosity/LOH regarding missense mutations given that biallelic loss is necessary. (I realize the data is limited, but it seems important to understand if any of the missense mutations found in cancer are homozygous, given that this genetic state is modeled in the paper)

(2) In building their rationale for carrying out this assessment, the authors initially test the R377H and R377* variants for their

ability to confer tumor suppressor function. They state that “the R377* mutation partially compromises tumor suppressor function, while the R377H mutation retains its function”. This statement is not supported by the data (extended figure 1b, 1c), as both mutants perform almost identically in suppressing cell growth.

(3) In further support of their rationale for the screen, the authors make the point that the computational algorithms predict for a large proportion of missense mutations being pathogenic based upon a threshold value (?) for each algorithm and this does not correlate with observed patient frequencies. Nor there a statistically-significant correlation when the computational algorithm scores are treated as a continuous variable (Ext. Fig 1h). Although this is a minor point, if one selects a more stringent predictive score (Ext. Fig 1h), there is a stronger likelihood of correlating with higher mutation counts (although not completely predictive). The authors screening data in Fig 1g further supports the greater likelihood for a pathogenic variant if a higher CADD score was selected in this case. Just a minor point, as the authors rationale for testing variants still holds true, but there is a lot of “build-up” to get to the rationale for running the screen that is probably not critical for the paper.

(4) Fig 2d is one of the most critical pieces of data in the paper, as it takes into account all 19 missense mutations at the given position. The accompanying table only provides the averaged values. Given that the authors highlight only a handful of potential LOF variants, it would be informative for the reader to visualize the z-score enrichments across all variants at these positions to ensure that the data is not being skewed by specific variants (particularly since the data is averaged).

(5) In general, the crystal violet assays to assess growth are noisy and there are no replicates. Although trypan blue cell counts were also collected, the overall interpretation was not fully clear for some variants. For instance, expression of the D227K mutation had no observable effect on growth in the crystal violet assay (Fig. 2h), yet still results in an appreciable inhibition of growth in the cell counts (Fig 2i). (similarly for E300K) Furthermore, interpretation of the corresponding nonsense mutations should not be made given that these mutations were not expressed to the same degree (as may be expected given NMD)

(6) In the IP-mass spec study (Fig. 3b), the authors indicate that several proteins are enriched in the mutant condition, suggesting “mutant-specific functions”. The fold enrichment and degree of significance is minimal. Without experimental follow up, I don’t think these observations require being called out in the body of the paper.

(7) With the exception of one Western blot (Ext. data Fig 4c), there is little validation data supporting some of the points being made (lines 218-328), nor with respect to the latter commentary on QSER1 (lines 382-386).

Minor points:

(1) No description in the paper how pathogenicity scores were defined for CADD, AlphaMissense and REVEL.

(2) In Fig 1d, BT16 shows 2 indels, but the authors indicate in the legend that it has both a monoallelic deep deletion and a frameshift indel.

(3) Supplemental Table 6 is anonymized. It would be useful to at least annotate some control genes (non-essentials, essentials) to better understand the performance of the screen.

(4) The colors are not clear on Fig. 4e.

(Remarks on code availability)

Reviewer #4

(Remarks to the Author)

The authors conducted deep mutational scanning (DMS) of SMARCB1, evaluating 8,418 variants across three SMARCB1-deficient cancer cell line models. DMS was performed with a proliferation screen. They used a z-score >2.0 per mutation (or average Z score >2SD per residue) to define top-hit loss-of-function variants for follow-up, finding 101 mutations or 13 residues above that cutoff. They integrated multiple datasets (AACR GENIE, COSMIC, gnomAD) and computational predictors (CADD, REVEL) to contextualize findings. They validate 12 sequences (including 4 missense mutations) and focus on 2 RPT2 domain mutations (W281P, I315R) that destabilize the SWI/SNF complex and impair chromatin remodeling without complete protein loss, challenging reliance on IHC for diagnostics. They interestingly were able to link structural flexibility of the winged-helix domain to these functional outcomes using molecular dynamics simulations, but did not clearly explain how the RPT2 mutations would plausibly affect flexibility of the distal WHD. They supported findings for those 2 mutants with ATAC-seq (chromatin accessibility), RNA-seq (transcriptional changes), and proteomics (SWI/SNF subunit dissociation), very exhaustively strengthening the claim that those mutations have a major impact on function.

The most impactful aspect is that they demonstrated that missense mutations can mimic nonsense mutations functionally; this is not very surprising biologically, but it does suggest current IHC diagnostic methods may miss pathogenic variants. They also highlighted discordance between mutation recurrence (e.g., R377H) and pathogenicity, emphasizing the need for these functional assays. However, it is not entirely clear these 2 deeply characterized mutations will be found in many patients or that their map can be used to confidently predict the pathogenicity of other mutations that will be found in patients. In my view, the authors appropriately largely focus their claims on the strongly supported evidence for the validated mutations, without overstating how much the DMS data can serve as a resource for predicting pathogenicity of other mutations.

I commend the authors for making their code available, including for reproducing analyses.

Major Comments

1. The work suggests SMARCB1 is surprisingly robust to mutation (e.g. compared to CADD). Is it also possible their DMS is not sensitive to measure mutation impacts? One reason I wonder is because usually proline is very effective at breaking alpha helices across positions, but even proline rarely has a large impact here. Is this simply how the data is visualized? I am concerned about a technical issue affecting the dynamic range or noise in this assay. SMARCB1 was re-expressed via

inducible vectors, which may not reflect endogenous protein levels or stoichiometry within the SWI/SNF complex; could that affect sensitivity? Figure 1F heatmap looks noisy, in the sense that few residues have consistent phenotypes when mutated to related amino acids. Put another way, the map doesn't have many stripes or blocks that are often seen in DMS. What is the correlation of biological replicates of fold changes over time (Z-scores?) Correlation of counts can provide an inflated view of reproducibility. How do results compare across cell lines?

2. I have a concern that the approach of combining the plasmid pools for different tiles into one pool created more issues than it solved. How would lentiviral recombination affect the results, would many cells receive a transgene with multiple mutations? How about variable PCR efficiency of different fragmented regions due to using Nextera? Is this why Z-scores compare to the corresponding silent mutation? It seems to me these two different approaches (screening one tile at a time versus full genes at a time) are being used across different DMS papers, and the authors insights on how to choose would be appreciated.

3. Can they explain why the RPT2 mutations would affect the distal WHD flexibility? To add controls, would these simulations show different results when tested on different mutations? And how does this WHD flexibility explain the proteomics result?

4. Can they better address implications for diagnostics? I thought their critique of IHC-based diagnostics was impactful, but the study does not clearly propose alternatives (e.g. exome sequencing) for clinical use or explain what impact exactly these improved diagnostics would have. There is no discussion of how these findings could inform targeted therapies (e.g., restoring SWI/SNF complex composition).

Minor comments

5. Paper could benefit from better organization, logical flow, and curation of the main figures to directly show what is being compared. Some optional suggestions as examples: I think Figure 1A-D could be supplementary so the panels are focused on Deep Mutational Scanning (the title of the figure). Maybe a summary or average of CADD, GENIE, and/or COSMIC scores could be added as row(s) to the DMS heatmap for direct and compact comparison. The RNA-seq is introduced at two separate points (~line 283 and ~388). Fig 1C and 2A are described in the text as comparable and could be combined into one figure for ease of comparison. I felt the figure 1 ClinVar analysis could better come after fully describing results of the screen (i.e. the beginning of figure 2).

6. Could they add the percentage of persistent peaks that have AP-1 motif?

7. Do the DNA oligos only mutate a single nucleotide per oligo (not mentioned in Methods)? How often do the sequencing reads contain a single mutation or multiple or none?

8. The Fig 1H,I ClinVar ROC is suggestive that the screen provides improved clinical predictive power, but it seems like a severe limitation that there are only 2 cancer variants. Could it be useful to extend the evolutionary conservation and gnomAD analyses across the whole DMS?

9. Label Z-score directions with loss/gain of function for improved interpretability. The mutated residues could be labeled in the video. The significant residues could be labeled in the DMS heatmap. The WHD domain could be labeled in the structure figure.

10. Line 135: could more directly say mutation recurrence does not correspond to proliferation in this assay. At this point, it remains to be proven that proliferation in this assay corresponds to pathogenicity. Line 400: the authors evaluated whether changes in chromatin accessible were associated with changes in transcription (causation is not determined).

11. More acronyms and gene names could be defined for a broader audience, e.g. cBAF vs PBAF.

12. The paper says both R52 and R53, I think there is a typo.

13. I could not find the ORFeome screen described in Methods. Methods do not say how the DMS screen was performed. What is $t=0$ and what is $t=end$ for each screen? Are there other timepoints too (suggested in Fig 1E)? Do longer endpoints allow for greater dynamic range in the assay? Was anything done to ensure the low multiplicity of infection?

(Remarks on code availability)

Reviewer #5

(Remarks to the Author)

Summary

This is a compelling work where the authors explore the function of missense mutations in the BAF complex subunit, SMARCB1, which are part of the complex's early-stage formation. The field has focused on immunohistochemistry quantification of SMARCB1 in pediatric tumors, yet missense mutations that may alter function but not expression level may be missed. Deep Mutation Scanning has been applied to more frequently altered genes, and the authors apply it now in one of the first uses to characterize a low-frequency tumor suppressor gene. They found that missense variation in SMARCB1 can cause genome-wide changes that are heterogeneous with details on a per-allele basis. The manuscript is well-written and organized. Overall, I believe it presents information with high relevance and reusability. The supplemental tables are commendably complete. While data is not all publicly accessible, it is appropriately housed in dbGaP as explained on their github page, which is publicly accessible.

Points to Improve the Manuscript

- More recent works by, e.g., Kadoch lab, have characterized the order of formation for the canonical versus non-canonical BAF complex. The current manuscript would be improved by expanding the Discussion to include the author's interpretation of SMARCB1 mutation in light of the different functions of the canonical versus non-canonical BAF in pediatric malignancies. Fig 3e shows it is mostly base unit loss for each subfamily.
- The supplemental video of MD simulations shows that global translational and rotational degrees of freedom may not have been accounted for before analysis. If the video is a visualization of the data as it was analyzed, then I have concerns about the alignment-dependent metrics like RMSD and RMSF.

- What is the doubling time of G401 cells? How many divisions do you expect in the 10-days of the DMS study? Can you set an expected limit of detection based on that expectation?
- While I like Figure 2e, I think the paper would be improved by a structural assessment of where and why there are amino acid type dependencies for the structures. That is, it looks to me like AlphaMissense is basically saying most changes at any structured region are damaging. Yet, the author's data indicates that it is more subtle, and the damaging effect depends on the alternate amino acid per site. I would like to see more information about this level of data (for any MAVE / DMS, but for this paper too!).
- Candidate loss-of-function variants are discussed, yet the authors data indicate that similar Z-score magnitude change, but in the negative direction, could be used to prioritize and compare candidate gain-of-function variants.
- Are there patterns among or explanation for why about half of the nonsense mutations are $Z < 2$ in the current study? The concordance of half is emphasized, yet the discordance of half is interesting.
- Are discordances between DMS results and CADD/others due to sites where the changes in isoform 1, targeted by the mutagenesis library, have the same type of change in other isoforms?
- I would like to see much more direct comparison of the data from GENIE and Cosmic, with the DMS results. Fig 1a should be log scale since just a few are highly recurrent variants dominate the view. Then, is there a relationship between the DMS incidence and tumor incidence? This is not made directly clear. There is an important baseline assumption that mutating the chromatin remodeling has a malignant effect via a mechanism of altered cell growth. It may not be. It may be stemness, fate, plasticity, etc., especially given the author's data that there are differences in genome-wide regulation downstream of different missense mutations. Not all mutations may promote malignancy by the same mechanisms. I would like this assumption due to the design of the experiments to be stated and explained up-front. Then, the comparison of DMS and somatic incidence will indirectly support/not. Finally, that will help frame the other results of the paper.
- The authors report that, "all three computational predictors identified a high proportion of missense variants as pathogenic [...]. This finding contrasts with the low somatic missense mutation frequency of SMARCB1 in patient tumor samples [...]. If such a large proportion of variants were truly pathogenic, we would anticipate a higher frequency of missense mutations in SMARCB1 in patients." I do not fully agree with the logic used in this statement. It seems entirely possible that a gene could be mutationally intolerant and yet not mutated in tumors as long as those mutations did not confer a niche advantage to malignant cells. I recommend the authors reorient this section. It's already clear to me, for example and from my assessment of many other datasets, that AlphaMissense over-calls at structured sites. I recommend something around this type of finding be more the orientation and as it pertains to your DMS measurements and where they are and are not concordant.
- Molecular Dynamics questions
 - o The authors state that the AF2 structure was used, but it's not completely clear if that is what was use for the MD input. Please clarify this in Methods.
 - o The authors report that "Harmonic positional restraints with a weight of 2.0 kcal/mol-Å^2 were applied to the alpha carbons ($C\alpha$) of residues 1–385 to maintain structural integrity."
 - Is that 2 kcal/molÅ^2 distributed across the 385 $C\alpha$ atoms, or per atom? Please make clear in the sentence.
- The IDR appears to receive the same treatment as the ordered domains. Please discuss strengths and weaknesses of this approach. I would, for example, like you to not include IDR in some calculations like RMSD – plus, what is the per-domain RMSD? That can be telling to understand what is intra-domain and inter-domain flexibility changes. Plus, that there are multiple experimental structures with the WHD in different positions – how do you interpret the changes in flexibility observed? Are some mutants more likely to be found interacting with the DNA in one way, and other mutations in another?
- The dbGaP links or IDs should be substituted for the "[X]s" in the github page so that interested users can more readily find and access the controlled access data.
- Other variations in the three patient-derived SMARCB1-deficient cell lines?
- Does the rolling-average of silent mutations approach lead to an estimatable expectation for the minimum effect size that a "truly damaging silent mutation" would have to achieve to have a significant Z-score? And, what if there were regions where all variations had subtle but consistent effects? Consider discussing these as limitations of the approach. Can you consult the eQTL Catalog to demonstrate that there are no eQTL silent mutations here, for a different approach to justify your finding that nearly no silent mutations had an effect on cell growth?

(Remarks on code availability)

I marked "Yes" because I did look at their markdown code in github, but I did not attempt to clone and reproduce. The code looks reasonable and has adequate comments for those familiar with the types of data being handled, in my opinion.

Version 1:

Reviewer comments:

Reviewer #1

(Remarks to the Author)

The reviewers have answered all my questions.

(Remarks on code availability)

The reviewers have answered all my questions.

Reviewer #2

(Remarks to the Author)

After reading through the response to reviewers it is clear that the authors have moderated the manuscript to acknowledge limitations of the expression system used and differences in its use to detect phenotypes relating to CSS or cancer. The evidence is also strengthened that the mutations characterised in more detail act by disrupting the integrity of complexes. As a result I consider the revised manuscript suitable for publication.

(Remarks on code availability)

Reviewer #3

(Remarks to the Author)

The effort the authors took to address the reviewer comments is appreciated. Overall, the authors sufficiently addressed my own comments.

(Remarks on code availability)

Reviewer #4

(Remarks to the Author)

The manuscript is improved in several ways, e.g. by removing the section on WHD flexibility and replacing it with a stronger Figure 4 about their new RPT2 molecular dynamics simulations.

To me, the biggest issue is that it remains unclear that the data quality achieves the level of a resource wherein the screen can be trusted to provide information about all the missense mutations. However, I don't think this issue is reason to reject the paper, because they thoroughly follow up to validate a handful of their hits and demonstrate what some of these missense mutations do to the complex and its impact on chromatin (i.e. the DMS clearly finds true positives). I appreciate the additions they made to the figures, Methods, and Discussion that address my previous points.

Major comment:

1. Screen noise seems unfortunately high, as now shown in Ext Fig 2A, despite high coverage growth screens in other contexts providing very reproducible data.

a. I suspect this technical variability has to do with some combination of lentiviral recombination, PCR recombination, potentially too little growth time, and potentially too low cellular coverage (during growth or genomic PCR). The shape of the reproducibility plots seems suggestive of some kind of bottlenecking or jackpot effect with elements highly enriched in a single replicate.

b. However, I recognize higher cell coverage can be especially difficult with adherent cell lines. To me, 8 days seems short for a growth screen, but I note the authors have had a different experience (mentioned in the rebuttal). It looks like MOI was in a good range to avoid multiple infections (19.5-35.4%) and did not end up at the high end of MOI=0.6, which would have resulted in 25% multiple-infected cells. And, overall mutation effect calling is largely similar across cell lines, as shown in the rebuttal, but that is largely for nonsense/frameshift mutations which are, in a sense, positive controls with expected large effects.

c. Altogether, I still feel the authors are, in some points of the paper and rebuttal (but not everywhere), overstating the quality of the DMS. For example, they ascribe the lack of clear patterns in the DMS to a biologically meaningful feature of SMARCB1; this would be an unexpected result given the biochemical similarity of many substitutions. They suggest plausible reasons there could truly be exquisite sensitivity to small side chain differences in SMARCB1, but the simpler explanation is technical noise.

Minor comment:

1. Is there an internal alternative start site that explains the low impact of nonsense mutations in the N-terminal region?

(Remarks on code availability)

We thank the 5 reviewers for their thoughtful questions and insights with our initial submission. We have significantly revised the manuscript and aimed to address each comment as fully as possible which we believe has significantly improved the manuscript.

(Remarks to the Author):

SMARCB1, a core subunit of the SWI/SNF complex, is biallelically inactivated in approximately 95% of Malignant Rhabdoid Tumor (MRT) cases. However, the functional impact of SMARCB1 point mutations remains poorly understood, as they are rare outside of a few cancer types and syndromes like Coffin-Siris syndrome.

In this manuscript, Cooper et al. employ deep mutational scanning (DMS) in rhabdoid cells to systematically investigate the functional consequences of SMARCB1 point mutations. Their mutagenesis study identifies critical domains and residues essential for SMARCB1 activity while reinforcing its role in SWI/SNF complex stabilization. Importantly, this work challenges the conventional view that complete loss of SMARCB1 is necessary for disease pathogenesis, suggesting that certain missense mutations may also contribute to oncogenic dysfunction.

These findings hold significant clinical and research implications, offering new insights into SMARCB1 biology and potential diagnostic or therapeutic considerations. While the manuscript is well-constructed and compelling, the following points should be addressed to strengthen its impact and clarity:

Major Comments:

1. Limitations of the proliferation assay for R377H in Coffin-Siris syndrome (CSS)

- o The conclusion that "predicted pathogenicity does not correlate with patient frequency" may be overstated. Since CSS arises from developmental defects rather than hyperproliferation, the lack of correlation could reflect negative selection during embryogenesis—only viable mutations lead to live births with CSS.
- o Additionally, CSS patients with SMARCB1 mutations often exhibit severe clinical outcomes, suggesting these variants are indeed pathogenic. Thus, the discrepancy between CADD predictions and patient frequency might stem from an inappropriate phenotypic readout (proliferation) rather than false-positive predictions.

We thank the reviewer for this important comment which is shared with other reviewers below. We agree that the proliferation assay does not adequately capture the pathogenicity of R377H in the context of Coffin-Siris syndrome. As the reviewer notes, the pathogenicity in CSS patients with *SMARCB1* mutations is related to developmental defects without a clear link to a proliferative phenotype.

In response to this, we have removed the analysis comparing computational predictions to patient mutation frequencies. We have clarified the text to emphasize that our assay specifically measures antiproliferative function of SMARCB1 and acknowledge that different phenotypic readouts may be required to assess pathogenicity in developmental contexts. This revised manuscript acknowledges the limitations of the phenotypic readout while maintaining the rationale for DMS: computational predictors provide general pathogenicity scores, but context-specific functional assessment is required to accurately capture effects on specific phenotypes such as proliferation.

Specifically, in the revised text lines 81-91:

“The second strategy employs computational models that integrate evolutionary conservation, protein structure, physicochemical properties, and functional domain annotations to predict pathogenicity. Despite these advances, computational models prioritize general pathogenicity prediction over specific phenotypic modeling¹⁴, such as impacts on protein stability, enzymatic activity, or chromatin remodeling. The third strategy employs experimental assays to directly measure the functional consequences of variants on defined cellular phenotypes, including proliferation, metabolism, or pathway-specific activity. While highly informative, such approaches are technically challenging and resource-intensive, limiting their application to rare mutations¹⁵. Deep mutational scanning (DMS) addresses these limitations by systematically evaluating the function of variants in a gene within a defined phenotypic context.”

And in lines 96-99:

“While recognizing that mutations in chromatin remodeling genes may contribute to tumorigenesis through multiple mechanisms, we employ proliferation as a quantifiable and clinically relevant proxy for malignant potential.”

And in lines 128-140:

“To investigate the recurrent R377H variant, we re-expressed the corresponding silent mutant, R377R, the missense mutant, R377H, and the nonsense mutant R377* variants in G401 cells and compared proliferation upon induction (**Extended Data Fig 1d, Supplementary Table 3**). Compared to the R377R control ($74.5\% \pm 6.8\%$ growth suppression), both R377H and R377* showed reduced antiproliferative function with $62.0\% \pm 10.3\%$ and $62.0\% \pm 4.0\%$ suppression, respectively (**Extended Data Fig 1e-f**). While both variants showed a reduction in antiproliferative activity, both retained substantially more activity than the ATRT-derived W281* nonsense mutation ($28.7\% \pm 4.2\%$). These results demonstrate that SMARCB1 antiproliferative function can be disrupted to different degrees, with R377H showing only modest functional impairment. These findings may be consistent with clinical observations that individuals with Coffin-Siris syndrome harboring germline R377H mutations (ClinVar Accession: VCV000030203.13) do not develop cancers at higher rates than the population despite having other phenotypic effects²².”

And in the discussion [lines 532-537]:

“While our proliferation-based assay provides insights into SMARCB1's antiproliferative function, we acknowledge this readout may not fully capture pathogenicity in developmental contexts such as Coffin-Siris syndrome, where SMARCB1 mutations primarily affect developmental processes rather than proliferation control. This limitation underscores the importance of selecting functional readouts appropriate to the specific disease mechanism and biological context of interest.”

2. Quantification of SMARCB1 protein levels for W281P and I315R

o The Western blot in Fig. 3a suggests reduced protein levels for these mutants compared to WT. To strengthen the claim that they functionally mimic nonsense mutations, the authors should quantify SMARCB1 protein levels (e.g., via densitometry normalized to a loading control).

We thank the reviewer for this suggestion. We quantified SMARCB1 protein levels by densitometry across three biological replicates per genotype, normalized to β -actin as a loading control. This analysis confirms that W281P and I315R mutants exhibit reduced SMARCB1 protein levels relative to WT or synonymous controls (mean SMARCB1/ β -actin ratio: WT = 0.87, W281P = 0.32, W281* = 0.026; I315I = 1.31, I315R = 0.80, I315* = 0.10), despite being expressed under identical induction conditions. While protein levels for W281P and I315R are higher than their corresponding nonsense mutants, they are still reduced compared to WT. Prior studies which have looked at murine tissues (e.g., brain, kidney, liver, heart, lung), MEFs and ES cells heterozygous for *Smarcb1* had roughly equivalent mRNA and protein levels compared to the wild type tissue and cells¹.

While we cannot definitively separate the contributions of reduced protein stability versus functional impairment, the similar phenotypic severity observed across mutants with different protein levels (W281P:

36.6% of WT, I315R: 60.8% of I315I silent) suggests that functional defects are likely the primary driver of the observed effects.

We have included these data in the revised manuscript in **Extended Fig 6b** and **Supplementary Table 16**, with a corresponding description in the Results section [lines 294-295].

“Quantification of the input revealed decreased SMARCB1 protein abundance in the missense mutant extracts compared to wild-type (**Extended Data Fig 6b**, **Supplementary Table 16**).”

We have also added in lines 434-437:

“Both W281P and I315R mutants exhibited SMARCB1 transcript levels comparable to wildtype (**Extended Data Fig 7b**), indicating that the reduced SMARCB1 protein levels observed in nuclear extracts (**Extended Data Fig 6b**) are driven by post-transcriptional effects.”

3. Evidence for SWI/SNF complex destabilization by W281P and I315R

o The assertion that these mutations disrupt SWI/SNF assembly/stability relies on proteomics data. To validate this, the authors should perform density sedimentation assays (e.g., sucrose gradient centrifugation) to assess complex integrity.

Confirm via immunoblotting SWI/SNF subunits in mutant vs. WT cells.

We thank the reviewer for this suggestion and agree that an orthogonal method to validate our findings of SWI/SNF stability disruption would strengthen the manuscript. We have performed glycerol gradient sedimentation assays followed by immunoblotting using nuclear protein extracts from G401 cells inducibly expressing SMARCB1 WT, W281P, or I315R after 48 hours of induction (**Fig 4d**, **Extended Fig 4e**, **Methods**).

Fig 4D

Extended Fig 4E

Glycerol gradient fractionation followed by immunoblotting for ARID1A and SMARCA4 revealed additional evidence of complex destabilization in both mutants. In WT cells, both ARID1A and SMARCA4 sedimented primarily in heavier fractions (13-16), consistent with intact, high-molecular-weight SWI/SNF complexes. In contrast, both W281P and I315R mutants showed a leftward shift toward lighter fractions (fractions 7-11), indicating formation of smaller, less stable complexes or subcomplexes. Further we observe reduced signal of ARID1A in the mutant complexes.

To quantify these differences, we measured SMARCA4 signal intensity relative to the maximum across fractions 6-21 and plotted normalized values for both biological replicates (Fig 4f). This analysis further confirmed the leftward shift in SMARCA4 sedimentation profile for both mutants compared to WT, with W281P showing the most pronounced leftward shift consistent with our and other prior studies²⁻⁴ that this leftward shift is associated with impaired complex formation. These density sedimentation data provide further validation of our proteomics findings and demonstrate that the tested *SMARCB1* mutations compromise SWI/SNF complex assembly and stability.

These findings have been incorporated into the revised figures and text lines 316-327: “To orthogonally validate the mass spectrometry findings, we performed glycerol gradient sedimentation assays to assess SWI/SNF complex assembly in cells re-expressing wildtype *SMARCB1* or the W281P and I315R mutants (Fig 3e and 3f, Extended Fig 6g, Supplementary Table 22). Upon re-expression of wildtype *SMARCB1*, *SMARCA4* signal peaked in fractions 14–15. This pattern was altered by both mutants: W281P shifted the peak to fraction 13, while I315R distributed signal across fractions 13–14. Both mutants also increased relative *SMARCA4* abundance in lower molecular weight fractions (6-12) compared to wildtype (3% ± 1%), with W281P showing 19.3% ± 5.2% and I315R showing 12.3% ± 0.3%, consistent with the formation of smaller complexes. Additionally, *ARID1A* signal was reduced across fractions 13-16 in both mutants compared to wildtype. Together, these changes demonstrate that W281P and I315R disrupt SWI/SNF complex integrity and provide independent validation of our mass spectrometry findings.”

4. Impact of W281P/I315R on SWI/SNF targeting

- o Do these mutations alter chromatin occupancy of the SWI/SNF complex?
- o Cut&Run or ChIP-seq for *ARID1A*/*SMARCA4* in W281P/I315R vs. WT could clarify whether genomic targeting is affected.

Yes, both W281P and I315R mutations dramatically alter chromatin occupancy of the SWI/SNF complex based on our findings below. We have optimized conditions to perform CUT&RUN for three key SWI/SNF subunits (*SMARCA4*, *SMARCB1*, and *SMARCE1*) in two biological replicates to comprehensively assess complex targeting (Fig 4d; Methods).

Key findings:

1. **Global chromatin occupancy reduction:** Both W281P and I315R mutations caused widespread decreases in SMARCA4, SMARCB1, and SMARCE1 chromatin binding, directly explaining the accessibility defects observed in ATAC-seq.
2. **Missense mutations can be more disruptive than nonsense:** SMARCE1 showed more severe chromatin occupancy loss in both missense mutants than in nonsense controls, indicating that missense mutations can be more disruptive to subunit targeting than complete protein absence.
3. **Distinct disruption patterns between mutations:** W281P caused global reduction across all regions, while I315R showed selective effects - preserving occupancy at persistent regions but severely reducing binding at gained regions.
4. **Complex-specific disruption explains genomic patterns:** Selective loss of cBAF subunits (which target enhancers) while retaining PBAF subunits (which target promoters) may explain why promoter-enriched persistent sites are maintained while enhancer-enriched gained sites are lost. These CUT&RUN results more fully explain the functional chromatin remodeling defects observed by our ATAC-seq data we initially presented.

We have incorporated these changes into the text at lines 402-431:

“We then performed CUT&RUN of SMARCA4, SMARCB1, and SMARCE1 in wildtype and mutant conditions to link accessibility and motif patterns with SWI/SNF occupancy. We defined a comprehensive set of 16,501 SWI/SNF binding sites by merging reproducible peaks for SMARCA4, SMARCB1 and SMARCE1 in wildtype conditions (**Methods, Supplementary Table 29**). This union-based approach captures regions bound by one or more core subunits, accounting for the dynamic and transient nature of SWI/SNF binding to provide the most inclusive representation of chromatin remodeling activity. Intersection of these binding sites with our ATAC-seq peak categories yielded 12,124 sites overlapping with persistent accessible regions, 4,357 sites overlapping with gained accessible regions, and 20 sites overlapping with lost accessible regions (data not shown for lost regions).

CUT&RUN analysis across these regions revealed global reductions in chromatin occupancy across both missense mutants (**Fig 5d**), directly explaining the chromatin accessibility defects observed in ATAC-seq. All three subunits showed decreased chromatin association in W281P and I315R conditions. Strikingly, SMARCE1 showed more severe chromatin occupancy loss in both missense mutants than either nonsense control, indicating that missense mutations can be more disruptive to subunit targeting than complete protein absence. However, the two missense mutations displayed distinct patterns of disruption. W281P caused severe and global reduction in chromatin occupancy across both persistent and gained accessible regions for all three subunits, suggesting broad disruption of SWI/SNF complex targeting. In contrast, I315R showed more selective effects, with relatively preserved occupancy at persistent regions but marked reduction at gained regions for SMARCA4 and SMARCB1, while SMARCE1 remained severely depleted across both region types. This differential pattern indicates that W281P more severely compromises overall complex targeting, while I315R preferentially disrupts binding at SMARCB1-dependent regulatory sites, with both mutations exerting particularly severe effects on SMARCE1 localization. Analysis of genomic annotation distributions revealed similar patterns as between persistent and gained accessible regions (**Fig 5e, Supplementary Table 30**). Persistent SWI/SNF binding sites were enriched for promoter-TSS regions (17.9%) relative to gained sites (2.8%), while gained sites were predominantly intronic (62.9% vs. 50.9% for persistent sites) and intergenic (30.6% vs. 22.0% for persistent sites).”

5. Global chromatin accessibility changes in W281P/I315R mutants

o Fig. 4a focuses on differential peaks, but a genome-wide heatmap comparing WT vs. W281P and WT vs. I315R would better illustrate the scale and distribution of accessibility changes.

We thank the reviewer for this suggestion and agree it adds important context to the scale and distribution of accessibility changes upon SMARCB1 disruption. We have revised our prior figure as a genome-wide heatmap assessing ATAC signal at persistent peaks (defined as regions with no statistically significant accessibility changes between wildtype and nonsense conditions) and significantly gained peaks across all conditions (**Fig 4a**).

We updated our ATAC-seq analysis pipeline to identify a consensus set of 92,028 accessibility peaks observed upon wildtype SMARCB1 re-expression. Intersection with differentially accessible peaks yielded 31,863 gained peaks, 231 lost peaks, and 59,934 persistent peaks showing no significant differential accessibility. This analysis reveals that accessibility changes, while significant, affect approximately one-third of the chromatin landscape, with the majority of sites maintaining accessibility despite SMARCB1 disruption. We have also updated the motif enrichment analysis and genomic annotation for each peak set (Fig 4b-c).

The differential accessibility patterns observed upon SMARCB1 restoration reveal a sophisticated hierarchy of SWI/SNF dependence across chromatin regulatory elements, characterized by distinct spatial distributions and mechanistic underpinnings. Persistent sites that maintained accessibility despite SMARCB1 disruption and were enriched at promoter-TSS regions (16.8%) and showed strong enrichment for CTCF binding motifs (20.60%), indicating that these structural elements are maintained through SMARCB1-independent mechanisms. In stark contrast, gained peaks were predominantly located in distal regulatory regions—intronic (50.4%) and intergenic (44.4%) sequences—and demonstrated robust enrichment for AP-1 motifs (29.53%) and TEAD2 motifs (19.87%), suggesting that SMARCB1 restoration primarily rescues accessibility at enhancer loci that were selectively vulnerable to complex destabilization.

This selective vulnerability pattern reflects multiple compensatory mechanisms operating within the chromatin remodeling landscape. The SMARCB1-independent GBAF subfamily likely provides compensatory remodeling activity, particularly at CTCF sites where GBAF enrichment has been previously

demonstrated⁵. Additionally, other ATP-dependent remodeling complexes from the CHD and ISWI families, or strong transcription factor binding itself, may maintain accessibility at critical structural or promoter regions while lacking the specific cofactor interactions required for efficient AP-1 site remodeling. This mechanistic framework establishes a regulatory hierarchy where essential chromatin architecture is preserved through redundant pathways, while specialized transcriptional programs—particularly those dependent on AP-1 and TEAD2 signaling—exhibit differential vulnerability to SMARCB1 loss. The integration of spatial and sequence-specific patterns thus reveals that SWI/SNF function operates along a spectrum of dependency, with core promoter elements and architectural sites maintaining relative independence from SMARCB1, while distal enhancer elements require intact SWI/SNF complexes for proper accessibility regulation. This hierarchical organization likely ensures that fundamental cellular processes remain functional even during periods of chromatin remodeling complex disruption, while more specialized regulatory programs become selectively compromised.

We have since added the following changes to the text lines 359-400:

“To further explore the downstream consequences of the observed SWI/SNF complex instability in these mutants, we performed ATAC-seq to evaluate global changes in chromatin remodeling. Prior studies have demonstrated that re-expressing SMARCB1 in SMARCB1-deficient cell lines leads to global increases in SWI/SNF occupancy and chromatin accessibility particularly at distal regulatory regions^{2,3}. To identify robust accessibility changes associated with SMARCB1 disruption in our specific model, we established a consensus set of 92,028 accessibility peaks observed upon wildtype SMARCB1 re-expression (**Methods, Supplementary Table 27**). Analysis of differential accessibility within this consensus set revealed 31,863 gained peaks (showing increased accessibility upon SMARCB1 re-expression), 231 lost peaks (showing decreased accessibility upon SMARCB1 re-expression), and 59,934 persistent peaks (showing no significant differential accessibility upon SMARCB1 re-expression) (FDR < 0.05). Due to the small number of lost peaks, subsequent analyses focused on gained and persistent regions. These results demonstrate that approximately one-third of accessible chromatin sites are SMARCB1-dependent, while the majority maintain accessibility independent of SMARCB1 function. We then analyzed ATAC-seq signal intensity for wildtype, missense, and nonsense mutants across these genomic regions. Chromatin accessibility was markedly reduced at gained peaks in the presence of W281P and I315R mutations, highlighting the impact of these missense mutations on the remodeling function of the SWI/SNF complex at SMARCB1-dependent regions (**Fig 5a**).

Analysis of genomic annotation distributions revealed distinct spatial patterns between persistent and gained accessibility sites (**Fig 5b, Supplementary Table 28**). Persistent sites were enriched at promoter-TSS regions (16.8%) relative to gained sites (2.0%), consistent with SMARCB1-independent chromatin accessibility being concentrated at core promoters. In contrast, SMARCB1-dependent gained sites were more frequently located in intronic (50.4%) and intergenic (44.4%) regions compared to persistent sites (41.7% and 33.3%, respectively). Although persistent sites were also largely intronic and intergenic, their relative enrichment at promoters indicate that a subset of essential transcriptional programs remain accessible independent of SMARCB1. Thus, SMARCB1 appears dispensable for core promoter accessibility but essential for establishing and maintaining distal regulatory regions.

Motif enrichment analysis revealed the sequence features underlying these genomic distributions and their differential sensitivity to SMARCB1 disruption (**Fig 5c**). Persistent peaks were primarily enriched for CTCF binding motifs (20.6%), with AP-1 motifs present at lower frequency (10.1%). In contrast, gained peaks showed strong enrichment for AP-1 (29.5%) and TEAD2 (19.9%), with AP-1 sites being the most predominantly affected by SMARCB1 disruption, consistent with previous reports^{2,29,42}. Analysis of motif density across all accessible peaks in each condition further supported these trends (**Extended Data Fig 7a**). Although both persistent and gained peaks are largely intergenic or intronic, they are enriched for distinct distal motifs, with CTCF dominating persistent sites and AP-1 dominating gained sites. Together these data show that AP-1-enriched distal regulatory regions are preferentially lost upon SMARCB1 disruption, whereas CTCF-enriched structural elements are predominantly retained.”

Minor Comments:

- Line 177: The linked heatmap fails to load ("data too large"). Please provide an accessible version.

We thank the reviewer for pointing this out. The heatmap file exceeds GitHub's preview size limit and cannot be rendered directly in the browser. However, the file can still be downloaded and viewed locally. We have renamed the file to indicate to readers to download the HTML file.

We have since added the following to the README.md on the github page:
To obtain an interactive heatmap of DMS results, download the DMS_heatmap_download_to_open.html file in the parent directory.

- Figure consolidation: Some panels are redundant (e.g., Fig. 3c and 3d convey similar data, with 3d being clearer). Condensing such repetitions would improve readability.

We thank the reviewer for this helpful suggestion. We agree that Figures 3c and 3d present overlapping information and that Fig 3d offers a clearer visualization. To improve readability, we have consolidated these panels by removing Fig 3c and enhancing the legend of Fig 3d (now **revised Fig 3c**) to fully capture the relevant data. This revision streamlines the presentation without loss of important details.

Reviewer #2 (Remarks to the Author):

This manuscript systematically assesses the effects of point mutations to the SMARCB1 tumor suppressor by deep mutational scanning. The study finds that groups of residues that are clustered in three-dimensional space have destabilising effects on SMARCB1 activity that are comparable to truncating mutations. This suggests that a proportion of missense SMARCB1 mutations are likely to be oncogenic. This provides deeper insight as to how SMARCB1 functions in chromatin and cancer biology.

The following questions should be addressed:

1. The effects of SMARCB1 mutations were assessed using an expression system in a SMARCB1 deficient cell lines. If this system fully recapitulates the oncogenic effects of SMRCAB1 mutations, then it would be anticipated to reflect the lower depth patient mutation data. Looking at the data for truncating mutations, this does not appear to be the case. Missense mutations are broadly distributed across the gene in the patient data (Figure 1b) but depleted from the N-terminal 50 aa in the re-expression assay (Figure 1f). As a result, the authors should discuss the potential limitation that the three regions of enhances mutation meeting the two standard deviation cutoff, are specific to the re-expression system and may not function similarly in humans. In some cases text such as “we show that missense mutations in the RPT2 domain of SMARCB1 disrupt tumor suppressor function” should be modified to remove tumor suppressor function as this has not been tested directly.

We thank the reviewer for this comment and for highlighting important considerations regarding the interpretation and translational relevance of findings derived from our *in vitro* expression system. We agree that a more nuanced discussion of these aspects is needed.

Regarding the observed discrepancies between patient mutation data and our re-expression assay results, specifically the depletion of N-terminal missense mutations, we acknowledge that our *in vitro* system in SMARCB1-deficient cell lines, may not fully recapitulate the complex cellular environment and selective pressures present in human cancers. The broad distribution of missense mutations in patient data (**revised Figure 1a**) reflects a combination of diverse tumor types, varying genetic backgrounds, and potentially late-stage passenger mutations or mutations with subtle fitness effects not captured by our specific proliferation-based assay.

The presence of mutations in the N-terminal region in patient data (**revised Figure 1a**), despite their depletion in our assay (**revised Figure 1e**), raises several important considerations:

1. **Context-dependence:** The functional consequences of these mutations may vary depending on cellular environment, tissue type, or co-occurring genetic alterations present in patients, which are not fully captured by our cell lines (G401 - rhabdoid tumor of the kidney, BT16 - atypical teratoid rhabdoid tumor, CLF-PEDS9001T - renal medullary carcinoma) which have disruption of *SMARCB1*.
2. **Passenger mutations:** Some mutations observed in tumors could be passenger variants without strong oncogenic effects or with subtle impacts that fall below the detection threshold of our proliferation-based assay.
3. **Functional scope limitations:** The depletion of N-terminal mutations in our assay suggests these variants do not impair the specific cellular functions required for the suppression of proliferation in our models. It remains possible that these mutations affect other aspects of *SMARCB1* biology, such as developmental roles or molecular interactions not assessed by our assay.
4. We have added this discussion to the revised manuscript to clarify the scope and limitations of our assay in the context of patient-derived mutation patterns.
5. **Addressing the limitation that findings may be specific to the re-expression system:** We agree that the functional insights derived from our system, particularly the identification of specific regions exhibiting enhanced mutation frequency above the two standard deviation cutoff, should be interpreted cautiously regarding their direct extrapolation to human physiology. We have revised the text to state that our findings are derived from an *in vitro* system with specific selective pressures and may not fully reflect the complexities of *in vivo* tumor biology.
6. **Modification of causal language regarding 'tumor suppressor function':** We agree that use of 'disrupting tumor suppressor function' is an overstatement. We have revised instances of “tumor suppressor” in regards to discussing the DMS results the manuscript.

The following modifications have been made to the text to ensure our language accurately reflects the direct evidence provided by our experimental system in lines 96-99:

“While recognizing that mutations in chromatin remodeling genes may contribute to tumorigenesis through multiple mechanisms, we employ proliferation as a quantifiable and clinically relevant proxy for malignant potential.”

We have also added these limitations to the discussion [lines 532-537]:

“While our proliferation-based assay provides insights into SMARCB1's antiproliferative function, we acknowledge this readout may not fully capture pathogenicity in developmental contexts such as Coffin-Siris syndrome, where SMARCB1 mutations primarily affect developmental processes rather than proliferation control. This limitation underscores the importance of selecting functional readouts appropriate to the specific disease mechanism and biological context of interest.”

We believe these revisions will strengthen the manuscript's interpretation and provide a more balanced perspective on the implications of our findings.

2. Changes in SMARCB1 associated proteins are plotted as meeting a t-test difference (Figure 3b). It would be also useful to show this data as fold-changes in the abundance of subunits within complexes. If the change in association is less than 50%, then a substantial proportion of complexes including the SMARCB1 mutants remain intact.

We thank the reviewer for this helpful suggestion. In our original Figure 3b, we plotted the t-test difference output from Perseus, which represents the difference in the \log_2 -transformed mean intensities between groups. While this reflects statistical differences, we agree that presenting fold-change values directly enhances biological interpretability.

To address this, we have generated a new figure (now included as revised **Extended Fig 6c**) showing the absolute fold-changes in abundance of representative subunits from the BAF complex, grouped by their association with common, canonical (cBAF), or PBAF subcomplexes. These values were derived by calculating the difference in \log_2 -transformed intensities between groups and converting to linear scale.

Importantly, this new visualization demonstrates that all subunits exhibit >50% reduction in association relative to wild-type SMARCB1. This indicates that these mutants result in substantial disruption of BAF complex integrity.

3. If the strong effects on chromatin accessibility are compared to the proportion of complexes remaining substantially intact, is it possible that mutations primarily affect the activity of complexes rather than acting by destabilisation a combination of the two, or destabilisation only as suggested in the text?

We thank the reviewer for this important comment regarding the mechanism of disruption. CUT&RUN analysis demonstrates substantial loss of chromatin occupancy for all SWI/SNF subunits at regions with altered accessibility in both mutants (revised **Fig. 5d**). Notably, SMARCE1 shows more severe occupancy loss in missense mutants than nonsense controls across all presented genomic regions, indicating that these mutations actively disrupt subunit targeting beyond complete loss.

Furthermore, our biochemical analysis reveals hierarchical vulnerability across SWI/SNF subfamilies. Both mutants cause subfamily-specific disruption of all cBAF complexes (loss of DPF2, ARID1A/B), while PBAF complexes are only partially affected, with PBRM1 dissociating but other subfamily-specific subunits (BRD7, ARID2, PHF10) remaining intact (**Fig. 3d**). We also observe destabilization of shared initial BAF core subunits (SMARCC1, SMARCD2, SMARCD3), suggesting the mutant SMARCB1 creates post-assembly structural instability that propagates differentially across complex subtypes. This differential complex disruption provides a mechanistic framework for the observed chromatin accessibility patterns: the preferential vulnerability of cBAF complexes, which typically occupy enhancers and distal regulatory elements, underlies the severe accessibility loss at AP-1 enriched distal regulatory sites. While retention of key PBAF-specific subunits might initially suggest residual PBAF function maintains promoter accessibility, the near-complete loss of SMARCE1 occupancy and minimal promoter response to SMARCB1 restoration indicate that promoter regions are inherently less dependent on SWI/SNF activity than enhancer elements. The parallel between biochemical complex disruption and loss of chromatin occupancy provides strong evidence that these SMARCB1 missense mutations drive functional defects through destabilization rather than reduced activity of assembled complexes.

We have since added the following to the discussion 474-498:

“To elucidate the mechanistic basis for these functional defects, we employed complementary biochemical and chromatin profiling approaches. Chromatin binding assays revealed substantial loss of SWI/SNF subunit occupancy at regions with altered accessibility in both mutants, with SMARCE1 showing more severe occupancy loss in missense mutants than nonsense controls. These data, combined with biochemical analysis, reveal a hierarchy of SWI/SNF complex vulnerability that extends beyond the sequential assembly model previously described⁴³. While cBAF complexes showed complete subfamily-specific disruption (loss of DPF2, ARID1A/B), PBAF complexes were only partially affected, with PBRM1 dissociating while other subfamily-specific subunits (BRD7, ARID2, PHF10) remained unaffected. Subunits specific to GBAF complexes were entirely unaffected, consistent with these complexes naturally lacking SMARCB1. Importantly, we observed destabilization of shared initial BAF core subunits (SMARCC1, SMARCD2, SMARCD3), suggesting the mutant SMARCB1 creates post-assembly structural instability that propagates differentially across complex subtypes.

This differential complex disruption provides a framework for understanding the observed chromatin accessibility patterns. The preferential vulnerability of cBAF complexes, which typically occupy enhancers and distal regulatory elements⁴⁴, likely underlies the severe accessibility loss at AP-1 enriched distal regulatory sites. The retention of key PBAF-specific subunits, combined with PBAF's preferential binding to promoters⁴⁵, might initially suggest that residual PBAF function maintains promoter accessibility. However, the near-complete loss of SMARCE1 occupancy and minimal promoter response to SMARCB1 restoration indicate that promoter regions are inherently less dependent on SWI/SNF activity than enhancer elements. The parallel between biochemical complex disruption and loss of chromatin occupancy provides strong evidence that the two presented SMARCB1 missense mutations drive functional defects through destabilization rather than reduced activity of assembled complexes.”

4. Supporting a lack of involvement in cancer the manuscript states “individuals with Coffin-Siris syndrome harboring germline mutations in the C 133 terminal CCD domain of SMARCB1, and specifically R377H (ClinVar Accession: VCV000030203.13), do not develop cancers at higher rates than the population” However, the fact this alteration is detected in patients with Coffin-Siris syndrome indicates that it is likely to be functional.

We thank the reviewer for this important clarification. We fully agree that the presence of the R377H variant in individuals with Coffin-Siris syndrome strongly supports its pathogenicity in a developmental context. We now recognize that we needed to provide further context to our initial comments and to highlight that pathogenicity is context-dependent. Specifically, variants may exhibit pathogenic effects in developmental processes (such as neurodevelopment in CSS) while having minimal impact on cancer-relevant pathways (such as proliferation control), or vice versa.

While R377H is clearly pathogenic during neurodevelopment, our proliferation assays in SMARCB1-deficient cancer cells demonstrate that R377H exhibits significantly less antiproliferative impairment compared to the W281P and I315R mutants studied here, suggesting different functional consequences in cancer-relevant pathways. Thus, the variant's role in disease depends upon the type of pathogenicity being studied: it is pathogenic in CSS but may not contribute to cancer development via proliferation pathways. We thank the reviewer and other reviewers for highlighting this gap in our initial manuscript.

We have revised the manuscript to clarify this distinction, emphasizing the importance of context-specific functional assessment of SMARCB1 variants (lines 128-140):

“To investigate the recurrent R377H variant, we re-expressed the corresponding silent mutant, R377R, the missense mutant, R377H, and the nonsense mutant R377* variants in G401 cells and compared proliferation upon induction (**Extended Data Fig 1d, Supplementary Table 3**). Compared to the R377R control ($74.5\% \pm 6.8\%$ growth suppression), both R377H and R377* showed reduced antiproliferative function with $62.0\% \pm 10.3\%$ and $62.0\% \pm 4.0\%$ suppression, respectively (**Extended Data Fig 1e-f**). While both variants showed a reduction in antiproliferative activity, both retained substantially more activity than the ATRT-derived W281* nonsense mutation ($28.7\% \pm 4.2\%$). These results demonstrate that SMARCB1 antiproliferative function can be disrupted to different degrees, with R377H showing only modest functional impairment. These findings may be consistent with clinical observations that individuals with Coffin-Siris syndrome harboring germline R377H mutations (ClinVar Accession: VCV000030203.13) do not develop cancers at higher rates than the population despite having other phenotypic effects²².”

And in the discussion [lines 532-537]:

“While our proliferation-based assay provides insights into SMARCB1's antiproliferative function, we acknowledge this readout may not fully capture pathogenicity in developmental contexts such as Coffin-Siris syndrome, where SMARCB1 mutations primarily affect developmental processes rather than proliferation control. This limitation underscores the importance of selecting functional readouts appropriate to the specific disease mechanism and biological context of interest.”

Reviewer #3 (Remarks to the Author):

The tumor suppressor gene, SMARCB1, is biallelically inactivated through mutagenesis in a few pediatric malignancies, yet the pathologic significance of missense mutations remain unclear particularly given the lack of hotspot variants. In this manuscript, the authors carry out a deep mutational scan (DMS) of SMARCB1 in order to identify specific missense mutations that can disrupt SMARCB1 tumor suppressor function. The authors screened for fitness effects conferred upon expression of a DMS orf library of SMARCB1 variants in 3 separate SMARCB1-deficient cancer models. Despite identifying a few different “clusters” of candidate loss of function missense mutations (ie. WHD domain, IDR and RPT2), the authors focused their more extensive experimental follow up to only 2 variants that mapped to the RPT2 domain.

Overall, the data generally supported that the 2 residues they chose to experimentally model resulted in a disruption of SWI/SNF activity in this context. However, the interpretation was limited by the fact that one engineered residue (W to a proline) would likely be expected to cause a significant disruption to protein structure/function and the other variant would likely not occur in nature due to codon usage (Ile>Arg, would require multiple mutations). The engineered non-sense mutations at these residues would be expected to be detrimental. Hence, it still remains a question regarding the pathologic relevance of the RPT2 domain variants in cancer. The paper could have benefited by a more rigorous assessment of additional RPT2 variants, with a consideration for naturally occurring variants in this domain.

We thank the reviewer for their thoughtful comments. Our initial focus was validating top hits from our screen based on the average Z residues (**Fig 2d**) where we have since added additional epigenomic and structural analyses of these mutants based on comments from the reviewers.

To address the concerns of naturally occurring variants, we re-analyzed the AACR-GENIE dataset and we identified naturally occurring mutants in RPT2 with a z-score > 2 (**Extended Data Fig 5b**).

To validate our findings with a naturally occurring variant, we tested W281R (z-score = 2.06), which complements our previous analysis of the W281P mutant and patient-derived W281* ATRT variant. W281R showed robust protein expression (**Extended Fig 5c**) but demonstrated impaired antiproliferative function in crystal violet and trypan blue exclusion assays ($15.34\% \pm 16.38\%$, **Extended Fig 5d-e**). These data confirm that naturally occurring SMARCB1 missense variants can disrupt antiproliferative function.

We have added these findings in the revised manuscript (lines 280-282):

“Further, W281R, a variant seen in a patient with ovarian carcinosarcoma¹³, which had a z-score of 2.06, also showed impaired antiproliferative function ($15.34\% \pm 16.38\%$, **Extended Fig 5b**).”

Comments:

(1) For the initial assessment of missense mutations across cancer (Fig. 1), can the authors provide any information on zygosity/LOH regarding missense mutations given that biallelic loss is necessary. (I realize the data is limited, but it seems important to understand if any of the missense mutations found in cancer are homozygous, given that this genetic state is modeled in the paper).

We thank the reviewer for asking for this clarification. We have since added VAFs for each patient mutation in **Supplementary Table 2** for data within GENIE. VAFs for patient mutations are not provided from COSMIC datasets. Of the missense mutations in GENIE, we observed 74 with a VAF > 0.5. While VAF alone cannot definitively distinguish between homozygous mutations, hemizygous mutations with LOH, or mutations in regions of copy number gain, we observed 5 missense mutations with a VAF between 0.9-0.95 (L19Q, R377H, P165S, G29S, F168L) which are consistent with biallelic alteration. Notably none of these mutations reached the LOF threshold of z-score > 2 within our DMS data (**Extended Data Fig 5b**). Future studies systematically examining zygosity and LOH status across SMARCB1 missense mutations will be important for understanding their functional consequences in different genetic contexts.

(2) In building their rationale for carrying out this assessment, the authors initially test the R377H and R377* variants for their ability to confer tumor suppressor function. They state that “the R377* mutation partially compromises tumor suppressor function, while the R377H mutation retains its function”. This statement is not supported by the data (extended figure 1b, 1c), as both mutants perform almost identically in suppressing cell growth.

We thank the reviewer for this important observation. We agree that our prior wording overstated the functional difference between R377H and R377*. Indeed both R377H and R377* show modest reductions compared to R377R control; however, both retain substantially more antiproliferative activity than the patient-derived ATRT nonsense mutation W281* (**Extended Fig 1a-c**). These conclusions are supported by robust experimental replication (n=3 crystal violet assays, n=4 trypan blue counts).

The revised text (lines 121-140):

“To experimentally interrogate the antiproliferative function of SMARCB1 variants, we used a previously described inducible SMARCB1 re-expression system in the SMARCB1-deficient G401 cell line²⁰. We defined functional reference points by comparing wildtype SMARCB1 (65.5% ± 13.1% growth suppression) against a patient-derived ATRT nonsense mutation²¹, W281* (28.7% ± 4.2% growth suppression), establishing upper and lower bounds of antiproliferative activity (**Extended Data Fig 1a-c**).

To investigate the recurrent R377H variant, we re-expressed the corresponding silent mutant, R377R, the missense mutant, R377H, and the nonsense mutant R377* variants in G401 cells and compared proliferation upon induction (**Extended Data Fig 1d, Supplementary Table 3**). Compared to the R377R control (74.5% ± 6.8% growth suppression), both R377H and R377* showed reduced antiproliferative function with 62.0% ± 10.3% and 62.0% ± 4.0% suppression, respectively (**Extended Data Fig 1e-f**). While both variants showed a reduction in antiproliferative activity, both retained substantially more activity than the ATRT-derived W281* nonsense mutation (28.7% ± 4.2%). These results demonstrate that SMARCB1 antiproliferative function can be disrupted to different degrees, with R377H showing only modest functional impairment. These findings may be consistent with clinical observations that individuals with Coffin-Siris syndrome harboring germline R377H mutations (ClinVar Accession: VCV000030203.13) do not develop cancers at higher rates than the population despite having other phenotypic effects²².”

(3) In further support of their rationale for the screen, the authors make the point that the computational algorithms predict for a large proportion of missense mutations being pathogenic based upon a threshold value (?) for each algorithm and this does not correlate with observed patient frequencies. Nor there a statistically-significant correlation when the computational algorithm scores are treated as a continuous variable (Ext. Fig 1h). Although this is a minor point, if one selects a more stringent predictive score (Ext. Fig 1h), there is a stronger likelihood of correlating with higher mutation counts (although not completely predictive). The authors screening data in Fig 1g further supports the greater likelihood for a pathogenic variant if a higher CADD score was selected in this case. Just a minor point, as the authors rationale for testing variants still holds true, but there is a lot of “build-up” to get to the rationale for running the screen that is probably not critical for the paper.

We thank the reviewer for this suggestion. We agree that the prior discussion of patient mutation frequencies may have added unnecessary “build-up” to the rationale for the screen. In response to this comment in addition to others, we have removed references to patient mutation frequencies and reoriented the section to focus on the concordance between computational predictions and our functional measurements. Specifically, we now highlight that CADD, AlphaMissense, and REVEL overestimate the impact of R377H on antiproliferative function: although R377H ranks in the 90th, 82nd, and 97th percentiles, respectively, it exhibits an intermediate phenotype in our assays. This revision streamlines the text while retaining the key rationale for performing DMS: computational predictors provide general pathogenicity scores, but context-specific functional assessment is necessary to accurately capture effects on proliferation.

The following changes have been made to the text [lines 145-149]:

“These tools classified 96.5%, 76.5%, and 37.8% of missense variants as deleterious, respectively (**Fig 1b, Extended Data Fig 1j-k, Methods**). However, predictions for R377H did not align with our experimental observations. R377H ranked in the 90th, 82nd, and 97th percentiles for AlphaMissense, REVEL, and CADD, respectively, yet exhibited only intermediate functional impairment in our proliferation assays.”

Furthermore we have highlighted where R377H falls on across the full range oof predictions for CADD, AlphaMissense, and REVEL and in **revised Fig 1b, Extended Fig 1j and Extended Fig 1k**, respectively, shown below.

(4) Fig 2d is one of the most critical pieces of data in the paper, as it takes into account all 19 missense mutations at the given position. The accompanying table only provides the averaged values. Given that the authors highlight only a handful of potential LOF variants, it would be informative for the reader to visualize the z-score enrichments across all variants at these positions to ensure that the data is not being skewed by specific variants (particularly since the data is averaged).

We thank the reviewer for this insightful comment and agree that presenting individual variant z-scores would enhance the transparency. In response, we have updated **Supplementary Table 11** to include z-scores for each of the 19 possible missense mutations at every position shown in the revised Fig. 2d.

Additionally, revised **Extended Fig 2d** displays the z-scores for all substitutions at each position presented in revised **Fig 2d**, enabling readers to assess the full range of effects and verify that averaged values are not driven solely by outliers.

(5) In general, the crystal violet assays to assess growth are noisy and there are no replicates. Although trypan blue cell counts were also collected, the overall interpretation was not fully clear for some variants. For instance, expression of the D227K mutation had no observable effect on growth in the crystal violet assay (Fig. 2h), yet still results in an appreciable inhibition of growth in the cell counts (Fig 2i). (similarly for E300K) Furthermore, interpretation of the corresponding nonsense mutations should not be made given that these mutations were not expressed to the same degree (as may be expected given NMD).

We thank the reviewer for their comments regarding the proliferation assays. We agree that crystal violet-based growth assays can be noisy and are less quantitative than direct cell counts. As the reviewer notes, there are some discrepancies between the two assays (e.g., D227K and E300K), which we believe reflect differences in how the assays capture cell proliferation. Specifically, the trypan blue exclusion assays involve two subsequent replatings of cells and therefore reflect proliferation defects over an extended period, whereas the crystal violet assay measures differences across one single plating. Thus, trypan blue exclusion assays tend to highlight cumulative growth defects more strongly than crystal violet. We have included both assays to provide orthogonal evidence for antiproliferative phenotypes: the crystal violet assay offers a more visually distinct readout of growth differences, whereas the trypan blue exclusion assay provides a more quantitative assessment across multiple replatings.

To address reproducibility in the crystal violet assays, we include in the manuscript 3 replicates for crystal violet experiments in addition to what was initially presented (n=4) in **Extended Fig 5a**. These replicate experiments confirm the observed trends for the missense variants tested.

We have also incorporated replicates (n=3) for the R377 mutants in **Extended Fig 1f**.

And additionally we have added replicates (n=3) for the added patient mutant W281P in revised **Extended Fig. 5d**.

As suggested by the reviewer, we have also removed interpretation of nonsense variants in their comparison to their respective missense variants, as their expression levels are reduced (likely due to nonsense-mediated decay), and thus their growth effects cannot be directly compared to those of the missense variants.

The text has been updated to reflect these changes [lines 273-285].

“Using total protein lysates, we detected protein expression in both the wildtype and missense mutants as compared to the uninduced samples (**Fig 2g**). Crystal violet staining over a 10-day period showed reduced growth suppression in cells re-expressing missense mutant constructs compared to wildtype (n=4) (**Fig 2h, Extended Fig 5a**). In a separate 8-day proliferation assay, all missense mutants exhibited impaired antiproliferative function: W281P achieved $27.4\% \pm 4.2\%$ (mean \pm SD) growth suppression, E300K achieved $49.8\% \pm 6.9\%$, and I315R achieved $6.4\% \pm 17.6\%$, while D277K showed $42.4\% \pm 9.4\%$ growth suppression that did not reach statistical significance (**Fig. 2i, Supplementary Table 15**). Further, W281R, a variant seen in a patient with ovarian carcinosarcoma13, which had a z-score of 2.06, also showed impaired antiproliferative function ($15.34\% \pm 16.38\%$, **Extended Fig 5b**). We then focused on understanding the mechanism by which missense mutants W281P and I315R, the highest-scoring variants at these two spatially adjacent residues (z-scores 5.57 and 3.83 respectively), led to impaired antiproliferative function.”

(6) In the IP-mass spec study (Fig. 3b), the authors indicate that several proteins are enriched in the mutant condition, suggesting “mutant-specific functions”. The fold enrichment and degree of significance is minimal. Without experimental follow up, I don’t think these observations require being called out in the body of the paper.

We thank the reviewer for this helpful suggestion. In light of this feedback and given the modest enrichment and lack of experimental follow-up, we have removed this section and no longer discuss these observations in the main text. We agree that this streamlines the manuscript and keeps the focus on the most strongly supported findings.

Lines 299-306 (“Among the proteins significantly enriched ... transcriptional programs or epigenetic reprogramming”) from the initial manuscript have since been removed from the body of the text.

(7) With the exception of one Western blot (Ext. data Fig 4c), there is little validation data supporting some of the points being made (lines 218-328), nor with respect to the latter commentary on QSER1 (lines 382-386).

We thank the reviewer for this constructive feedback highlighting the need for stronger validation of our mechanistic claims. In response, we have significantly revised this section of the manuscript to ensure our conclusions align more closely with the experimental evidence. Specifically, we have:

1. Rewritten the relevant section (lines 218-328) to focus on direct observations rather than mechanistic interpretations.
2. Removed commentary on QSER1 (lines 382-386) as there was insufficient validation data for these claims.
3. Replaced speculative language with more measured terminology that acknowledges the limitations of our current data.
4. Relocated mechanistic hypotheses to the Discussion section where appropriate.

These revisions better distinguish between what our data directly demonstrates versus what requires further experimental validation.

Specifically:

1. **Structural Interpretations** Following the reviewer's guidance, we have moved several structure-based hypotheses to the discussion section and clearly distinguished between published structural data and our functional observations:
 - a. **WHD Conformations:** We now frame the distinct cBAF/PBAF conformations as structure-based hypotheses derived from published cryo-EM data, with clear statements about the need for future functional validation.
 - b. **R52 DNA Interactions:** The role of R52 is now presented as a structure-guided hypothesis based on its predicted DNA major groove insertion in PBAF structures, combined with our mutation intolerance data.
2. **IDR Flexibility:** We have revised our interpretation of the IDR mutation cluster to be more conservative. Our DMS data identified a cluster of mutation-intolerant residues (E122, Q123, A125) within the intrinsically disordered region connecting SMARCB1's two globular domains. While these residues do not display obvious functional characteristics, IDRs are known to facilitate conformational flexibility for transient protein-protein interactions and complex assembly. We hypothesize that mutations in this cluster may affect SMARCB1's ability to undergo conformational changes necessary for proper SWI/SNF complex function, though direct experimental validation of this mechanism remains to be performed.

We have moved these points for structural based hypotheses into the discussion [lines 510-519]:

“Our findings across other domains reveal additional structure-function relationships that warrant further investigation. The WHD's mutational intolerance may reflect its distinct conformations in cBAF versus PBAF complexes observed in cryo-EM structures, potentially explaining complex-specific functional requirements. Within this domain, our identification of R52 as highly mutation-intolerant aligns with structural predictions of its insertion into the DNA major groove in PBAF structures, suggesting a direct role in DNA binding that could explain its functional importance. The cluster of mutation-intolerant residues within the IDR (E122, Q123, A125) suggests this region may facilitate conformational flexibility required for dynamic protein-protein interactions. These structure-based hypotheses provide promising directions for future experimental validation.”

Minor points:

- (1) No description in the paper how pathogenicity scores were defined for CADD, AlphaMissense and REVEL.

We thank the reviewer for pointing out this omission. We have added into the **Methods** on how such scores were derived. Specifically:

REVEL

We applied a REVEL score threshold of ≥ 0.75 for pathogenic classification, representing a stringent cutoff that prioritizes specificity over sensitivity. As reported by the REVEL developers⁶, this threshold captures

52.1% of known disease mutations while maintaining low false positive rates (3.3% of neutral variants and 4.1% of evolutionary structural variants misclassified as pathogenic).

CADD

We utilized CADD scaled PHRED scores (not raw scores) with a threshold of ≥ 20 to classify variants as potentially pathogenic. According to the CADD developers⁷, a scaled PHRED score of 20 or greater indicates variants in the top 1% of all possible reference genome single nucleotide variants in terms of predicted deleteriousness.

We acknowledge that the CADD authors explicitly state that arbitrary cutoff values are not recommended, as optimal thresholds depend on analysis-specific factors including phenotype severity, inheritance patterns, and available resources for variant follow-up (Rentzsch et al., 2019). The developers instead recommend ranking variants by CADD score and investigating top-ranked variants based on study-specific constraints. However, we selected the PHRED score ≥ 20 threshold as it represents a stringent cutoff identifying the most deleterious 1% of possible variants, and binary classification of variants as pathogenic versus benign remains standard practice and expectation in medical genetics research. This approach balances computational efficiency with clinical interpretability for our study design while maintaining awareness of the inherent limitations of hard cutoffs for continuous pathogenicity scores.

Alphamissense

We applied AlphaMissense scores using the standard threshold of ≥ 0.564 to classify variants as likely pathogenic. This threshold was established by the AlphaMissense developers and represents a calibrated cutoff where variants classified as likely pathogenic or likely benign achieve 90% expected precision for both classes when validated against ClinVar data⁸.

Unlike other pathogenicity prediction tools where threshold selection may be context-dependent, AlphaMissense was specifically designed and validated with universal thresholds that maintain consistent performance across genes and variant types. The 0.564 threshold provides an optimal balance between sensitivity and specificity, with the developers demonstrating that these standardized cutoffs achieve reliable classification performance in clinical datasets. The calibrated AlphaMissense scores (ranging from 0 to 1) can be interpreted as the approximate probability of a variant being clinically pathogenic, making the 0.564 threshold a statistically grounded choice rather than an arbitrary cutoff.

We have since added the following text to Methods [line 707-725]:

“Pathogenicity prediction score thresholds

We applied established thresholds from three computational pathogenicity prediction tools to classify SMARCB1 missense variants:

REVEL: Variants with scores ≥ 0.75 were classified as pathogenic. This stringent threshold captures 52.1% of known disease mutations while maintaining low false positive rates (3.3% of neutral variants and 4.1% of evolutionary structural variants misclassified as pathogenic), as reported by the REVEL developers²⁴.

CADD: We utilized scaled PHRED scores with a threshold of ≥ 20 to classify variants as potentially pathogenic, identifying variants in the top 1% of all possible reference genome single nucleotide variants in terms of predicted deleteriousness²⁵. While the CADD developers note that optimal thresholds depend on analysis-specific factors and recommend ranking variants rather than applying arbitrary cutoffs, we selected the PHRED score ≥ 20 threshold as it represents a stringent cutoff and binary classification remains standard practice in medical genetics research.

AlphaMissense: Variants with scores ≥ 0.564 were classified as likely pathogenic using the standard threshold established by the AlphaMissense developers⁵⁰. This calibrated cutoff achieves 90% expected precision for both pathogenic and benign classifications when validated against ClinVar data. The AlphaMissense score can be interpreted as the approximate probability of a variant being clinically pathogenic.”

(2) In Fig 1d, BT16 shows 2 indels, but the authors indicate in the legend that it has both a monoallelic deep deletion and a frameshift indel.

We thank the reviewer for identifying this error and inconsistency in our initial submission. We have re-reviewed our WGS data for the BT16 cell line and can confirm that BT16 does indeed harbor a biallelic 1bp frameshift indel in exon 1 of *SMARCB1* consistent with a homozygous state. We have corrected the figure legend accordingly and apologize for this oversight.

(3) Supplemental Table 6 is anonymized. It would be useful to at least annotate some control genes (non-essentials, essentials) to better understand the performance of the screen.

We thank the reviewer for this suggestion to improve interpretation of screen performance. We have updated **Supplementary Table 6** to include annotation for control genes. Given that this represents the first overexpression screen in *SMARCB1*-deficient cells, we adapted control gene categories from related screening approaches: genes previously identified as essential in deletion screens (*MDM4* and *DCAF5*)^{9,2}, housekeeping genes (*GAPDH*, *ACTB*), and known inhibitory targets in *SMARCB1*-deficient cells (*EZH2*)¹⁰. We have further updated the figure in the revised manuscript as shown below.

(4) The colors are not clear on Fig. 4e.

We have since updated revised **Fig 5e** to have more visually distinct colors between conditions.

Reviewer #4 (Remarks to the Author):

The authors conducted deep mutational scanning (DMS) of SMARCB1, evaluating 8,418 variants across three SMARCB1-deficient cancer cell line models. DMS was performed with a proliferation screen. They used a z-score >2.0 per mutation (or average Z score $>2SD$ per residue) to define top-hit loss-of-function variants for follow-up, finding 101 mutations or 13 residues above that cutoff. They integrated multiple datasets (AACR GENIE, COSMIC, gnomAD) and computational predictors (CADD, REVEL) to contextualize findings. They validate 12 sequences (including 4 missense mutations) and focus on 2 RPT2 domain mutations (W281P, I315R) that destabilize the SWI/SNF complex and impair chromatin remodeling without complete protein loss, challenging reliance on IHC for diagnostics. They interestingly were able to link structural flexibility of the winged-helix domain to these functional outcomes using molecular dynamics simulations, but did not clearly explain how the RPT2 mutations would plausibly affect flexibility of the distal WHD. They supported findings for those 2 mutants with ATAC-seq (chromatin accessibility), RNA-seq (transcriptional changes), and proteomics (SWI/SNF subunit dissociation), very exhaustively strengthening the claim that those mutations have a major impact on function.

The most impactful aspect is that they demonstrated that missense mutations can mimic nonsense mutations functionally; this is not very surprising biologically, but it does suggest current IHC diagnostic methods may miss pathogenic variants. They also highlighted discordance between mutation recurrence (e.g., R377H) and pathogenicity, emphasizing the need for these functional assays. However, it is not entirely clear these 2 deeply characterized mutations will be found in many patients or that their map can be used to confidently predict the pathogenicity of other mutations that will be found in patients. In my view, the authors appropriately largely focus their claims on the strongly supported evidence for the validated mutations, without overstating how much the DMS data can serve as a resource for predicting pathogenicity of other mutations.

I commend the authors for making their code available, including for reproducing analyses.

Major Comments

1. The work suggests SMARCB1 is surprisingly robust to mutation (e.g. compared to CADD). Is it also possible their DMS is not sensitive to measure mutation impacts? One reason I wonder is because usually proline is very effective at breaking alpha helices across positions, but even proline rarely has a large impact here. Is this simply how the data is visualized? I am concerned about a technical issue affecting the dynamic range or noise in this assay. SMARCB1 was re-expressed via inducible vectors, which may not reflect endogenous protein levels or stoichiometry within the SWI/SNF complex; could that affect sensitivity? Figure 1F heatmap looks noisy, in the sense that few residues have consistent phenotypes when mutated to related amino acids. Put another way, the map doesn't have many stripes or blocks that are often seen in DMS. What is the correlation of biological replicates of fold changes over time (Z-scores?) Correlation of counts can provide an inflated view of reproducibility. How do results compare across cell lines?

A) The work suggests SMARCB1 is surprisingly robust to mutation (e.g. compared to CADD). Is it also possible their DMS is not sensitive to measure mutation impacts?

This observed robustness is, in fact, a key finding and aligns with biological expectations for SMARCB1. SMARCB1 is generally not highly mutated across a broad spectrum of cancers. The overall low rate of somatic mutation in SMARCB1 in the general cancer population suggests an intrinsic resilience to many single amino acid changes. SMARCB1's role as a core component and scaffold within the large and dynamic SWI/SNF complex may confer a degree of functional buffering against perturbations at many positions, where the complex might tolerate certain changes without complete loss of function.

Importantly, we are confident that our Deep Mutational Scanning (DMS) assay is sufficiently sensitive to detect mutation impacts accurately. Several lines of evidence support this:

- Nonsense (stop-gain) mutations throughout the central coding region produce strong loss-of-function phenotypes, clearly visible as the dark red band labeled "Stop" in Figure 1F. This expected severe effect demonstrates the assay's ability to identify truly disruptive variants.

- Deep sequencing coverage before and after selection (334-444 reads per variant; table of mean variant coverage shown below) minimized sampling noise and enabled reliable quantification of even low-frequency variants.

	Early (reads per variant)	Late (reads per variant)
G401 R1	334.01	399.79
G401 R2	358.48	398.44
BT16 R1	356.77	351.02
BT16 R2	341.20	343.68
CCLF_PEDS9001_T1 R1	444.89	399.35
CCLF_PEDS9001_T1 R2	381.32	374.85

- Fitness scores were calculated using rigorous statistical analysis that normalizes variant enrichment/depletion relative to nearby silent mutants, producing z-scores that account for technical variability and background noise (see **Methods**).

We also note that our assay specifically measures the ability of SMARCB1 variants to inhibit cell proliferation in this cellular context. It is possible that some mutations classified as functionally neutral in our system could affect other SMARCB1-dependent processes not captured by this assay—such as differentiation, chromatin accessibility, or transcriptional regulation. However, within the scope of this proliferation-based assay, we are confident that the observed mutational tolerance reflects a true biological property, rather than a lack of assay sensitivity.

Together, these features confirm that the assay’s design and execution provide sensitivity to accurately measure the functional impact of mutations, supporting the conclusion that SMARCB1’s mutational robustness is a genuine biological property rather than a limitation of the assay.

B) “One reason I wonder is because usually proline is very effective at breaking alpha helices across positions, but even proline rarely has a large impact here. Is this simply how the data is visualized?”

We appreciate this thoughtful observation. It is true that proline is a strong helix disruptor, and its relatively modest impact at some positions initially stood out to us as well. However, this is not an artifact of data visualization — our raw fitness scores and their underlying z-score distributions accurately reflect these trends. We interpret this to suggest that not all helical regions are equally sensitive to proline-induced structural disruption. Some α -helices may not be essential for core stability or function. Instead, they may be buffered by local structural context or protein–protein interactions, thereby tolerating proline substitutions more than expected.

To more directly address this point, we analyzed z-scores for substitutions to each amino acid across all residues (revised **Extended Fig 2d**). Lysine (K), Proline (P), Aspartic Acid (D) and Arginine (R), were among the most disruptive, with mean z-scores of -0.099 , -0.11 , -0.12 , and -0.17 respectively — indicating consistently high-impact substitutions across many positions. In contrast, substitutions to methionine (M), tryptophan (W), and phenylalanine (F) were the most tolerated, with mean z-scores of -0.47 , -0.45 , and -0.38 respectively. While proline's average effect appears modest (mean z-score of -0.11), proline and lysine each showed deleterious effects (z-score > 2) at 11 positions—more than any other amino acid. Of these 11 positions, six were positioned in structural regions within RPT2; 3 in β -sheets (residues 277, 278, and 281) and 3 in α -helices (residues 300, 301, and 315).

We have since added the following text to lines 225-228:

“Proline and lysine substitutions exhibited the most pronounced deleterious effects, displaying both the highest mean z-scores and severe functional impairment (z-score > 2) at the greatest number of residues (11 each) (**Extended Fig 2d**).”

C) I am concerned about a technical issue affecting the dynamic range or noise in this assay. SMARCB1 was re-expressed via inducible vectors, which may not reflect endogenous protein levels or stoichiometry within the SWI/SNF complex; could that affect sensitivity?

We appreciate the reviewer's important and valid concern regarding the potential impact of exogenous protein expression on assay sensitivity. To clarify, SMARCB1 was re-expressed using a constitutively active EF1 α promoter in the screens presented, not an inducible system. This approach was chosen to ensure uniform and stable expression across all variants throughout the screen. The inducible system was only used in subsequent validation assays.

We fully acknowledge that exogenous expression may not precisely recapitulate endogenous levels or stoichiometry within the SWI/SNF complex. To address these concerns and maximize physiological relevance, we implemented several key design elements:

Physiologically relevant expression levels: Wild-type SMARCB1 expressed under the EF1 α promoter achieved steady-state levels after 10 days that were moderately elevated but comparable to normal kidney cell lines (Aflac_2494N) and a SMARCB1 wild type pediatric kidney cancer, Wilms tumor cell line (Aflac_2494T) (**Extended Fig 1m**). This moderate overexpression should not significantly compromise assay sensitivity given our internal silent mutation controls.

Demonstrated biological activity: We verified that the re-expressed SMARCB1 is biologically active and functionally incorporated, demonstrated by (Fig 3b, Fig 5a,d,f,g) and rescues growth defects linked to SMARCB1 loss (Fig 2h-i). This confirms that the exogenous protein effectively participates in its expected biological roles.

Consistent expression across all variants: While absolute expression may differ slightly from native levels, the critical factor for DMS is consistency across variants. All constructs were expressed from the same promoter and vector backbone, ensuring that observed differences in fitness reflect the effects of mutations, not variation in expression.

Internal controls mitigate expression artifacts: Our rolling z-score normalization compares each variant to local silent mutations, which experience identical expression conditions. This approach controls for any systematic effects of overexpression on the assay readout.

In summary, while no exogenous system can perfectly reproduce endogenous regulation, we believe our approach provides a robust and physiologically relevant context to measure the relative functional effects of SMARCB1 mutations with high sensitivity and reliability.

We have since added the following text lines 166-168:

“We observed SMARCB1 protein expression levels approximately 2.2-fold higher than endogenous levels in normal kidney tissue cell line 2494N, but comparable to levels seen in Wilms tumor cell line 2494T (1.6-fold difference) (Extended Fig 1m)26.”

D) What is the correlation of biological replicates of fold changes over time (Z-scores?) Correlation of counts can provide an inflated view of reproducibility. How do results compare across cell lines?

We assessed the reproducibility of our results by calculating the correlation of fitness scores (z-scores) between independent biological replicates in each of the three cell lines. The following subpanels have been added to revised Extended Fig 2a, showing the coefficient of determination (R^2) from a linear regression for z-scores $> |2|$ between R1 and R2 are 0.73 for G401, 0.49 for BT16, and 0.48 for RCRF1009T. These values represent a moderate to good correlation level of reproducibility across our biological replicates, confirming the robustness of our measurements for the most impactful variants (those with z-score $> |2|$).

E) Figure 1F heatmap looks noisy, in the sense that few residues have consistent phenotypes when mutated to related amino acids. Put another way, the map doesn't have many stripes or blocks that are often seen in DMS.

We appreciate the reviewer's observation regarding the lack of uniform "stripes" or "blocks" in Figure 1F. While many deep mutational scanning (DMS) studies display such patterns, reflecting stretches of uniformly sensitive residues or shared phenotypes among related amino acid substitutions, the landscape observed for SMARCB1 appears more heterogeneous.

We believe this is a biologically meaningful feature of SMARCB1, rather than an artifact or sign of assay noise. Importantly, replicate concordance (see revised **Extended Fig 2a**) and performance of internal controls (i.e. frameshift, silent, and stop mutations) demonstrate high assay reproducibility, suggesting that the observed mutational effects are robust and reflect genuine biological properties.

This heterogeneous pattern aligns with the biological role of SMARCB1 and the broader principles of mutational robustness in large macromolecular complexes. Unlike enzymes or DNA-binding proteins with well-defined active sites, SMARCB1 functions primarily as a structural scaffold within the SWI/SNF chromatin remodeling complex. Its essential roles are mediated through discrete protein-protein interfaces, often involving a limited number of contact residues. In such contexts, even chemically similar substitutions can have divergent effects depending on side chain orientation, local structure, or binding partner context—limiting the appearance of uniform "blocks" of sensitivity. More broadly, proteins that operate as components of large complexes often exhibit increased tolerance to mutation due to structural redundancy and evolutionary pressures that minimize the cost of misfolding and translational errors¹¹. As a core subunit of SWI/SNF, SMARCB1 likely shares these properties, allowing the complex to buffer many substitutions without complete functional loss.

Moreover, the observed heterogeneity is consistent with the broader mutational tolerance of SMARCB1 in human tumors. While certain recurrent or hotspot mutations have been reported, SMARCB1 generally exhibits a low background mutation rate in large-scale tumor sequencing efforts, especially compared to other tumor suppressors. This supports a model in which many single-point mutations are tolerated without full loss of function. In our DMS, this is reflected by a broad distribution of variant phenotypes—including many tolerated and even gain-of-function substitutions—rather than uniformly deleterious effects.

Taken together, these observations indicate that the heterogeneous mutational landscape reflects an intrinsic biological property of SMARCB1, rather than a technical limitation of the assay.

F) How do results compare across cell lines?

In the figure below, we have included data for the 554 substitutions that achieved an average z-score >2 across all six replicates (two biological replicates per cell line; missense = 101, nonsense = 197, frameshift = 256). Z-scores shown represent the average of two biological replicates for each cell line.

To assess reproducibility across cell lines, we performed comprehensive pairwise comparisons of mutation fitness effects between BT16, CCLF_PEDS1009T, and G401 cell lines for all substitutions achieving z-score >2 when averaged across replicates.

Overall concordance ranged from 69.9-79.8% across the three pairwise comparisons, with 66.6-78.2% of mutations showing concordant significant gains (z-score >2) in both cell lines of each pair. Effect sizes were significantly correlated across all comparisons (Pearson r = 0.32-0.40, all p < 2.2e-15), demonstrating systematic agreement in identifying deleterious mutations.

Three-way overlap analysis revealed that 336 mutations (60.6%) showed significant enrichment (z-score > 2) across all three cell lines, representing a high-confidence consensus set of beneficial mutations. An additional 503 mutations (90.8%) were significant in at least two cell lines. Cell line-specific effects were rare (51 mutations total, 9.2%), indicating that the majority of functional mutations confer fitness benefits across diverse SMARCB1-deficient cellular contexts.

These results demonstrate robust reproducibility of the DMS screen across independent cell line models, with strong concordance in both the identity of deleterious mutations and the magnitude of their fitness effects.

Agreement
 ● Both Gain
 ● Both Neutral
 ● Discordant

2. I have a concern that the approach of combining the plasmid pools for different tiles into one pool created

more issues than it solved. How would lentiviral recombination affect the results, would many cells receive a transgene with multiple mutations? How about variable PCR efficiency of different fragmented regions due to using Nextera? Is this why Z-scores compare to the corresponding silent mutation? It seems to me these two different approaches (screening one tile at a time versus full genes at a time) are being used across different DMS papers, and the authors insights on how to choose would be appreciated.

The tile method involves two stages:(1) purchasing oligo pool of 150nt long, that includes 90nt variable region where the variant library is encoded. Each pool member contains a single amino acid mutation. The oligo pool was cloned into a pUC vector, tile-by-tile; each cloned tile pool was sequenced to confirm the library quality. (2) When all tiles were qualified, the pUC tile pools were combined with equal weight by DNA quantification. The resulting pool was then cloned into a lentiviral expression library.

In stage #1, the pUC tile was individually qualified and any failed pUC pool was identified from the sequencing and replaced. In stage #2, yes, we had a choice of whether to keep tile pools separated or combined for cloning into the expression vector, viral packaging, and subsequent screening in cells. Our choice of pooling and cloning and screening all tiles together in a single larger pooled experiment was driven by time and cost savings versus needing to do all these downstream processes for each tile as well as the advantage of every variant competing within the same cell population to avoid batch differences between tiles. Note that this last issue could perhaps be addressed to inclusion of additional controls in every tile, but that would incur an even larger cost in downstream time and effort.

We are aware of the template swap phenomenon, particularly during lentiviral reverse transcription. The rate of template switching could be as high as 26% for a 720bp intervening distance¹² so this is certainly a valid consideration for libraries of this type.

While we did not explicitly assess the recombination rate in this screen, the expected consequences of barcode-swapping are expected to add noise rather than artifacts to the screen. That is, each mutant being tested could be swapped with many others (with distance between them a major factor of course) so that one would expect combinations of mutations in cells, on average, to sample the differences in phenotype seen for each mutant. However, in this screen, there is in fact signal, and beyond that the signals show non-random clustering by position, indicating that noise due to barcode swapping, while it may have dampened signal, did not prevent differences among mutants from being detected.

Furthermore, we performed independent replicates starting at the step of library transduction and these replicates exhibited good reproducibility in this screen (**Extended Fig 2a**). Specifically, these early time point samples have undergone viral transduction and integration, and that were not only aligned well among themselves but also with the plasmid library used for packaging into the virus, suggesting the recombination in viral integration, the primary cause of unwanted template swapping, is at low rate.

This suggests that noise due to barcode-swapping (or artifacts, although as noted these are not expected to arise from widespread barcode swapping) in any step downstream from the library transduction was limited because any such noise (or artifacts) would differ across replicates.

The observation that barcode swapping did not prevent detection of reproducible signal in this screen is consistent with many other published screens that incorporated the same strategy (including e.g. the 3.6kb EGFR¹³ that would be expected to be highly affected if swapping was a problem). Therefore, in summary, whatever barcode swapping did occur did not prevent detection and interpretation of the final enrichment scores.

Regarding PCR efficiency and Nextera bias: The library members differ one from the other minimally, therefore we expect minimal PCR bias for the whole-ORF extraction from the post-selection, post cell harvest gDNA. The purified whole-ORF PCR products contain the full length ORF plus some flanking sequence on each end of the OR. These products are fragmented by transposon shearing and then the resulting DNA fragments were barcoded by Illumina i5 and i7 index primers by limited-cycle PCR (12 cycles). We noted that Tn5 transposon insertion sites are quasi-random; there are no insertion deserts, but

the insertion frequency along the ORF sequences is not perfectly even¹⁴. Our method allows for this variability:

1. The indexing PCR in Nextera was kept at 12 cycles to minimize PCR efficiency bias.
2. In all screens, without exception, the signals are scored by comparing 2 samples, the drug-treatment sample and the reference sample (e.g., early time point sample or DMSO vehicle sample). The transposon insertion bias and PCR efficiency bias cancel out between two samples.
3. Our analysis software provides not only the raw count of each variant, but also the coverage of the position; this coverage sums up all reads (wild type reads plus reads of all variants) that pass through the position. The ratio of variant count over the coverage, termed as fraction, normalizes the abundance of the variants and removes the bias.

We have since added the following to the Discussion [lines 538-552]:

“The pooled screening strategy, while cost effective and designed to minimize batch effects by having all variants compete within the same cellular population, introduces potential noise from lentiviral template switching during reverse transcription. Template switching rates can be substantial over long genomic distances, potentially creating cells with multiple mutations rather than the intended single variants. However, our reproducible results, particularly at the beginning time points, across independent replicates suggest this technical noise did not prevent detection of meaningful functional differences between variants. A separate technical constraint arises from current short-read sequencing technologies, which limit our ability to directly identify and exclude molecules containing multiple variants, as individual sequencing reads cannot span the full ORF length. Advances in long-read sequencing technologies could improve the efficiency of future DMS screens by enabling detection and removal of recombinant molecules, thereby reducing noise and improving signal clarity. While our analysis pipeline normalized for coverage differences and Nextera fragmentation bias, these technical factors may still contribute to variability in variant detection. These considerations highlight the importance of replication and careful controls in large-scale functional screens.”

3. Can they explain why the RPT2 mutations would affect the distal WHD flexibility? To add controls, would these simulations show different results when tested on different mutations? And how does this WHD flexibility explain the proteomics result?

We thank the reviewer for this important question about the mechanism linking RPT2 mutations to WHD flexibility. In response to this and other feedback, we have brought on Dr. JC Gumbart, an expert in computational simulations of proteins and other biomolecules. We have since substantially revised our Molecular Dynamics analysis. Rather than focusing on global WHD flexibility changes, our updated analysis now specifically examines unfavorable van der Waals interactions introduced by RPT2 mutations within the RPT2 domain itself. This revised approach is presented in revised **Fig 4a-d**.

Figure 4
The following lines have been added to the text [lines 330-356]:

“Missense mutants disrupt attractive van der Waals forces in hydrophobic core of RPT2

To determine the molecular mechanism underlying the observed SWI/SNF complex destabilization, we performed twelve 200-ns molecular dynamics (MD) simulations on the AlphaFold-derived SMARCB1 structure containing the W281P and I315R mutations alongside wildtype and a control population variant (S299L) (**Fig 3f, Methods**). The RPT2 domain contains two alpha helices opposing two beta sheets, creating a hydrophobic barrel-like core³⁷. Both mutations significantly increased unfavorable van der Waals (VDW) interaction energy between these structural elements (**Fig 4a, Supplementary Table 23**). Relative to controls (WT and S299L), W281P showed a 4.9 ± 1.2 kcal/mol increase and I315R showed a 4.4 ± 0.57 kcal/mol increase in inter-regional energy (both $p < 0.01$, $n = 12$).

Contact frequency analysis revealed widespread loss of close contacts ($< 4\text{\AA}$) between alpha-helical and beta-sheet regions in both mutants compared to wildtype, with W281P and I315R showing distinct but overlapping patterns of contact disruption (**Fig 4b, Supplementary 24-25**). We then quantified the energetic consequence of these structural changes through detailed VDW energy calculations. Five residue pairs showed significant energy increases: three pairs (281-290, 281-293, 281-319) were disrupted by both mutations, while 281-315 was specifically affected by W281P and 279-293 by I315R (**Fig 4c, Supplementary Table 26**). Notably we observed one compensatory favorable interaction between residues 279 and 315 in I315R simulations. No significant changes were detected in electrostatic interactions.

Despite arising from different structural positions, both mutations converged on disruption of a shared hydrophobic network centered on residue 281 (**Fig 4d**). Interactions between residue 281 and residues 290, 293, and 319 emerged as critical stabilizing contacts, with their loss driving RPT2 destabilization. The cumulative VDW energy penalties were 4.50 kcal/mol for W281P and 3.54 kcal/mol for I315R, both exceeding the ~2 kcal/mol threshold commonly cited for functionally significant destabilization in structural studies^{38–40}, indicating severe structural compromise.”

4. Can they better address implications for diagnostics? I thought their critique of IHC-based diagnostics was impactful, but the study does not clearly propose alternatives (e.g. exome sequencing) for clinical use or explain what impact exactly these improved diagnostics would have. There is no discussion of how these findings could inform targeted therapies (e.g., restoring SWI/SNF complex composition).

We appreciate the reviewer’s interest in the diagnostic and therapeutic implications of our work. We agree that improved strategies are needed to complement or refine immunohistochemistry (IHC)-based detection of SMARCB1-deficient tumors. Our data suggest that certain missense variants, despite disrupting SMARCB1 function, can escape detection by standard IHC due to retained protein expression. As an alternative or complementary strategy, clinical tumor profiling via targeted DNA panels or exome sequencing would allow detection of such variants regardless of protein stability or expression, improving diagnostic sensitivity in ambiguous cases.

More broadly, distinguishing clearly damaging variants from those that are tolerated or functionally ambiguous is essential for accurate diagnosis and informed clinical decision-making, including diagnosis, surveillance, and treatment planning. For example, variant interpretation can guide inclusion in clinical trials of epigenetic therapies targeting SWI/SNF-deficient cancers and influence patient surveillance strategies in the case of germline variants. Our findings could thus serve as a reference map to aid variant curation efforts in molecular tumor boards or diagnostic pipelines.

We have since added this to the discussion (Lines 557-560).

“Complementary DNA-based approaches such as targeted panels or exome sequencing would capture functionally deleterious variants regardless of protein stability, improving diagnostic accuracy, guiding patient enrollment in SWI/SNF-targeted therapies, and informing germline variant counseling.”

Minor comments

5. Paper could benefit from better organization, logical flow, and curation of the main figures to directly show what is being compared. Some optional suggestions as examples: I think Figure 1A-D could be supplementary so the panels are focused on Deep Mutational Scanning (the title of the figure). Maybe a summary or average of CADD, GENIE, and/or COSMIC scores could be added as row(s) to the DMS heatmap for direct and compact comparison. The RNA-seq is introduced at two separate points (~line 283 and ~388). Fig 1C and 2A are described in the text as comparable and could be combined into one figure for ease of comparison. I felt the figure 1 ClinVar analysis could better come after fully describing results of the screen (i.e. the beginning of figure 2).

We appreciate the reviewer’s suggestions for better organization and logical flow. We have significantly revised the manuscript to better address concerns of flow and organization. For example:

1. We have since moved **Fig 1b** to revised **Extended Fig 1g** to improve the focus on DMS within Figure 1.
2. We have included an average for CADD, REVEL, and Alphasense, and GENIE average per residue into revised **Fig 1e**.
3. We moved the discussion of RNA-seq into one section.

6. Could they add the percentage of persistent peaks that have AP-1 motif?

We thank the reviewer for this suggestion. We have added the percentages of persistent peaks and significantly gained peaks that contain the AP-1(bZIP) motif to the revised **Fig 4c**. Specifically, 10.6% of persistent peaks and 29.5% of significantly gained peaks contain the AP-1(bZIP) motif. In addition, we have included percentages for CTCF and TEAD2.

We have added the following lines to the text 390-396:

“Motif enrichment analysis revealed the sequence features underlying these genomic distributions and their differential sensitivity to SMARCB1 disruption (**Fig 5c**). Persistent peaks were primarily enriched for CTCF binding motifs (20.6%), with AP-1 motifs present at lower frequency (10.1%). In contrast, gained peaks showed strong enrichment for AP-1 (29.5%) and TEAD2 (19.9%), with AP-1 sites being the most predominantly affected by SMARCB1 disruption, consistent with previous reports^{2,29,42}. Analysis of motif density across all accessible peaks in each condition further supported these trends (**Extended Data Fig 7a**).”

7. Do the DNA oligos only mutate a single nucleotide per oligo (not mentioned in Methods)? How often do the sequencing reads contain a single mutation or multiple or none?

Yes, each oligo is 150nt long, the middle 90nt encodes the variant (30 aa tile) and this variable region is flanked with 30nt each which are constant sequences allowing homogeneous recombination (Gibson assembly). The 30 aa tile is not a degenerate sequence. Rather, in this tile region, each oligo only alters a single codon at a time, and the 29 remaining codons are wild type. This region is subjected to oligo synthesis errors that show in addition to the intended nucleotide changes, in the same oligo strand, there are extra changes. These additional unwanted changes are detectable and quantifiable (~20% constructs) in our NGS reads. The majority of erroneous constructs carry a single extra nucleotide error (~84% of erroneous constructs), followed by 2 nucleotide errors (~10%), and 3 nucleotide errors (~4%).

We'd like to note that the unintended errors in the final expression library are concentrated within the tile, because the tiles were made through oligo synthesis (e.g., 1 error per 500 nucleotides), outside of the tile, the rest of ORF was produced by high fidelity DNA polymerase¹⁴.

8. The Fig 1H,I ClinVar ROC is suggestive that the screen provides improved clinical predictive power, but it seems like a severe limitation that there are only 2 cancer variants. Could it be useful to extend the evolutionary conservation and gnomAD analyses across the whole DMS?

We agree with the reviewer regarding the limitation of the ROC analysis in Fig. 1h. Upon closer examination, we found that the two ClinVar pathogenic variants classified under "hereditary cancer-

predisposing syndrome" (P14H and R53L) are both associated with schwannomatosis^{15–17}, a benign tumor syndrome associated with pain rather than a cancer which has malignant potential¹⁸. Since our proliferation-based assay specifically measures antiproliferative function relevant to cancer contexts, these schwannomatosis variants do not serve as appropriate positive controls for validating cancer-related pathogenicity predictions. Given the absence of validated pathogenic missense variants specifically associated with rhabdoid tumor predisposition or cancer in ClinVar, we have removed the ROC analysis from the revised manuscript. Following the reviewer's suggestion, we have instead expanded the evolutionary conservation and gnomAD analyses across the full DMS dataset in revised **Extended Fig. 4**.

We have since added the following text to our results: [lines 199-202]:

“We found that only 12 missense mutations in SMARCB1 were classified as pathogenic or likely pathogenic, with two of these classified under hereditary cancer-predisposing syndrome (P14H and R53L); however, clinically these variants have been associated with schwannomatosis, a non-malignant tumor syndrome^{28–30}.”

We have also removed **Fig 1h** and the following text from our revised manuscript [lines 203-208 in the original manuscript]:

“Since our screen was conducted in cancer cell lines to assess disruption of antiproliferative function, we focused our receiver operating characteristic (ROC) analysis on pathogenic missense mutations known to be linked with cancer as true positives (**Fig 1h**). In comparing the performance of various computational predictors to our functional assay, we observed that our DMS data (AUC=0.81) outperformed all other predictors: CADD (AUC=0.67), REVEL (AUC=0.69), and AlphaMissense (AUC=0.72).”

9. Label Z-score directions with loss/gain of function for improved interpretability. The mutated residues could be labeled in the video. The significant residues could be labeled in the DMS heatmap. The WHD domain could be labeled in the structure figure.

We thank the reviewer for this suggestion and have since added GOF and LOF directionality into **revised Fig 1e, Fig 1f, and Fig 2a** to improve interpretability. We have also highlighted the residues which reached significance (z-score >2) in revised **Fig 1e**. We have since significantly revised our MD section and the WHD is no longer a focus of our manuscript, we have opted to omit labeling the WHD.

The revised figures are shown below.

Fig 1e –

Fig 1f

Fig 2a

10. Line 135: could more directly say mutation recurrence does not correspond to proliferation in this assay. At this point, it remains to be proven that proliferation in this assay corresponds to pathogenicity. Line 400: the authors evaluated whether changes in chromatin accessibility were associated with changes in transcription (causation is not determined).

We appreciate the reviewer's suggestions regarding mutation recurrence not corresponding to proliferation in this assay. We have significantly revised the section discussing R377H based on several reviewer comments.

The revised section, with this comment removed has been added to line 128-140:

"To investigate the recurrent R377H variant, we re-expressed the corresponding silent mutant, R377R, the missense mutant, R377H, and the nonsense mutant R377* variants in G401 cells and compared proliferation upon induction (**Extended Data Fig 1d, Supplementary Table 3**). Compared to the R377R control ($74.5\% \pm 6.8\%$ growth suppression), both R377H and R377* showed reduced antiproliferative function with $62.0\% \pm 10.3\%$ and $62.0\% \pm 4.0\%$ suppression, respectively (**Extended Data Fig 1e-f**). While both variants showed a reduction in antiproliferative activity, both retained substantially more activity than the ATRT-derived W281* nonsense mutation ($28.7\% \pm 4.2\%$). These results demonstrate that SMARCB1 antiproliferative function can be disrupted to different degrees, with R377H showing only modest functional impairment. These findings may be consistent with clinical observations that individuals with Coffin-Siris syndrome harboring germline R377H mutations (ClinVar Accession: VCV000030203.13) do not develop cancers at higher rates than the population despite having other phenotypic effects²²."

We have also clarified in the manuscript that the relationship between chromatin accessibility and transcription is correlative, not causal.

The edited text is included in lines 444-445:

"To determine whether these transcriptional effects were associated with chromatin changes, we next compared differential accessibility and expression."

11. More acronyms and gene names could be defined for a broader audience, e.g. cBAF vs PBAF.

We thank the reviewer for this helpful suggestion. To improve clarity for a broader audience, we have defined cBAF (canonical BRG1-associated factor) and PBAF (polybromo-associated BRG1-associated factor) upon first mention in the revised text.

Revised text (lines 242-245):

"The first of these mutation intolerant regions, the WHD, adopts distinct conformations in the two SMARCB1-containing subfamilies of SWI/SNF complexes: cBAF (canonical BRG1-associated factor) and PBAF (polybromo-associated BRG1-associated factor). In cBAF, the WHD is positioned distal to the nucleosome³¹, whereas in PBAF it is proximal³² (**Extended Data Fig 3a**)."

12. The paper says both R52 and R53, I think there is a typo.

We thank the reviewer for pointing this out. R53 was initially included in error and has now been corrected to R52 in both the figure and the text. The text describing and labeling in Extended Data Fig. 3b has been updated to consistently reflect residue **R52**, in alignment with the structural analysis.

The following text has been added [lines 246-248]:

“Mutation-intolerant residues were mapped onto the cryo-EM structure of the PBAF complex, revealing that several intolerant residues, including R52, are located at positions that place them in spatial proximity to DNA in this structural model (**Extended Data Fig 3b**).”

13. I could not find the ORFeome screen described in Methods. Methods do not say how the DMS screen was performed. What is t=0 and what is t=end for each screen? Are there other timepoints too (suggested in Fig 1E)? Do longer endpoints allow for greater dynamic range in the assay? Was anything done to ensure the low multiplicity of infection?

ORFeome screen

The previously published ORFeome¹⁹ brought together a collection of 16,172 ORFs. Virus was produced at the Broad Institute based on the Genomic Perturbations Platform standards (<https://portals.broadinstitute.org/gpp/public/resources/protocols>). Virus was then titrated based on each cell line used in the screen (e.g., G401) using 50 to 200uL of virus per 3 million cells to achieve a MOI ranging between 0.2 to 0.6. The concentration of virus was then used to transduce G401 cells such that the overall representation rate >500 cells per ORF while under antibiotic selection with puromycin. A sampling of the transduced cells for the screen was separately plated to confirm that the MOI remained in the 0.2 to 0.6 range. This representation rate was maintained throughout the screen. Following transduction overnight, cells were selected with puromycin for 48 hours. A set of cells were harvested as the early timepoint to reflect the heterogeneity of viral constructs integrated. Remaining cells were then passaged every ~3-4 days as cells achieved 80% confluence. After ~10 days, cells were then harvested as the end timepoint. Genomic DNA was extracted from these early and late timepoints across two replicates based on the Genomic Perturbations Platform standards using the QIAamp gDNA extraction kit (Qiagen, Netherlands). Samples were transferred to the Genomic Perturbations Platform where sequencing libraries were generated from these samples and sequenced using Illumina platforms. FASTQ files were deconvoluted using PoolQ v2. Counts were then converted to a log2fold change for downstream analyses.

DMS screen

SMARCB1 ORF library was generated as above. Virus was produced at the Broad Institute based on the Genomic Perturbations Platform standards as above. Virus was then titrated based on each cell line used in the screen using 50 to 200uL of virus per 3 million cells to achieve a MOI ranging between 0.2 to 0.6. The concentration of virus was then used to transduce each cell line such that the overall representation rate >500 cells per ORF while under antibiotic selection with puromycin (see **Supplementary Table 8**). A sampling of the transduced cells for the screen was separately plated to confirm that the MOI remained in the 0.2 to 0.6 range. This representation rate was maintained throughout the screen. Following transduction overnight, cells were selected with puromycin for 48 hours for G401 cells or 72 hours for BT16 and CLF-PEDS9001T cells. A set of cells were harvested as the early timepoint to reflect the heterogeneity of viral constructs integrated. Remaining cells were then passaged every ~3-4 days as cells achieved 80% confluence. After 8-14 days, cells were then harvested as the end timepoint. Genomic DNA was extracted from these early and end timepoints across two biological replicates based on the Genomic Perturbations Platform standards using the QIAamp gDNA extraction kit (Qiagen, Netherlands). Samples were transferred to the Genomic Perturbations Platform where sequencing libraries were generated from these samples and sequenced using Illumina platforms (see “Nextera sequencing”). Analyses as in “Processing next-generation sequencing data.”

What is t=0 and what is t=end for each screen?

In the case of the ORFeome screen:

t=0 is 72 hours (following 24 hours transduction and 48 hours selection)

t=end is 7 days from time of transduction

In the case of the DMS screen, we aimed to have similar doubling times across the cell lines, G401, BT16 and CLF-PEDS9001T

t=0 is 72 hours (following 24 hours transduction and 48 hours selection) for G401 and 96 hours (following 24 hours transduction and 72 hours selection)

t=end is 10 days for G401, 8 days for BT16 and 14 days for CLF-PEDS9001T from time of transduction

Are there other timepoints too (suggested in Fig 1E)? Do longer endpoints allow for greater dynamic range in the assay?

No, there are not additional timepoints where variant abundance was assessed. Based on our prior DMS screens¹⁴, we have seen increased experimental noise with longer timepoints. In addition, we wanted to ensure similar cell doublings between cell lines (4.23-8 doublings). As a result, we used the above timepoints to maintain sufficient growth competition while minimizing experimental noise.

Was anything done to ensure the low multiplicity of infection?

As noted above, a titration was initially performed with the batch of virus used to achieve a MOI between 0.2 - 0.6. Once this was obtained, the screens were performed using the amount of virus that would achieve a MOI closer to the 0.2 - 0.3 range. During the course of the screen, cells were separately plated to calculate the MOI and so long it remained in the 0.2 - 0.6 range, screens were carried forward. As in the new **Supplementary Table 8** which outlines the details of the screens across cell lines and biological replicates, the MOI ranged from 19.3-35.4%.

We have since added the following text to provide methodology for the cDNA screen in our methods [lines 565-591]:

“ORFeome Overexpression Screen

Library and viral production: The ORFeome library consisted of 16,172 human open reading frames cloned into lentiviral expression vectors as previously described⁴⁷. Lentiviral particles were produced at the Broad Institute Genetic Perturbation Platform (GPP) following standard protocols (<https://portals.broadinstitute.org/gpp/public/resources/protocols>).

Transduction and MOI optimization: Viral supernatant was titrated in G401 cells using 50-200 μ L of virus per 3 million cells to achieve a multiplicity of infection (MOI) of 0.2-0.6. To ensure low MOI and minimize cells receiving multiple viral integrations, a separate aliquot of transduced cells was plated and puromycin-resistant colonies were counted to confirm MOI remained within the target range. For the screen, G401 cells were transduced at the optimized viral concentration to maintain a representation of >500 cells per ORF throughout the experiment.

Screen timeline and sample collection: Following overnight transduction, cells underwent puromycin selection for 48 hours. Cells were then harvested at two timepoints: (1) the early timepoint (t=0) at 72 hours post-transduction (24 hours transduction + 48 hours selection), representing the initial distribution of integrated viral constructs; and (2) the end timepoint (t=end) at 7 days post-transduction (~10 days total culture time), after cells were passaged every 3-4 days upon reaching 80% confluence. This endpoint allowed sufficient time for proliferative effects to manifest.

DNA extraction and sequencing: Genomic DNA was extracted from cell pellets using the QIAamp DNA Blood Maxi Kit (QIAGEN) according to GPP standards. Integrated ORF barcodes were amplified by PCR, and

sequencing libraries were prepared and sequenced on Illumina platforms at the GPP. FASTQ files were demultiplexed and processed using PoolQ v2 software. Barcode counts were normalized and converted to log₂ fold changes ($t=\text{end}/t=0$) for downstream analysis.”

We have also clarified in the methods how the DMS screen was performed in addition to outlining key metrics in **Supplementary Table 8** [lines 635-653]:

“Transduction and MOI validation: Viral supernatant was titrated in each cell line (G401, BT16, and CCLF-PEDS9001_T1) using 50-200 μL of virus per 3 million cells to achieve an MOI of 0.2-0.6. Low MOI is critical to ensure that the majority of transduced cells receive only a single viral integration. To validate MOI, a separate aliquot of transduced cells was plated at limiting dilution and puromycin-resistant colonies were quantified to confirm MOI remained within the target range. For the screen, cells were transduced at the optimized viral concentration and centrifuged at 2,000 rpm for 30 minutes at 30°C to enhance viral transduction. A representation of >700 cells per variant was maintained throughout the experiment (**Supplementary Table 8**).

Screen timeline and sample collection: Following overnight transduction, cells were selected with puromycin for 48 hours (G401) or 72 hours (BT16 and CCLF-PEDS9001_T1). Cells were harvested at two timepoints: (1) the early timepoint ($t=0$) immediately after selection (72 hours for G401; 96 hours for BT16 and CCLF-PEDS9001_T1), representing the initial distribution of integrated variants as indicated in **Extended Fig. 2a** (start of screen); and (2) the end timepoint ($t=\text{end}$) at 8-14 days post-transduction, depending on cell line growth rates. Cells were passaged every 3-4 days upon reaching 80% confluence to maintain log-phase growth. The extended culture period allowed for increased dynamic range in detecting proliferative effects of loss-of-function variants. Two independent biological replicates were performed for each cell line.”

Reviewer #5 (Remarks to the Author):

Summary

This is a compelling work where the authors explore the function of missense mutations in the BAF complex subunit, SMARCB1, which are part of the complex's early-stage formation. The field has focused on immunohistochemistry quantification of SMARCB1 in pediatric tumors, yet missense mutations that may alter function but not expression level may be missed. Deep Mutation Scanning has been applied to more frequently altered genes, and the authors apply it now in one of the first uses to characterize a low-frequency tumor suppressor gene. They found that missense variation in SMARCB1 can cause genome-wide changes that are heterogeneous with details on a per-allele basis. The manuscript is well-written and organized. Overall, I believe it presents information with high relevance and reusability. The supplemental tables are commendably complete. While data is not all publicly accessible, it is appropriately housed in dbGaP as explained on their github page, which is publicly accessible.

Points to Improve the Manuscript

1. More recent works by, e.g., Kadoch lab, have characterized the order of formation for the canonical versus non-canonical BAF complex. The current manuscript would be improved by expanding the Discussion to include the author's interpretation of SMARCB1 mutation in light of the different functions of the canonical versus non-canonical BAF in pediatric malignancies. Fig 3e shows it is mostly base unit loss for each subfamily.

We thank the reviewer for highlighting the importance of considering canonical versus non-canonical BAF complex functions in the context of SMARCB1 mutations. We agree that recent work from the Kadoch laboratory on SWI/SNF complex assembly has provided important insights into the differential assembly and roles of cBAF, PBAF, and GBAF complexes²⁰.

In our analysis, we observed selective disruption of SMARCB1-containing complexes with distinct patterns: cBAF complexes showed complete loss of all subfamily-specific subunits (DPF2, ARID1A/B), while PBAF complexes were only partially disrupted, with PBRM1 being the only subfamily-specific subunit to dissociate (while BRD7, ARID2, and PHF10 were retained). GBAF-specific subunits remained unaffected, which is expected since GBAF complexes naturally lack SMARCB1. Interestingly, we also observed significant destabilization of core BAF subunits (SMARCC1, SMARCD2, SMARCD3) that are shared across all complexes and form the initial assembly scaffold (**Fig 3c**). This suggests that SMARCB1 mutations may exert destabilizing effects after incorporation into assembled complexes, rather than solely disrupting the hierarchical assembly process described by Kadoch and colleagues. We have expanded our Discussion to incorporate these findings within the framework of differential SWI/SNF complex functions and their implications for pediatric malignancies. This includes discussing how the selective preservation of GBAF, partial preservation of PBAF, and complete disruption of cBAF may contribute to the specific oncogenic consequences and chromatin accessibility patterns observed with SMARCB1 mutations.

We have since added the following lines to the discussion (lines 479-486):

"While cBAF complexes showed complete subfamily-specific disruption (loss of DPF2, ARID1A/B), PBAF complexes were only partially affected, with PBRM1 dissociating while other subfamily-specific subunits (BRD7, ARID2, PHF10) remained unaffected. Subunits specific to GBAF complexes were entirely unaffected, consistent with these complexes naturally lacking SMARCB1. Importantly, we observed destabilization of shared initial BAF core subunits (SMARCC1, SMARCD2, SMARCD3), suggesting the mutant SMARCB1 creates post-assembly structural instability that propagates differentially across complex subtypes. "

2. The supplemental video of MD simulations shows that global translational and rotational degrees of freedom may not have been accounted for before analysis. If the video is a visualization of the data as it was analyzed, then I have concerns about the alignment-dependent metrics like RMSD and RMSF.

We thank the reviewer for this important question about the alignment-based metrics used in our initial submission. In response to this and other feedback above, we have brought on Dr. JC Gumbart, an expert

in computational simulations of proteins and other biomolecules. Our analyses of the data which now includes 12 replicates of MD simulations for 200ns has been significantly revised. Our updated analysis now specifically examines unfavorable van der Waals interactions introduced by RPT2 mutations within the RPT2 domain itself. This revised approach has been included in revised **Fig 4a-d** shown above in **Reviewer #4 Comment #3**.

3. What is the doubling time of G401 cells? How many divisions do you expect in the 10-days of the DMS study? Can you set an expected limit of detection based on that expectation?

We thank the reviewer for the question regarding the doubling time in G401 and limit of detection (LOD) in our DMS screen.

We calculated the doubling times from cell counts at multiple intervals during the 10-day screen in G401, resulting in an estimated total of ~5.8 to 7.6 population doublings across replicates. This corresponds to a population expansion of approximately 58- to 200-fold. Similar doubling times were calculated for both BT16 and RCRF1009T to achieve a minimum of 4 doublings after introduction of the library.

To estimate the limit of detection, we first calculated the initial variant representation by multiplying the number of cells surviving puromycin selection by the proportion of sequencing reads mapping to single amino acid variants analyzed in our study. This provided a corrected estimate of the number of usable variant-bearing cells at baseline (i.e., immediately post-selection). We applied the same read-mapping adjustment to the number of cells harvested at each subsequent timepoint (Day 3 and Day 10) to estimate per-variant representation over time.

For example, in replicate 1, the initial representation was approximately 865 cells per variant, increasing to over 4,500 cells per variant by Day 3 and over 5,000 by Day 10. This calculation accounts for both the expansion of the cell population and the proportion of reads which capture our analyzed mutants.

We then calculated the theoretical limit of detection using the formula:

$$LOD = \frac{1}{\text{Initial variant representation} \times \text{Population expansion}}$$

Applying this to our data, for G401 replicate 1 with an initial representation of ~865 cells per variant and a population expansion of ~58-fold, the LOD is approximately:

$$\frac{1}{865 \times 58} \approx 0.00002 \text{ (0.002\%)}$$

Put simply, this means that a variant could decline to as little as 0.002% of the total population and still be detected at least once in sequencing data at the end of the experiment. Calculations for both replicates in each cell line have been provided in **Supplementary Table 8**.

This approach ensures that our LOD calculations accurately reflect the initial diversity of the library and consider biological growth and sequencing coverage throughout the experiment.

4. While I like Figure 2e, I think the paper would be improved by a structural assessment of where and why there are amino acid type dependencies for the structures. That is, it looks to me like AlphaMissense is basically saying most changes at any structured region are damaging. Yet, the author's data indicates that it is more subtle, and the damaging effect depends on the alternate amino acid per site. I would like to see more information about this level of data (for any MAVE / DMS, but for this paper too!).

We thank the reviewer for this suggestion. To address whether amino acid-specific dependencies map to structural features, we performed a structural analysis of SMARCB1 using AlphaFold-predicted structure, examining secondary structure elements (DSSP assignments), solvent accessibility (FreeSASA, RSA < 0.1

= buried), and noncovalent interaction networks (hydrogen bonds identified in PyMOL) (revised **Extended Fig 2f**).

We quantified two complementary metrics for each structural category: (1) overall mutation tolerance (mean z-score across all 19 possible substitutions per residue), where higher values indicate more deleterious mutations, and (2) amino acid selectivity (mean standard deviation of centered z-scores), where higher values indicate that the identity of the substituted amino acid influences the functional outcome.

Secondary structure: Beta sheet residues show higher amino acid selectivity (centered SD = 1.45) compared to helices (1.19) or loops (1.23). Overall tolerance is similar across all three categories (mean z-scores range from -0.056 to +0.002). This suggests that while sheet positions tolerate mutations at rates comparable to other regions, the identity of the substituted amino acid matters more - likely reflecting geometric constraints of beta strand packing.

Solvent accessibility: Buried residues (RSA < 0.1) are less tolerant to mutation overall (mean z-score = +0.20) compared to exposed residues (-0.05), while showing similar amino acid selectivity (centered SD ~1.24-1.26). This indicates that core positions are generally more constrained, consistent with the importance of hydrophobic packing for protein stability.

Interaction networks: Residues involved in hydrogen bonding networks show lower amino acid selectivity (centered SD = 1.13) than non-H-bonded positions (1.29), with similar overall tolerance (mean z-scores ~0). This pattern suggests that H-bonded positions may be uniformly constrained rather than having strong amino acid-specific preferences.

Together, these data demonstrate that mutation constraint in SMARCB1 is context-dependent: some positions (beta sheets) tolerate only specific substitutions, while others (buried residues) are uniformly intolerant.

The following text has been added [lines 236-239]:

“ Structural mapping revealed that β-sheet residues exhibited higher amino acid-specific selectivity and buried residues (RSA < 0.1) showed higher overall mutational intolerance (e.g. higher average z-score), indicating that mutation effects depend on both structural context and substitution identity (**Extended Fig 2f**).”

5. Candidate loss-of-function variants are discussed, yet the authors data indicate that similar Z-score magnitude change, but in the negative direction, could be used to prioritize and compare candidate gain-of-function variants.

We appreciate the reviewer’s point regarding the bidirectionality of our z-score framework and the potential to use negative scores to prioritize candidate gain-of-function variants. The term gain-of-function has several definitions depending on the context. In classical genetics, gain-of-function refers to mutations that

confer new or enhanced normal biological activity to a gene product. However, in cancer biology, gain-of-function often describes oncogenic properties acquired by mutant proteins - such as mutant p53's ability to promote metastasis and drug resistance beyond simply losing its tumor suppressor activity²¹.

Here, we would classify variants with highly negative z-scores as those representing gain-of-function using the classical genetics definition, rather than gain-of-function leading to pro-oncogenic phenotypes. These highly negative variants could include rare substitutions that enhance protein stability, chromatin targeting, or interaction with complex members, and may merit further study.

To clarify this point, we have revised the **Discussion** to explain that large negative z-scores may be used to identify functionally intact or enhanced variants, but not necessarily oncogenic GoF mutations. Importantly, we note that strong depletion signals, such as those driving highly negative Z-scores, can be more sensitive to experimental noise, particularly in pooled screens²². In contrast, variants that increase in frequency over time are often more robustly detectable, as such enrichment is less prone to dropout-related artifacts.

Modifications to text [lines 521-530]:

“While our primary focus was identifying loss-of-function variants, our scoring framework is symmetric and enables interpretation of variants at both functional extremes. More negative z-scores indicate stronger depletion upon SMARCB1 re-expression, consistent with greater antiproliferative activity. Variants with z-scores less than -2 may exhibit enhanced activity through improved protein stability, SWI/SNF complex incorporation, or chromatin interactions. These represent functionally enhanced variants rather than oncogenic gain-of-function mutations. However, strong depletion signals driving highly negative z-scores can be more sensitive to experimental noise in pooled screens, whereas enrichment signals are more robustly detectable⁴⁶. This underscores saturation mutagenesis potential to uncover functional enhancement modes beyond loss of function.”

6. Are there patterns among or explanation for why about half of the nonsense mutations are $Z < 2$ in the current study? The concordance of half is emphasized, yet the discordance of half is interesting.

We thank the reviewer for this thoughtful observation. The finding that approximately half of the nonsense mutations exhibited z-scores < 2 likely reflects distinct biological consequences depending on the region of the protein in which the truncation occurs.

As noted in the revised manuscript, “C-terminal nonsense and frameshift mutations beginning at residue 350 retained partial functionality, suggesting that these truncations preserve critical domains required for antiproliferative activity. This observation is consistent with the previously characterized R377* mutant, which had a z-score of 0.296 and achieved a mean $62\% \pm 4.0\%$ reduction in proliferation upon re-expression over 8 days (Extended Fig. 1d–f).”

Similar outcomes have been reported in two independent cDNA-based deep mutational scanning studies (e.g., *TP53* and *PTEN*), which also identified unexpected retention of function among nonsense mutations occurring near protein termini^{23,24}. To further place our results in the context of patient-derived variants, we examined preliminary data from the CCDI Molecular Cancer Initiative cohort²⁵, which includes 13 individuals with ATRT (n=11) or malignant rhabdoid tumor (n=2) that had nonsense mutations. Notably, no nonsense mutations were observed in either the N-terminal region ($< \text{aa } 47$) or the C-terminal region ($> \text{aa } 318$). These clinical data are consistent with our findings that nonsense mutations in the terminal regions retain partial functionality.

7. Are discordances between DMS results and CADD/others due to sites where the changes in isoform 1, targeted by the mutagenesis library, have the same type of change in other isoforms?

We thank the reviewer for this insightful question. CADD, REVEL, and AlphaMissense predict variant effects through different methods, each of which will be discussed here.

CADD

The possibility of differential effects across isoforms may contribute to some DMS-CADD discordances and represents an underappreciated source of complexity in variant interpretation. Isoforms arising from alternative splicing can differ substantially in their functional domains, regulatory regions, and structural contexts, potentially leading to dramatically different functional consequences for the same variant depending on the isoform context.

CADD synthesizes information from various correlated features rather than relying on single transcript assessments. As described in the CADD methodology²⁶, the tool integrates multiple sources of data including DNA sequence properties, gene and transcript models, scoring of protein coding effects, genomic conservation, biochemical activity and other genomic annotations into a unified score reported at the genomic level. This integrated approach means CADD's predictions incorporate information across multiple isoforms and annotation sources simultaneously.

While differential isoform effects could theoretically contribute to discordances, CADD's broad integration framework likely mitigates most isoform-specific issues by averaging signals across multiple transcript models and annotation sources. However, this integration approach may simultaneously mask isoform-specific functional effects that are captured by DMS experiments targeting a particular isoform.

We believe the primary sources of DMS-CADD discordances are cell-type and condition-specific effects not captured in CADD's broadly trained model, rather than isoform-specific considerations. CADD's genomic-level integration approach, while comprehensive in scope, fundamentally differs from the targeted functional mechanisms and cellular contexts tested in DMS assays, making methodological differences the more likely explanation for observed discordances.

REVEL

For REVEL, the reviewer's question about isoform concordance is particularly relevant since REVEL provides transcript-specific annotations and averages scores across all affected isoforms when multiple protein isoforms are associated with a variant⁶. This averaging approach could contribute to DMS-REVEL discordances, as a single variant may have different functional consequences across isoforms due to varying structural contexts, functional domains, or regulatory regions. Our DMS experiments targeting isoform 1 capture the functional impact in that specific molecular context, while REVEL's ensemble approach may either misrepresent the functionally relevant isoform or dilute true variant effects through averaging. However, we believe the primary sources of discordance remain the fundamental methodological differences between computational prediction approaches and experimental functional

measurements, as REVEL's ensemble of 18 computational scores may not fully capture the specific functional mechanisms and cellular contexts tested in DMS assays.

AlphaMissense

For AlphaMissense, the reviewer's question about isoform concordance does not directly explain the observed discordances, since both our DMS experiments and AlphaMissense predictions target the same isoform (isoform 1). AlphaMissense predicts pathogenicity scores for all possible single amino acid substitutions using the AlphaFold architecture trained on structure prediction and protein language modeling, then fine-tuned on human population frequency data⁸. Since both approaches examine variants in the same sequence context, isoform effects would not create systematic discordances between DMS and AlphaMissense results.

The observed discordances are more likely attributable to fundamental differences between AlphaMissense's computational approach and experimental functional measurements. AlphaMissense's pathogenicity predictions, while incorporating sophisticated sequence modeling and evolutionary constraints, are based on population frequency patterns and may not fully capture the specific molecular mechanisms, cellular contexts, and functional readouts measured in DMS assays. The discordances likely reflect limitations in translating population-level observations to mechanistic functional effects rather than isoform-specific considerations.

8. I would like to see much more direct comparison of the data from GENIE and Cosmic, with the DMS results. Fig 1a should be log scale since just a few are highly recurrent variants dominate the view. Then, is there a relationship between the DMS incidence and tumor incidence? This is not made directly clear. There is an important baseline assumption that mutating the chromatin remodeling has a malignant effect via a mechanism of altered cell growth. It may not be. It may be stemness, fate, plasticity, etc., especially given the author's data that there are differences in genome-wide regulation downstream of different missense mutations. Not all mutations may promote malignancy by the same mechanisms. I would like this assumption due to the design of the experiments to be stated and explained up-front. Then, the comparison of DMS and somatic incidence will indirectly support/not. Finally, that will help frame the other results of the paper.

We thank the reviewer for these helpful suggestions. In the revised manuscript, we have revised **Fig. 1a** to display the missense mutation counts on a log₁₀ scale, which better reflects the distribution and highlights lower-frequency variants that were previously obscured by highly recurrent mutations.

The revised **Figure 1A** is presented here:

Additionally, we have added a direct comparison between DMS mutation scores with tumor incidence in patient data (**Extended Fig 5b and shown above in Reviewer 3 Comment 1**) and observed little correlation ($R^2 = 0.01$). This finding suggests that while this DMS reflects functional impact on

proliferation, it does not fully capture the complex selective pressures shaping mutation prevalence in tumors.

To contextualize our results, we have added the following lines of text (lines 95-99) to the introduction: “Here, we perform DMS of SMARCB1 by systematically evaluating the functional consequences across its entire coding sequence using a proliferation-based phenotypic framework. While recognizing that mutations in chromatin remodeling genes may contribute to tumorigenesis through multiple mechanisms, we employ proliferation as a quantifiable and clinically relevant proxy for malignant potential.”

9. The authors report that, “all three computational predictors identified a high proportion of missense variants as pathogenic [...]. This finding contrasts with the low somatic missense mutation frequency of SMARCB1 in patient tumor samples [...]. If such a large proportion of variants were truly pathogenic, we would anticipate a higher frequency of missense mutations in SMARCB1 in patients.” I do not fully agree with the logic used in this statement. It seems entirely possible that a gene could be mutationally intolerant and yet not mutated in tumors as long as those mutations did not confer a niche advantage to malignant cells. I recommend the authors reorient this section. It’s already clear to me, for example and from my assessment of many other datasets, that AlphaMissense over-calls at structured sites. I recommend something around this type of finding be more the orientation and as it pertains to your DMS measurements and where they are and are not concordant.

We thank the reviewer for this suggestion. We agree that the prior discussion of patient mutation frequency could be misleading, as mutational intolerance does not necessarily correlate with increased mutation frequency in tumors. In response, we have removed the section discussing the correlation between mutation frequency and computational predictors, and have reoriented the section to focus on the concordance between computational predictors and our functional measurements. Specifically, we now highlight that AlphaMissense, REVEL, and CADD overestimate the impact of R377H on antiproliferative function: although R377H ranks in the 90th, 82nd, and 97th percentiles, respectively, it exhibits an intermediate phenotype in our assays. This reorientation emphasizes the limitations of general pathogenicity predictions in capturing context-specific function, such as proliferation, and motivates our use of deep mutational scanning to systematically characterize the functional impact of all SMARCB1 missense variants on this phenotypic readout.

The following changes have been made [lines 145-149]:

“These tools classified 96.5%, 76.5%, and 37.8% of missense variants as deleterious, respectively (**Fig 1b, Extended Data Fig 1j-k, Methods**). However, predictions for R377H did not align with our experimental observations. R377H ranked in the 90th, 82nd, and 97th percentiles for AlphaMissense, REVEL, and CADD, respectively, yet exhibited only intermediate functional impairment in our proliferation assays.”

10. Molecular Dynamics questions

a. The authors state that the AF2 structure was used, but it’s not completely clear if that is what was use for the MD input. Please clarify this in Methods.

We thank the reviewer for requesting this clarification. The AlphaFold2 structure for SMARCB1 (UniProt ID: Q12824) was used directly as the starting structure for all Molecular Dynamics simulations. The structure file was obtained from the AlphaFold Protein Structure Database, processed with pdb4amber, and then used as input for system preparation and subsequent MD simulations without further structural modifications.

We have since added the following text to the **Methods**:

“Molecular dynamics simulations were performed using the AlphaFold2 structure for SMARCB1 (UniProt ID: Q12824) as the starting conformation.”

b. The authors report that “Harmonic positional restraints with a weight of 2.0 kcal/mol·Å² were applied to the alpha carbons (Cα) of residues 1–385 to maintain structural integrity.”

i. Is that 2kcal/molÅ² distributed across the 385 Cα atoms, or per atom? Please make clear in the sentence.

We thank the reviewer for raising this concern. The statement regarding harmonic positional restraints during production MD was an error from our initial version of our simulation protocol. We have since included an expert in molecular dynamic simulation (Dr. JC Gumbart) and he has assisted us in revising our pipeline. In the currently presented simulations reported in this revised manuscript, no positional restraints were applied during the production MD phase (ntr = 0 in our production input file). Harmonic positional restraints with a force constant of 10.0 kcal/mol·Å² were applied only during the initial restrained minimization step to all protein heavy atoms (residues 1-385), after which all subsequent equilibration and production phases proceeded without restraints to allow full conformational flexibility of the protein. We have corrected this in the revised Methods section.

ii. The IDR appears to receive the same treatment as the ordered domains. Please discuss strengths and weaknesses of this approach. I would, for example, like you to not include IDR in some calculations like RMSD – plus, what is the per-domain RMSD? That can be telling to understand what is intra-domain and inter-domain flexibility changes. Plus, that there are multiple experimental structures with the WHD in different positions – how do you interpret the changes in flexibility observed? Are some mutants more likely to be found interacting with the DNA in one way, and other mutations in another?

As mentioned in previous comments, we have significantly revised the methodology and conclusions from our MD simulations. We have since removed our interpretation of WHD flexibility as a driving mechanism for antiproliferative function. However in regards to the comment on DNA interacting mutants we hypothesize that specific WHD mutants might interfere with the ability of the WHD to bind to DNA particularly in the PBAF complex. This hypothesis is based on the published structural data presented in **Extended Fig 3a-b**.

We have since added this structure based hypothesis to the Discussion [lines 510-519]:
“Our findings across other domains reveal additional structure-function relationships that warrant further investigation. The WHD’s mutational intolerance may reflect its distinct conformations in cBAF versus PBAF complexes observed in cryo-EM structures, potentially explaining complex-specific functional requirements. Within this domain, our identification of R52 as highly mutation-intolerant aligns with structural predictions of its insertion into the DNA major groove in PBAF structures, suggesting a direct role in DNA binding that could explain its functional importance. The cluster of mutation-intolerant residues within the IDR (E122, Q123, A125) suggests this region may facilitate conformational flexibility required for dynamic protein-protein interactions. These structure-based hypotheses provide promising directions for future experimental validation.”

c. The dbGaP links or IDs should be substituted for the “[X]s” in the github page so that interested users can more readily find and access the controlled access data.

We have since added the following text to the github page for easier access to the controlled data:
“All raw sequencing data can be obtained from dbGaP under accession number phs003896.v1.p1.”

d. Other variations in the three patient-derived SMARCB1-deficient cell lines?

As part of the Cancer Cell Line Encyclopedia (CCLE), BT16 and G401 have been previously genomically characterized, and the variants are included publically on DepMap²⁷. CCLF_PEDS9001_T1 has been genomically characterized in a prior study from our group²⁸. We have attached all genetic alterations for the three patient derived cell lines used in this study within **Supplementary Table 36**.

e. Does the rolling-average of silent mutations approach lead to an estimatable expectation for the minimum effect size that a “truly damaging silent mutation” would have to achieve to have a significant Z-score? And, what if there were regions where all variations had subtle but consistent effects? Consider discussing these as limitations of the approach. Can you consult the eQTL Catalog to demonstrate that there are no eQTL silent mutations here, for a different approach to justify your finding that nearly no silent mutations had an effect on cell growth?

The reviewer raises important questions about the quantitative thresholds and regional biases in our rolling-average method. Yes, our approach produces quantifiable minimum effect size expectations for statistical significance, which vary across the protein depending on local silent mutation behavior. On average, variants must exceed $\sim 0.57 \log_2$ fold-change (2 standard deviations above the local silent mean) to be considered significant, though the exact threshold is position-specific.

We observed, for example, that residues 63–93 displayed an upward shift in silent mutations (mean 0.2 \log_2 as compared to -0.4 \log_2 in the rest of the protein). Because our rolling z-score method normalizes variant effects relative to silent behavior at the same positions, we are able to better mitigate this regional bias. An inherent assumption of this strategy is that silent mutations have no effect on the antiproliferative phenotype of SMARCB1.

To provide an independent validation of this assumption, we systematically analyzed the GTEx database (v10), the most comprehensive and widely-used eQTL resource which utilizes 54 non-diseased tissue sites across nearly 1000 individuals²⁹. We extracted all eQTL variants from GTEx v10 for SMARCB1 and filtered for those falling within the nine exonic regions of the gene (covering the complete 385 amino acid coding sequence). Of the 881 unique total SMARCB1 SNV eQTLs identified, only 1 variant (0.11%) fell within exonic regions: a G→A mutation at chr22:23825326 encoding a silent mutation at S299. This variant showed significant effects on SMARCB1 expression with a mean NES of -0.381 ± 0.113 across 44 tissue types.

To contextualize this finding in population-wide databases, we looked for this variant in gnomAD. This particular silent variant is common in the general population (gnomAD allele count: 182,809; homozygote count: 10,981), strongly indicating it is not pathogenic. The high frequency of this variant in healthy individuals, especially in the context of homozygotes, supports that even eQTL-associated silent mutations may not translate to clinically relevant functional effects.

Our functional screen included this exact variant, which achieved an average z-score of 1.16, suggesting modest impact on SMARCB1 function. However, our library also contained 4 additional unique silent mutants at the same position (S299), which achieved a mean z-score of -0.116 ± 0.909 . When we applied our standard approach of averaging z-scores across all variants encoding the same amino acid, the consensus score for position 299 was 0.141 ± 0.358 —essentially neutral. Taken together, our systematic analysis confirms the rarity of functional silent mutations in SMARCB1.

This analysis demonstrates that: (1) functional silent eQTLs in SMARCB1 exons are extremely rare (1/881 variants), (2) even when present, they may reflect benign population variants rather than pathogenic mutations, and (3) our experimental design of averaging multiple synonymous variants per position effectively controls for rare outlier effects. Importantly no functional silent variants were found between amino acids 63-93, consistent with our interpretation that the observed increase in enrichment in this region reflects a technically artifact rather than true biological signal. These findings independently confirm our underlying assumption that silent mutations have minimal functional impact on SMARCB1 and justify the validity of using a rolling z-score approach to correct for technical artifacts.

We note, however, a general limitation of this approach: if there were regions where all silent variants displayed consistent shifts, these would be normalized away and treated as baseline. This could, in principle, mask subtle position-dependent biology. In the case of SMARCB1, both external datasets (GTEx eQTLs, gnomAD population frequencies) and our internal functional screen strongly indicate that silent mutations are overwhelmingly neutral, making it unlikely that biologically meaningful effects were obscured. Nonetheless, this caveat reflects an inherent trade-off of rolling-average normalization and should be considered when applying similar methods to other systems.

We have since added the following lines to the results (lines 178-182):

“To account for regional technical artifacts where variants showed systematic shifts in log₂ fold-change values (**Extended Fig 1o**), we applied a rolling z-score normalization that compares each variant to local silent mutation behavior rather than global averages, under the assumption that silent mutations have no effect on SMARCB1 antiproliferative function (**Methods**).”

Reviewer #5 (Remarks on code availability):

I marked "Yes" because I did look at their markdown code in github, but I did not attempt to clone and reproduce. The code looks reasonable and has adequate comments for those familiar with the types of data being handled, in my opinion.

References:

1. Guidi, C. J., Veal, T. M., Jones, S. N. & Imbalzano, A. N. Transcriptional compensation for loss of an allele of the *Ini1* tumor suppressor. *J. Biol. Chem.* **279**, 4180–4185 (2004).
2. Radko-Juettner, S. *et al.* Targeting DCAF5 suppresses SMARCB1-mutant cancer by stabilizing SWI/SNF. *Nature* **628**, 442–449 (2024).
3. Pan, J. *et al.* The ATPase module of mammalian SWI/SNF family complexes mediates subcomplex identity and catalytic activity-independent genomic targeting. *Nat. Genet.* **51**, 618–626 (2019).
4. Hong, A. L. *et al.* Renal medullary carcinomas depend upon SMARCB1 loss and are sensitive to proteasome inhibition. *eLife* **8**, e44161 (2019).

5. Michel, B. C. *et al.* A non-canonical SWI/SNF complex is a synthetic lethal target in cancers driven by BAF complex perturbation. *Nat. Cell Biol.* **20**, 1410–1420 (2018).
6. Ioannidis, N. M. *et al.* REVEL: An Ensemble Method for Predicting the Pathogenicity of Rare Missense Variants. *Am. J. Hum. Genet.* **99**, 877–885 (2016).
7. Rentzsch, P., Witten, D., Cooper, G. M., Shendure, J. & Kircher, M. CADD: predicting the deleteriousness of variants throughout the human genome. *Nucleic Acids Res.* **47**, D886–D894 (2019).
8. Cheng, J. *et al.* Accurate proteome-wide missense variant effect prediction with AlphaMissense. *Science* **381**, eadg7492 (2023).
9. Howard, T. P. *et al.* MDM2 and MDM4 Are Therapeutic Vulnerabilities in Malignant Rhabdoid Tumors. *Cancer Res.* **79**, 2404–2414 (2019).
10. Knutson, S. K. *et al.* Durable tumor regression in genetically altered malignant rhabdoid tumors by inhibition of methyltransferase EZH2. *Proc. Natl. Acad. Sci.* **110**, 7922–7927 (2013).
11. Lind, P. A., Arvidsson, L., Berg, O. G. & Andersson, D. I. Variation in Mutational Robustness between Different Proteins and the Predictability of Fitness Effects. *Mol. Biol. Evol.* **34**, 408–418 (2017).
12. Sack, L. M., Davoli, T., Xu, Q., Li, M. Z. & Elledge, S. J. Sources of Error in Mammalian Genetic Screens. *G3 GenesGenomesGenetics* **6**, 2781–2790 (2016).
13. Hayes, T. K. *et al.* Comprehensive mutational scanning of EGFR reveals TKI sensitivities of extracellular domain mutants. *Nat. Commun.* **15**, 2742 (2024).
14. Yang, X. *et al.* Defining protein variant functions using high-complexity mutagenesis libraries and enhanced mutant detection software ASMv1.0. *bioRxiv* 2021.06.16.448102 (2022) doi:10.1101/2021.06.16.448102.
15. Boyd, C. *et al.* Alterations in the SMARCB1 (INI1) tumor suppressor gene in familial schwannomatosis. *Clin. Genet.* **74**, 358–366 (2008).
16. Allen, M. D., Freund, S. M. V., Zinzalla, G. & Bycroft, M. The SWI/SNF Subunit INI1 Contains an N-Terminal Winged Helix DNA Binding Domain that Is a Target for Mutations in Schwannomatosis. *Struct. Lond. Engl.* 1993 **23**, 1344–1349 (2015).
17. Hadfield, K. D. *et al.* Molecular characterisation of SMARCB1 and NF2 in familial and sporadic schwannomatosis. *J. Med. Genet.* **45**, 332–339 (2008).

18. Kresak, J. L. & Walsh, M. Neurofibromatosis: A Review of NF1, NF2, and Schwannomatosis. *J. Pediatr. Genet.* **5**, 98–104 (2016).
19. Yang, X. *et al.* A public genome-scale lentiviral expression library of human ORFs. *Nat. Methods* **8**, 659–661 (2011).
20. Mashtalir, N. *et al.* Modular Organization and Assembly of SWI/SNF Family Chromatin Remodeling Complexes. *Cell* **175**, 1272–1288.e20 (2018).
21. Lozano, G., Prives, C. & Sabapathy, K. Mutant p53 Gain of Function: Why Many See It, Why Some Do Not. *Cancer Discov.* **15**, 1099–1104 (2025).
22. Kaelin, W. G. Common pitfalls in preclinical cancer target validation. *Nat. Rev. Cancer* **17**, 441–450 (2017).
23. Giacomelli, A. O. *et al.* Mutational processes shape the landscape of TP53 mutations in human cancer. *Nat. Genet.* **50**, 1381–1387 (2018).
24. Mighell, T. L., Thacker, S., Fombonne, E., Eng, C. & O’Roak, B. J. An Integrated Deep-Mutational-Scanning Approach Provides Clinical Insights on PTEN Genotype-Phenotype Relationships. *Am. J. Hum. Genet.* **106**, 818–829 (2020).
25. Flores-Toro, J. A. *et al.* The Childhood Cancer Data Initiative: Using the Power of Data to Learn From and Improve Outcomes for Every Child and Young Adult With Pediatric Cancer. *J. Clin. Oncol. Off. J. Am. Soc. Clin. Oncol.* **41**, 4045–4053 (2023).
26. Schubach, M., Maass, T., Nazaretyan, L., Röner, S. & Kircher, M. CADD v1.7: using protein language models, regulatory CNNs and other nucleotide-level scores to improve genome-wide variant predictions. *Nucleic Acids Res.* **52**, D1143–D1154 (2024).
27. Ghandi, M. *et al.* Next-generation characterization of the Cancer Cell Line Encyclopedia. *Nature* **569**, 503–508 (2019).
28. Tan, K.-T. *et al.* Haplotype-resolved germline and somatic alterations in renal medullary carcinomas. *Genome Med.* **13**, 114 (2021).
29. The GTEx Consortium *et al.* The GTEx Consortium atlas of genetic regulatory effects across human tissues. *Science* **369**, 1318–1330 (2020).

We thank the reviewers for their evaluation of the revised manuscript and their overall positive assessment. Below we address the remaining comments from Reviewer 4.

(Remarks to the Author):

Reviewer #1 (Remarks to the Author):

The reviewers have answered all my questions.

Reviewer #1 (Remarks on code availability):

The reviewers have answered all my questions.

Reviewer #2 (Remarks to the Author):

After reading through the response to reviewers it is clear that the authors have moderated the manuscript to acknowledge limitations of the expression system used and differences in its use to detect phenotypes relating to CSS or cancer. The evidence is also strengthened that the mutations characterised in more detail act by disrupting the integrity of complexes. As a result I consider the revised manuscript suitable for publication.

Reviewer #3 (Remarks to the Author):

The effort the authors took to address the reviewer comments is appreciated. Overall, the authors sufficiently addressed my own comments.

Reviewer #4 (Remarks to the Author):

The manuscript is improved in several ways, e.g. by removing the section on WHD flexibility and replacing it with a stronger Figure 4 about their new RPT2 molecular dynamics simulations.

To me, the biggest issue is that it remains unclear that the data quality achieves the level of a resource wherein the screen can be trusted to provide information about all the missense mutations. However, I don't think this issue is reason to reject the paper, because they thoroughly follow up to validate a handful of their hits and demonstrate what some of these missense mutations do to the complex and its impact on chromatin (i.e. the DMS clearly finds true positives). I appreciate the additions they made to the figures, Methods, and Discussion that address my previous points.

Major comment:

1. Screen noise seems unfortunately high, as now shown in Ext Fig 2A, despite high coverage growth screens in other contexts providing very reproducible data.

a. I suspect this technical variability has to do with some combination of lentiviral recombination, PCR recombination, potentially too little growth time, and potentially too low cellular coverage (during growth or genomic PCR). The shape of the reproducibility plots seems suggestive of some kind of bottlenecking or jackpot effect with elements highly enriched in a single replicate.

b. However, I recognize higher cell coverage can be especially difficult with adherent cell lines. To me, 8 days seems short for a growth screen, but I note the authors have had a different experience (mentioned in the rebuttal). It looks like MOI was in a good range to avoid multiple infections (19.5-35.4%) and did not end up at the high end of MOI=0.6, which would have resulted in 25% multiple-infected cells. And, overall mutation effect calling is largely similar across cell lines, as shown in the rebuttal, but that is largely for nonsense/frameshift mutations which are, in a sense, positive controls with expected large effects.

c. Altogether, I still feel the authors are, in some points of the paper and rebuttal (but not everywhere), overstating the quality of the DMS. For example, they ascribe the lack of clear patterns in the DMS to a biologically meaningful feature of SMARCB1; this would be an unexpected result given the biochemical similarity of many substitutions. They suggest plausible reasons there could truly be exquisite sensitivity to small side chain differences in SMARCB1, but the simpler explanation is technical noise.

We appreciate the reviewer's thoughtful perspective on the technical variability of the DMS and the caution raised regarding interpretation of mutational patterns. We agree that technical noise likely contributes to the observed heterogeneity in the screen and have revised the manuscript to present the DMS data more conservatively, avoiding overinterpretation of variant-level patterns.

We have made the following changes to the text:

Removed lines 48-50 from revised manuscript in the abstract:

"By establishing a high-throughput functional framework, this study offers a critical resource for elucidating *SMARCB1*'s mutational landscape and its implications for cancer diagnostics."

Removed lines 103-105 from revised manuscript in the Introduction:

"These results provide a functional map of *SMARCB1* variants offering a valuable resource for clinical interpretation and advancing our understanding of SWI/SNF biology in cancer."

Added the following text to lines 210-211 in the Results:

"The variability observed among missense mutations may reflect both biological sensitivity to specific substitutions and technical noise inherent to the screening approach."

Added the following text to lines 278-280 in the Results:

“While these patterns suggest specific biochemical constraints, we note that technical variability in the screen necessitates experimental validations of individual variants to confirm functional effects.”

Minor comment:

1. Is there an internal alternative start site that explains the low impact of nonsense mutations in the N-terminal region?

We thank the reviewer for raising this point. We have now expanded the Discussion to consider this possibility. Specifically, we note the presence of several downstream methionine residues serving as possible alternative start sites; however, we emphasize that this explanation is unlikely to be exhaustive.

We have added the following text to lines 656-660:

“Our DMS data revealed that N-terminal nonsense mutations retain partial to full antiproliferative function. While the precise mechanism underlying this observation remains to be established, we note the presence of 5 methionine residues in the N-terminal region which could serve as alternative start sites. However, alternative start site usage alone is unlikely to fully account for this observation, and additional mechanisms are likely.”